# On the Design of One-step Diffusion via Shortcutting Flow Paths

**Haitao Lin**[*1], **Peiyan Hu**[*1,2] **& Minsi Ren**[1]
{linhaitao, hupeiyan, renminsi}@westlake.edu.cn

**Zhifeng Gao**[3],
gaozf@dp.tech

**Zhi-Ming Ma**[2], **Guolin Ke**[†3], **Tailin Wu**[†1] **& Stan Z. Li**[†1]
mazm@amt.ac.cn      kegl@dp.tech      {wutailin, stan.zq.li}@westlake.edu.cn

[1]Department of Artificial Intelligence, School of Engineering, Westlake University;
[2]Academy of Mathematics and Systems Science, Chinese Academy of Sciences;
[3]DP Technology, Beijing.

 **Code Repository:** https://github.com/EDAPINENUT/ExplicitShortCut.git
 **Project page:** https://edapinenut.github.io/explicitshortcut-project-page

## Abstract

Recent advances in few-step diffusion models have demonstrated their efficiency and effectiveness by shortcutting the probabilistic paths of diffusion models, especially in training one-step diffusion models from scratch (*a.k.a.* shortcut models). However, their theoretical derivation and practical implementation are often closely coupled, which obscures the design space. To address this, we propose a common design framework for representative shortcut models. This framework provides theoretical justification for their validity and disentangles concrete component-level choices, thereby enabling systematic identification of improvements. With our proposed improvements, the resulting one-step model achieves a new state-of-the-art FID50k of 2.85 on ImageNet-256×256 under the classifier-free guidance setting with one step generation, and further reaches FID50k of 2.53 with 2× training steps. Remarkably, the model requires no pre-training, distillation, or curriculum learning. We believe our work lowers the barrier to component-level innovation in shortcut models and facilitates principled exploration of their design space.

## 1 Introduction

Diffusion-based models have become the dominant paradigm in deep generative modeling (Sohl-Dickstein et al., 2015; Ho et al., 2020; Song et al., 2020), progressively transforming samples from a prior distribution toward the data distribution. However, dozens or even hundreds of neural function evaluations (NFEs) are typically required, resulting in slow inference and limited real-time use (Song & Ermon, 2020; Salimans & Ho, 2022; Lu et al., 2025; Zheng et al., 2023). Consistency models (Song et al., 2023; Song & Dhariwal, 2023) are pioneering works that attempt to achieve one-step generation (Luo et al., 2023; Wang et al., 2023; Yin et al., 2024a;c; Salimans et al., 2024; Geng et al., 2023; 2025b), but a costly two-stage training process is required, *i.e.*, first training a reliable diffusion model and then distilling velocity or score from it. Despite the costly two-stage training, they offer fast generation, which motivates further research into improving training efficiency.

Recently, one-step diffusion models trained from scratch have emerged, such as Consistency Training (CT) (Song et al., 2023) as the training-from-scratch variant of consistency models, Inductive Moment Matching (IMM) (Zhou et al., 2025), and Shortcut Diffusion (SCD) (Frans et al., 2025). These models aim to learn direct shortcut mappings between intermediate states along the probability flow trajectories of the probability flow, thus enabling one-step generation; we refer to such models as *shortcut models*. Building on this principle, continuous-time shortcut models such as sCT (Lu & Song, 2025) and MeanFlow (Geng et al., 2025a) have been introduced, achieving state-of-the-art performance in one-step generation for image synthesis. Their efficiency and effectiveness in both training and generation have stimulated further exploration in improving their sampling fidelity.

---

[*]Equal contribution.
[†]Corresponding authors.

Although these models share the same objective, the barrier to understanding the working mechanisms remains non-trivial. Specifically, the literature on them is dense on theory, derivations of method formulations and the corresponding learning objectives, as well as technical details like time samplers and curriculum, and training tricks, *etc.*, leading to a less intuitive design paradigm. As a result, it may inadvertently obscure the underlying design space, making each carefully crafted module appear indispensable, so that altering a single component seems to threaten the integrity of the entire system.

Therefore, we first contribute to *proposing a common design framework for these shortcut models from a practical standpoint*. We summarize that both discrete- and continuous-time variants share the principle of approximating two-step flow map targets with one-step parameterized predictions. We also provide a general theoretical justification for the validity of this design paradigm. This framework allows us to disentangle the concrete modules within these models, offering clearer insights into how the components interact and what flexibility remains in shaping the overall method design.

Secondly, our contribution lies in *elucidating the design space of shortcut models*. We decompose each model into distinct modules aligned with their learning objectives, and then conduct an in-depth empirical investigation and theoretical analysis of different module combinations. In summary, we demonstrate the advantages of linear paths in settings of shortcut model trained from scratch, discuss the scenarios where continuous-time variants exhibit superior sampling fidelity over discrete-time ones, and figure out the impacts of time samplers on training convergence.

Further, the third set of contributions centers on *improvements to the training of continuous-time shortcut models*. Building on the previous analysis, we introduce three technical refinements for enhancing training stability: (i) the use of plug-in velocity and its correction under classifier-free-guidance training, (ii) a gradual time sampler, and (iii) several established training techniques such as variational adaptive loss weighting. Our experiments demonstrate that these techniques consistently improve performance. Finally, we conduct a scaling-up evaluation on ImageNet-256×256. By incorporating the proposed improvements into our modeling framework, we achieve an FID50k of 2.85 under one-step generation, setting a new state of the art among shortcut models trained from scratch. We believe that our work facilitates component-level innovation and thereby enables more systematic and targeted exploration of the design space of shortcut models.

## 2 EXPRESSING ONE-STEP DIFFUSION THROUGH SHORTCUT MODELS

### 2.1 SHORTCUTTING FLOWS WITH FLOW MAP SOLVERS

**Diffusion models.** Let $p_{\text{data}}(\boldsymbol{x})$ be the data distribution, and $p_{\text{prior}} = \mathcal{N}(\boldsymbol{0}, \sigma^2 \mathbf{I})$ be a Gaussian distribution with zero mean and variance $\sigma^2$. In the following, we write $\sigma = 1$ by default for notational simplicity. According to stochastic interpolants (Albergo et al., 2023), diffusion models establish a probabilistic path between $p_0 = p_{\text{data}}$ and $p_1 = p_{\text{prior}}$ such that $\boldsymbol{x}_t = \alpha_t \boldsymbol{x}_0 + \sigma_t \boldsymbol{\varepsilon}$, where $\boldsymbol{x}_0 \sim p_0, \boldsymbol{\varepsilon} \sim p_1$, and $\alpha_t, \sigma_t \geq 0$; with boundary conditions $\alpha_0 = \sigma_1 = 1$ and $\alpha_1 = \sigma_0 = 0$. Both the forward noising and inverse denoising processes are governed by the probability flow ODE (PF-ODE) as $\dot{\boldsymbol{x}}_t = \boldsymbol{v}_t(\boldsymbol{x}_t)$, where $\boldsymbol{v}_t(\boldsymbol{x})$ is the marginal *velocity* $\boldsymbol{v}_t(\boldsymbol{x}) = \dot{\alpha}_t \mathbb{E}(\boldsymbol{x}_0 | \boldsymbol{x}_t = \boldsymbol{x}) + \dot{\sigma}_t \mathbb{E}(\boldsymbol{\varepsilon} | \boldsymbol{x}_t = \boldsymbol{x})$.

**Flow paths.** Probabilistic paths satisfying the above are defined as flow paths. For example, with the reformulation by Lu & Song (2025), the EDM preconditioner path (Karras et al., 2022) can be transformed to a cosine path (Ma et al., 2024) with $\sigma = \sigma_{\text{data}}$, $\alpha_t = \cos(\frac{\pi}{2} t)$, and $\sigma_t = \sin(\frac{\pi}{2} t)$; in Rectified Flow (Liu et al., 2022), $\sigma = 1$, $\alpha_t = 1 - t$ and $\sigma_t = t$, leading to the linear path (Lipman et al., 2023; Tong et al., 2024). Since $\boldsymbol{v}_t(\boldsymbol{x})$ is inaccessible, the conditional path is established for tractable training, where the corresponding conditional velocity is $\boldsymbol{v}_{t|0} = \boldsymbol{v}_t(\boldsymbol{x}_t | \boldsymbol{x}_0) = \dot{\alpha}_t \boldsymbol{x}_0 + \dot{\sigma}_t \boldsymbol{\varepsilon}$ that neural networks $F^\theta(\boldsymbol{x}_t, t)$ are trained to approximate. In sampling, one can first sample $\boldsymbol{x}_1 = \boldsymbol{\varepsilon} \sim p_{\text{prior}}$, and then simulate a trajectory of the flow through the PF-ODE as $\dot{\boldsymbol{x}}_t = F^\theta(\boldsymbol{x}_t, t)$.

**Flow maps.** In order to shortcut established flow paths from time $t$ to $r$ ($0 \leq r \leq t \leq 1$), we introduce the flow map notation (Boffi et al., 2025b; Liu, 2025; Boffi et al., 2025a) to express the design frame for simplicity. A flow map $X_{t,r}$ is defined as the unique map such that $X_{t,r}(\boldsymbol{x}_t) = \boldsymbol{x}_r$, for all $(t, r) \in [0, 1]^2$, where $\boldsymbol{x}_r$ is the solution of PF-ODE, which corresponds to *position* in physics.

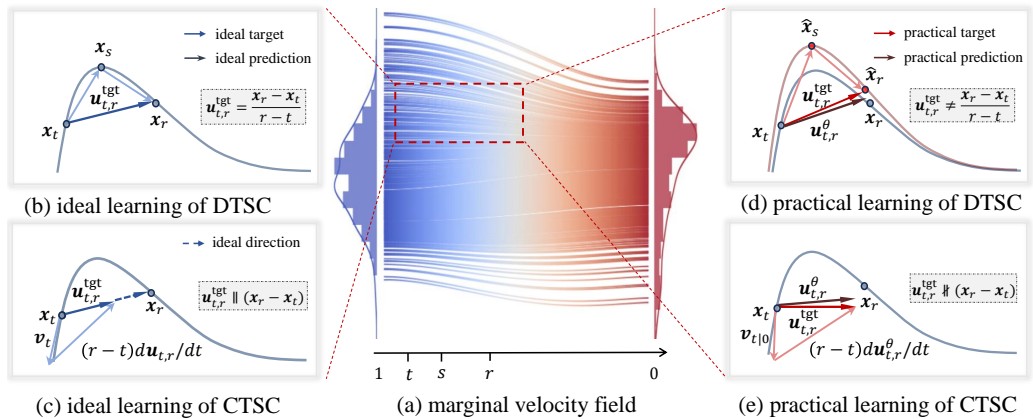

Figure 1: The physical picture of ideal and practical learning of discrete- and continuous-time shortcut models (DTSC&CTSC) where $\boldsymbol{u}_{t,r}^{\text{tgt}}$ denotes the target obtained by the two-step flow maps, and $\boldsymbol{u}_{t,r}^{\theta}$ is the models' prediction for one-step flow maps. (a) shows the marginal velocity field from $\mathcal{N}(0,1)$ to a Gaussian Mixture. (b) and (c) illustrate the ideal learning of DTSC and CTSC, where $\boldsymbol{x}_r$ is sampled from the same trajectory of PF-ODE, and thus $\boldsymbol{u}_{t,r}^{\text{tgt}}$ serves as the correct supervisory signal for training. (d) and (e) depict the practical learning of DTSC and CTSC, where the targets deviate from the trajectory, thus leading to models' prediction drifts away correspondingly.

According to the PF-ODE, one can easily derive the flow map solution through

$$\boldsymbol{x}_r = X_{t,r}(\boldsymbol{x}_t) = \boldsymbol{x}_t + \int_t^r \boldsymbol{v}_\tau(\boldsymbol{x}_\tau)d\tau, \tag{1}$$

where $\int_t^r \boldsymbol{v}_\tau(\boldsymbol{x}_\tau)d\tau$ corresponds to *displacement* in physics.

**Flow map solvers.** With the definition of *average velocity* over time (Geng et al., 2025a) as $\boldsymbol{u}_{t,r}$, we can rewrite Eq. 1 to express the flow map solution $\boldsymbol{x}_r = X_{t,r}(\boldsymbol{x}_t)$ through

$$X_{t,r}(\boldsymbol{x}_t) = \boldsymbol{x}_t + (r-t) \cdot \boldsymbol{u}_{t,r}(\boldsymbol{x}_t)$$
$$\text{where} \quad \boldsymbol{u}_{t,r}(\boldsymbol{x}_t) = \frac{1}{r-t}\int_t^r \boldsymbol{v}_\tau(\boldsymbol{x}_\tau)\,d\tau, \tag{2}$$

or to infer $X_{t,r}(\boldsymbol{x}_t)$ with the *instantaneous velocity* $\boldsymbol{v}_t$, through DDIM-solver (Song et al., 2021) as first-order approximation of DPM-solver (Lu et al., 2022), which reads

$$X_{t,r}(\boldsymbol{x}_t) \approx \text{DDIM}(\boldsymbol{x}_t, \boldsymbol{v}_t, t, r) = \bar{\alpha}_{t,r}\boldsymbol{x}_t + \bar{\beta}_{t,r}\boldsymbol{v}_t, \tag{3}$$

where $\bar{\alpha}_{t,r} = \cos(\frac{\pi}{2}(r-t))$ and $\bar{\beta}_{t,r} = \frac{2}{\pi}\sin(\frac{\pi}{2}(r-t))$ in cosine paths; and $\bar{\alpha}_{t,r} = 1$ and $\bar{\beta}_{t,r} = r-t$ in linear paths. The general formulation and detailed derivation are given in Appendix A.2.

With the solvers, if a model learns the solution to the flow maps from any $t$ to $r$, it can bypass the costly iterative procedure and achieve one-step generation by predicting $X_{1,0}^{\theta}(\boldsymbol{x}_1)$.

## 2.2 LEARNING TO SHORTCUT FLOW PATHS

**Overall design frame.** We claim that the previous methods shortcut the flow paths of a diffusion model by regularizing a one-step flow map prediction against the two-step flow map target. In practice, they first sample time points $r, s, t \sim p(\tau)$ with $r \leq s \leq t$, and then use the consistency property (Liu, 2025; Boffi et al., 2025b) as detailed in Appendix A.1 to design a shortcut model trained from scratch, which reads

$$X_{s,r}(X_{t,s}(\boldsymbol{x}_t)) = X_{t,r}(\boldsymbol{x}_t). \tag{4}$$

Specifically, these methods aim to construct a two-step flow map target from $t$ to $s$, then to $r$, *i.e.*, $X_{s,r} \circ X_{t,s}(\boldsymbol{x}_t)$, and then make the parameterized flow map $X_{t,r}^{\theta}(\boldsymbol{x}_t)$ to approximate this target in a

single step. It allows the model to achieve one-step generation by predicting $\boldsymbol{x}_0^\theta = X_{1,0}^\theta(\boldsymbol{x}_1)$ with $\boldsymbol{x}_1 = \boldsymbol{\varepsilon} \sim p_1$. As a result, their learning objectives $\mathcal{L}$ can be expressed as

$$\arg\min_\theta \mathbb{E}_{r,s,t\sim p(\tau),\, \boldsymbol{x}_t\sim p_t} \big[ \underbrace{w(r,s,t) \cdot d(\ \overbrace{X_{t,r}^\theta(\boldsymbol{x}_t)}^{\text{one-step prediction}}\ ,\ \overbrace{\mathrm{sg}(\hat{X}_{s,r} \circ \hat{X}_{t,s}(\boldsymbol{x}_t))}^{\text{two-step target}}\ )}_{l(\boldsymbol{x}_t,r,s,t;\theta)} \big], \qquad (5)$$

where $w$ is the weight term, $\hat{X}$ and $X^\theta$ are flow maps obtained with the conditional velocity or the neural network $F^\theta$, $d(\cdot,\cdot)$ is a loss metric function, such as the squared $l_2$-distance, and $\mathrm{sg}(\cdot)$ is the stop gradient operator in backpropagation. We call $\hat{X}_{s,r} \circ \hat{X}_{t,s}(\boldsymbol{x}_t)$ two-step flow map targets and $X_{t,r}^\theta(\boldsymbol{x}_t)$ one-step flow map predictions, and write the inner loss term of expectation as $l(\boldsymbol{x}_t,r,s,t;\theta)$.

**Time sampler.** To construct the training objective, time points $\{r,s,t\}$ are sampled with $r \le s \le t$. We refer to this as the discrete-time shortcut model (DTSC) when $\{r,s,t\}$ are discrete time points. For example, in CTs, $r$ is fixed at 0, and $t$ and $s$ are sampled from a non-uniform discretization curriculum that gradually changes from sparse to dense such that $t$ is always chosen to be one time step ahead of $s$; SCD divides the time interval into equal segments based on different powers of 2, and samples uniformly between adjacent grid points with spacing $h$, as denoted by $(t,h) \sim \mathrm{Uniform}\log_2(t,h)$ ; IMM samples time with $r$ and $t$ uniformly from $[0,1]$, and $\{s,t\}$ separated by a fixed gap. As the gap between two time points becomes infinitesimal, the discrete-time shortcut model converges to a continuous-time form (CTSC). For example, sCTs and MeanFlows recover this by setting $s \to t$.

**Network parameterization and flow map solution.** We denote by $F^\theta$ the neural network with parameters $\theta$, whose architecture is instantiated as U-Net (Song et al., 2020) in the pixel space, or as DiT (Peebles & Xie, 2022) / SiT (Ma et al., 2024) in the latent space. To obtain the flow map, Eq. 1, Eq. 2, and Eq. 3 can all serve as solutions. Since the integral term $\int_t^r \boldsymbol{v}_\tau(\boldsymbol{x}_\tau)d\tau$ in Eq. 1 is intractable in general, the DDIM solver with instantaneous velocity is adopted practically when estimating the flow map with $\boldsymbol{v}_t^\theta$ parameterized by $F^\theta$ or the conditional velocity $\boldsymbol{v}_{t|0}$ through Eq. 3. Alternatively, if $F^\theta$ parameterizes average velocity $\boldsymbol{u}_{t,r}^\theta$, the flow map can be obtained directly through Eq. 2.

## 2.3 Examples: discrete- and continuous-time shortcut models

**Discrete-time shortcut models.** CTs, SCDs and IMMs are representative DTSCs. If parameterizing velocity with neural networks as $F^\theta(\boldsymbol{x}_t,t) = \boldsymbol{v}_t^\theta(\boldsymbol{x}_t)$, we can then adopt the DDIM as flow map solvers. Specifically, we first use the parameterized velocity $\boldsymbol{v}_t^\theta(\boldsymbol{x}_t)$ to solve the flow map $\boldsymbol{x}_r^\theta = X_{t,r}^\theta(\boldsymbol{x}_t)$, which serves as the one-step prediction. For the two-step target, we alternate between the conditional velocity $\boldsymbol{v}_{t|0}$ to obtain $\hat{\boldsymbol{x}}_s = \hat{X}_{t,s}(\boldsymbol{x}_t)$, and the parameterized velocity $\boldsymbol{v}_s^\theta(\hat{\boldsymbol{x}}_s)$ to obtain $\hat{\boldsymbol{x}}_r = \hat{X}_{s,r}(\hat{\boldsymbol{x}}_s)$. In this way, we can derive $l(\boldsymbol{x}_t,r,s,t;\theta)$ in Eq. 5 as

$$l_{\mathrm{ct}}(\boldsymbol{x}_t,r,s,t;\theta) = \mathrm{LPIPS}\Big(\mathrm{DDIM}(\boldsymbol{x}_t,\boldsymbol{v}_t^\theta(\boldsymbol{x}_t),t,r),\ \mathrm{sg}\big(\mathrm{DDIM}(\hat{\boldsymbol{x}}_s,\boldsymbol{v}_s^\theta(\hat{\boldsymbol{x}}_s),s,r)\big)\Big), \qquad (6)$$

where the loss metric is LPIPS (Zhang et al., 2018) applied directly in pixel space with $w = 1$, and $l_{\mathrm{ct}}(\boldsymbol{x}_t,r,s,t;\theta)$ coincides with its formulation in CTs with details shown in Appendix B.1. Alternatively, if we parameterize the average velocity as $\boldsymbol{u}_{t,r}^\theta(\boldsymbol{x}_t) = F^\theta(\boldsymbol{x}_t,t,r)$, and estimate both the one-step prediction and the two steps in the target with neural networks $F^\theta$, we thus obtain the $l(\boldsymbol{x}_t,r,s,t;\theta)$ in SCD through Eq. 2. Due to equi-spacing time points as $t - s = s - r = h$, it reads

$$l_{\mathrm{scd}}(\boldsymbol{x}_t,r,s,t;\theta) = \left\| \boldsymbol{u}_{t,r}^\theta(\boldsymbol{x}_t) - \frac{1}{2}\mathrm{sg}\big(\boldsymbol{u}_{t,s}^\theta(\boldsymbol{x}_t) + \boldsymbol{u}_{s,r}^\theta(\hat{\boldsymbol{x}}_s)\big) \right\|_2^2, \qquad (7)$$

where the loss metric is set as squared $l_2$-distance, $w = \frac{1}{4h^2}$, and $\hat{\boldsymbol{x}}_s = \boldsymbol{x}_t + (s-t)\boldsymbol{u}_{s,t}^\theta(\boldsymbol{x}_t)$, with details shown in Appendix B.2. Moreover, for IMMs, conditional samples $\{\boldsymbol{x}_0^{(i)},\boldsymbol{\varepsilon}^{(i)}\}_{i=1}^B$ within a mini-batch of size $B$ are first partitioned into different groups, and $\{r,s,t\}$ are drawn for each group. With similar flow map construction to CTs, the loss metric is implemented with a grouped kernel function as to minimize MMD (Gretton et al., 2012), applied to measure both inter- and intra-sample similarities between $\{\hat{\boldsymbol{x}}_r,\boldsymbol{x}_r^\theta\}$ within the group, as further detailed in Appendix B.3.

Table 1: Specific design choices employed by different shortcut models. 'sg EMA decay' means that the parameters $\theta$ in the stop-gradient targets are updated in a delayed manner with EMA.

| | | CT | SCD | IMM | sCT(note: △) | MeanFlow |
|---|---|---|---|---|---|---|
| **Diffusion basis†** | | | | | | |
| Flow path | | Cosine | Linear | Linear | Cosine | Linear |
| Network $F^\theta$ | Architecture | U-Net | DiT | DiT | U-Net | DiT |
| | Output | $v^\theta$ | $u^\theta$ | $v^\theta$ | $v^\theta$ | $u^\theta$ |
| **Flow map construction** | | | | | | |
| Time sampler | | (note: *) $t = \frac{\pi}{2}\arctan([\sigma_{\max}^{1/\rho} + \frac{\tau}{K}(\sigma_{\min}^{1/\rho} - \sigma_{\max}^{1/\rho})]^\rho)$ $s = \frac{\pi}{2}\arctan([\sigma_{\max}^{1/\rho} + \frac{\tau+1}{K}(\sigma_{\min}^{1/\rho} - \sigma_{\max}^{1/\rho})]^\rho)$ $r = 0$, where $\tau \sim \mathcal{U}\{0,\dots,K-1\}$ | $t = \tau$ $s = \tau - h$ $r = \tau - 2h$ and with $p_{\text{teq}}$, $r = s = t$, where $\tau, h \sim$ Uniform $\log_2(\tau, h)$ | (note: ⋆) $t \sim \mathcal{U}[0,1]$ $n_s = \frac{1}{1-t} - \frac{1}{2^\gamma}$ $s = \frac{n_s}{n_s+1}$ and $r \sim \mathcal{U}[0,t]$ | $t = \frac{2}{\pi}\arctan(\exp(\tau))$ $s = t - dt$ $r = 0$, where $\tau \sim \mathcal{N}(P_{\text{mean}}, P_{\text{std}}^2)$ | $r, t = \{\text{sigmoid}(\tau_1),\ \text{sigmoid}(\tau_2)\}$ $s.t.\ r \le t,$ $s = t - dt$, and with $p_{\text{teq}}, r = s = t$, where $\tau_1, \tau_2 \sim \mathcal{N}(P_{\text{mean}}, P_{\text{std}}^2)$ |
| Two-step target: 1st-step ($\hat{x}_s$) | | DDIM($x_t, v_{t\|0}, t, s$) | $x_t - h u_{t,s}^\theta(x_t)$ | DDIM($x_t, v_{t\|0}, t, s$) | (note: ‡) DDIM($x_t, v_{t\|0}, t, s$) | (note: ‡) DDIM($x_t, v_{t\|0}, t, s$) |
| Two-step target: 2nd-step ($\hat{x}_r$) | | DDIM($\hat{x}_s, v_s^\theta, s, r$) | $\hat{x}_s - h u_{s,r}^\theta(\hat{x}_s)$ | DDIM($\hat{x}_s, v_s^\theta, s, r$) | DDIM($\hat{x}_s, v_s^\theta, s, r$) | $\hat{x}_s + (r-s) u_{s,r}^\theta(x_s)$ |
| One-step prediction ($x_r^\theta$) | | DDIM($x_t, v_t^\theta, t, r$) | $x_t - 2h u_{t,r}^\theta(x_t)$ | DDIM($x_t, v_t^\theta, t, r$) | DDIM($x_t, v_t^\theta, t, r$) | $x_t + (r-t) u_{t,r}^\theta(x_t)$ |
| **Training** | | | | | | |
| Loss metric $d$ | | LPIPS | Squared $l_2$-distance | Grouped kernel | Squared $l_2$-distance | Squared $l_2$-distance |
| sg EMA decay | | ✓ | ✗ | ✗ | ✗ | ✗ |

†Demonstration of the configuration on ImageNet. *In CT, $\rho, \sigma_{\max}, \sigma_{\min}$ are adopted from EDM, usually set as 7, 0.001 and 80. $K$ gradually increases from $K_{\min}$ (usually set as 2) to $K_{\max}$ (usually about 200); In CT's original paper, network output's are the score function and the reformulation is given in Appendix A.3. ⋆$\gamma$ is usually set as 12. ‡In sCT and MeanFlow, since $s = t - dt$, which involves differentiation *w.r.t.* $t$, terms in loss metrics are normalized by $dt$. The expression is an intuitive analogy, while the derivation is given in Appendix B. △Although sCT is originally initialized from a teacher diffusion model, we suppose that it can attain comparable performance when trained from scratch, similar to the behavior observed in CT and MeanFlow.

**Continuous-time shortcut models.** When the difference between two time points is infinitesimal, the resulting shortcut models are referred to CTSCs, by setting $s = t - dt$ and normalizing $l(x_t, r, s, t; \theta)$ by $dt$. For instance, MeanFlows are continuous-time shortcut models in which $s = t - dt$. They leverage linear paths with squared $l_2$-distance as the loss metric and parameterizes the average velocity $u_{t,r}(x_t)$ with neural networks $F^\theta$. By writing $l(x_t, r, t - dt, t; \theta) = w \cdot \left\| \frac{d}{dt}\left( X_{t,r}^\theta(x_t) - \hat{X}_{t-dt,r} \circ \hat{X}_{t,t-dt}(x_t) \right) \right\|^2$ and $\frac{d}{dt} u_{t,r}^\theta(x_t) = \partial_t u_{t,r}^\theta(x_t) + (\nabla_x u_{t,r}^\theta)(x_t) \cdot v_t$, and applying Eq. 1 and 2 with approximation shown in Appendix B.4, we correspondingly obtain

$$l_{\text{mf}}(x_t, r, t-dt, t; \theta) = w \cdot \left\| u_{t,r}^\theta(x_t) - \text{sg}\left( v_{t|0} + (r-t)\frac{d u_{t,r}^\theta(x_t)}{dt} \right) \right\|_2^2, \tag{8}$$

under squared $l_2$-distance with adaptive weighting $w$, as detailed in Appendix B.4. Note that there is a predefined probability $p_{\text{teq}}$ such that $r = t$, which results in $l_{\text{mf}} = w\|u_{t,t}^\theta - v_t\|^2$ during training. This training technique of instantaneous conditional velocity supervision is also employed in SCDs.

sCTs, as the continuous-time variants of CTs, use squared $l_2$-distance instead of LPIPS. Under $s = t - dt$, Appendix B.5 shows that the gradient of $l_{\text{ct}}(x_t, r, s, t; \theta)$ *w.r.t.* $\theta$ can be approximated as $\nabla_\theta l(x_t, r, s, t; \theta) \approx \nabla_\theta \|v_t^\theta(x_t) - \text{sg}(v_t^\theta(x_t) + w(t)\frac{d}{dt}X_{t,r}^\theta(x_t))\|_2^2$. By setting $w(t) = \cos(\frac{\pi}{2}t)$,

$$l_{\text{sct}}(x_t, r, t-dt, t; \theta) = \left\| v_t^\theta(x_t) - \text{sg}\left( v_t^\theta(x_t) + w(t)\frac{d\text{DDIM}(x_t, v_t^\theta(x_t), t, r)}{dt} \right) \right\|_2^2, \tag{9}$$

**Remark 2.1.** *sCT with linear paths is of the same form as MeanFlow, as proved in Appendix C.1.*

**Putting it together.** Table 1 summarizes the deterministic variants reproduced from the discussed representative methods, including DTSCs and CTSCs, within our framework. The goal of this reframing is to disentangle the independent components that are often intertwined in prior work. Within our framework, these components can be explicitly separated, such that any reasonable combination of components will yield a functioning model. In practice, the relative effectiveness of different choices and combinations is the focus of our investigation in Sec. 3.

## 2.4 DISCUSSION: SHORTCUTTING FLOW PATHS UNDER MARGINAL VELOCITY FIELDS

### - Q.1: Why share a common design frame?

We inherently aim to simulate the PF-ODE with the *marginal velocity field*, written as $\boldsymbol{v}_t(\boldsymbol{x})$. Consequently, shortcut models essentially operate along the sampling trajectories of the flow governed by $\boldsymbol{v}_t(\boldsymbol{x})$, as shown in Fig. 1(a). Intuitively, the ideal construction of the learning target is to sample two distinct states $\boldsymbol{x}_t$ and $\boldsymbol{x}_r$ along the same curved trajectory from the flow paths, so that the neural network can directly map $\boldsymbol{x}_t$ to $\boldsymbol{x}_r$ such that $F^\theta(\boldsymbol{x}_t, t, r) \approx \boldsymbol{x}_r$, as illustrated in Fig. 1(b) and (c).

However, such pairs $\{\boldsymbol{x}_t, \boldsymbol{x}_r\}$ cannot be obtained via simulation-based sampling: once $\boldsymbol{x}_t$ is sampled, $\boldsymbol{x}_r$ remains inaccessible because both $\boldsymbol{v}_t(\boldsymbol{x})$ and its integral from $t$ to $r$ are intractable. To overcome this, a common design paradigm is employed, which is to let the network's outputs, or the conditional velocity alternatively, estimate $\boldsymbol{x}_r$ in two steps: first producing an intermediate $\hat{\boldsymbol{x}}_s$, and then constructing an estimated target $\hat{\boldsymbol{x}}_r$, as shown in Eq. 5 in Sec. 2.2. This makes training feasible. Although one may also construct multi-step (*i.e.*, more than two) flow map targets for simulating the $\{\boldsymbol{x}_t, \boldsymbol{x}_r\}$ pairs (Kim et al., 2023), the paradigms of two-step target construction approximated by one-step prediction are sufficiently general according to the following theoretical justification with detailed proof in Appendix C.2. Note that we classify the aforementioned methods' training objective into DTSC and CTSC. For example, $l_{\text{mf}}$ and $l_{\text{sct}}$ are instances of $l_{\text{ctsc}}$.

> **Theorem 2.2** (Error bound of DTSC&CTSC (brief)). *Under the mild assumptions with details given in Theorem C.1 of (i) one-sided Lipschitz condition of marginal velocity and (ii) twice continuous differentiability with bounded second derivatives of $X^\theta_{\tau_1,\tau_2}$ for any $\tau_1, \tau_2 \in [0, 1]$. Let $p_0$ the density of $\boldsymbol{x}_0$, and $p_0^\theta$ the density of $\boldsymbol{x}_0^\theta = X^\theta_{1,0}(\boldsymbol{x}_1)$, under the squared $l_2$-distance:*
>
> $$W_2^2(p_0, p_0^\theta) \leq C_1 \mathcal{L}_{dtsc}(\theta) + C_2(t - s); \qquad W_2^2(p_0, p_0^\theta) \leq C_3 \mathcal{L}_{ctsc}(\theta),$$
>
> *where we write the training objective in Eq. 5 as $\mathcal{L}_\bullet(\theta) = \mathbb{E}_{r,s,t\sim p(\tau),\,\boldsymbol{x}_t \sim p_t}[l_\bullet(\boldsymbol{x}_t, r, s, t; \theta)]$ with $\bullet \in \{dtsc, ctsc\}$, $W_2(\cdot, \cdot)$ is the Wasserstein-2 distance, $\{C_1, C_2\}$ are given in Theorem C.3, and $C_3$ is given in Theorem C.4 and C.5 in Appendix C.2.*

### - Q.2: What challenges in constructing flow map targets?

From this perspective, ideal learning for DTSC and CTSC shares a similar physical picture as shown from Fig. 1(b) and (c). However, the practical construction of the two-step flow map target inevitably causes the obtained $\hat{\boldsymbol{x}}_s$ and $\hat{\boldsymbol{x}}_r$ to deviate from $\boldsymbol{x}_s$ and $\boldsymbol{x}_r$ on the sampling trajectory governed by marginal velocity fields as shown in Fig. 1(d) and (e), leading to bias and variance in estimating $\boldsymbol{x}_r$ with $\hat{\boldsymbol{x}}_r$. Introducing this deviation into the supervision of model training greatly affects the performance differences across various shortcut model designs as justified in Prop. 3.1.

### - Q.3: Why distillation from pretrained velocity fields performs better?

From another perspective, this explains why distilling from a pretrained diffusion model is often more effective than training from scratch (Song et al., 2023; Lu & Song, 2025). Unlike (s)CT, which are trained from scratch, (s)CM benefits from distillation by learning from a pretrained velocity field $\boldsymbol{v}^\phi_t(\boldsymbol{x})$. In practical training, the conditional velocity $\boldsymbol{v}_{t|0}$ and network output $\boldsymbol{v}^\theta_t$ in $\text{sg}(\cdot)$ in Eq. 6 and 9 are replaced with $\boldsymbol{v}^\phi_t$, which closely approximates $\boldsymbol{v}_t(\boldsymbol{x}_t)$. This substantially reduces errors in estimating the two-step flow targets, providing more accurate supervision for network training.

## 3 ELUCIDATING THE DESIGN SPACE OF SHORTCUT MODELS

According to our design framework, we analyze existing shortcut models from several key perspectives, including the ***choice of flow path*** and ***design of time sampler***, which primarily determine how the flow map is constructed. In the following, we aim to address several corresponding questions to empirically and theoretically elucidate the design space of one-step shortcut models.

Empirically, we evaluate the proposed formulation using a unified codebase implementation with the same training iterations and batch sizes. For unconditional generation on CIFAR-10, we employ U-Nets (Song et al., 2020) ($\sim$55M param.) as the network architecture operating directly in the pixel space. For conditional generation in ImageNet-256×256, with and without classifier-free guidance, we use a SiT-B/2 (Ma et al., 2024) architecture ($\sim$131M param.), operating in the latent space via

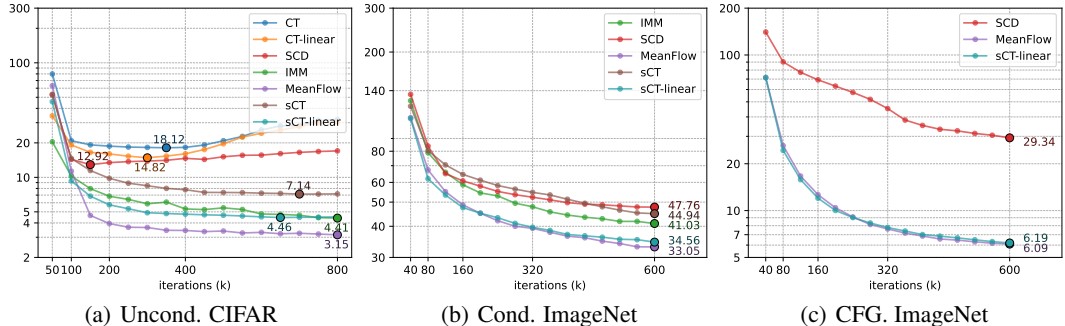

|  (a) Uncond. CIFAR | (b) Cond. ImageNet | (c) CFG. ImageNet |

Figure 2: Comparison of FID50k during training among different shortcut models described in Table 1. (a) is the unconditional (Uncond.) generation on CIFAR-10; (b) is class-conditional (Cond.) generation; and (c) is classifier-free-guidance (CFG.) training on ImageNet-256×256.

a pretrained VQVAE (Rombach et al., 2021). While sCT is originally initialized from the teacher diffusion model as stated in Lu & Song (2025), we train all the discussed models from scratch, for a fair comparison. Fig. 2 summarizes the results of one-step generation on the two datasets, with additional setting of classifier-guidance-free learning, as discussed in Geng et al. (2025a). Further details on settings are provided in Appendix D.1.

*- Q.1: Following linear or cosine paths?*

Linear paths are generally regarded as more analytically tractable and easier to employ for training and sampling tricks (*e.g.*, classifier-free guidance), owing to their simple formulation. By contrast, in pixel-space generative modeling, cosine paths are often considered more stable for training convergence, because they induce a stochastic process with fixed variance. Exploration of these two flow paths in the context of shortcut models remains underexplored. Here we extend cosine-path-based models (CT and sCT) to their linear-path counterparts. Fig. 2 shows that shortcut models with linear paths are more competitive. We attribute this to the fact that the marginal velocity fields generated by linear paths as conditional paths are at lower convex transport cost (Liu et al., 2022), implying lower curvature of the velocity-field-governed trajectories. Consequently, the simulated two-step flow map targets are less likely to deviate from the ideal. Furthermore, while cosine paths are optimal in the setting of diffusion and flow matching (Santos & Lin, 2023) under Fisher information metrics, we theoretically justify that *linear paths in the setting of shortcut models are optimal conditioned on data samples* in Appendix C.3. Based on this, our subsequent analysis will focus on the linear path.

*- Q.2: Shortcutting flow paths discretely or continuously?*

Under the same training setup and within a unified codebase, continuous-time shortcut models clearly outperform their discrete-time counterparts. As shown in Fig. 2(a), both sCT and MeanFlow achieve lower FID50k scores on CIFAR-10 compared to CT and SCD. A similar conclusion can be drawn on ImageNet-256×256 from Fig. 2(b)&2(c). Below, we analyze the inference error of the discussed methods with linear paths in Prop. 3.1, and characterize the regimes in which each objective is preferable. We denote sCT and MeanFlow with linear paths by subscripts ctsc, thanks to their same formulations according to Remark 2.1, and discrete-time models by dtsc as well. In addition, we write the parameterized $v_t^\theta$ in sCT as $u_{t,0}^\theta$ under the linear path according to Appendix C.1.

**Proposition 3.1** (Inference error analysis). *Under regularity conditions shown in Appendix C.4.1, the Wasserstein-2 distance of shortcut models with one-step generation is bounded as:*

$$W_2^2(p_0, p_0^\theta) \leq 2\left(\text{BV}_{ctsc} + 8\text{Var}\left[\frac{d}{dt}u_{t,r}^\theta(\boldsymbol{x}_t)\right] + 8\sigma_{\boldsymbol{v}_{t|0}}^2\right)\Big|_{r=0,t=1}, \tag{10}$$

$$W_2^2(p_0, p_0^\theta) \leq 2\left(\text{BV}_{dtsc} + 8\delta_2^2\,\text{Var}\left[u_{s,r}^\theta(\boldsymbol{x}_t)\right] + 8(1 + \ell^2\delta_2^2)\delta_1^2\,\sigma_{dtsc}^2\right)\Big|_{r=0,t=1}, \tag{11}$$

*where* $\text{BV}_\bullet = \text{Bias}_{\bullet\text{-}tgt}^2 + \text{Bias}_{\bullet\text{-}loss}^2 + 2\text{Var}[u_{t,r}^\theta(\boldsymbol{x}_1)]$ *with* $\bullet \in \{\text{ctsc}, \text{dtsc}\}$, *and* $\text{Bias}_{\bullet\text{-}tgt}^2$ *and* $\text{Bias}_{\bullet\text{-}loss}^2$ *are defined in Prop.C.8 ;* $\delta_1 = t - s$, $\delta_2 = s - r$; $\ell$ *is the local Lipschitz constant of* $u^\theta$; $\sigma_{\boldsymbol{v}_{t|0}}^2$ *is the variance of the conditional velocity, defined by* $\sigma_{\boldsymbol{v}_{t|0}}^2 := \text{Var}(v_t(\boldsymbol{x}_t|\boldsymbol{x}_0))$; $\sigma_{dtsc}^2 = \sigma_{\boldsymbol{v}_{t|0}}^2$ *for CT's two-step flow map targets, or* $\sigma_{dtsc}^2 = \text{Var}[u_{t,s}^\theta(\boldsymbol{x}_t)]$ *when using SCD's flow map targets.*

From Theorem 2.2, for CT and CTSC, we conclude that if $\delta_2^2\text{Var}\big[\boldsymbol{u}_{t,r}^\theta(\boldsymbol{x}_t)\big]$ and $\text{Var}\big[\frac{d}{dt}\boldsymbol{u}_{t,r}^\theta(\boldsymbol{x}_t)\big]$ are of the same order, the right-hand side of Eq. 11 contains an additional term $\ell^2\delta_2^2\delta_1^2\sigma_{\text{dtsc}}^2$ compared with Eq. 10, which is likely to result in higher inference error and instability in training, as the proof in Appendix C.8 shows the inference error already subsumes the training error bound. Further, when $s \to t$, and $\sigma_{\boldsymbol{v}_{t|0}}^2$ dominates both $\delta_2^2\text{Var}\big[\boldsymbol{u}_{t,r}^\theta(\boldsymbol{x}_t)\big]$ and $\text{Var}\big[\frac{d}{dt}\boldsymbol{u}_{t,r}^\theta(\boldsymbol{x}_t)\big]$, the training convergence and sampling fidelity of CTSC and CT are both closely tied to the variance of the conditional velocity used for supervision. Therefore, being able to provide a low-variance velocity supervision during training, such as one obtained from a pretrained neural network, helps to improve shortcut models.

**- Q.3: Fixing the terminal time or not?**

Since sCT-linear is a special case of MeanFlows where the terminal time $r$ is fixed at 0, the empirical results on CIFAR-10 in Fig. 2(a) and on ImageNet-256×256 in Fig. 2(b)&2(c) demonstrate that, in general, random sampling of $r$ is beneficial in capturing the overall shortcut patterns. However, in the early stage of training (approximately before 20–40k epochs), sCT-linear exhibits faster convergence in terms of FID50k for one-step generation. We conjecture that in the early stages, continually adding supervision of $\boldsymbol{x}_0$, akin to a denoising task, provides a simpler learning task that accelerates convergence toward favorable local optima. Yet, without intermediate flow path targets $\boldsymbol{x}_r$ where $r > 0$, the model may remain stuck in these sub-optima during the later training stage.

## 4  IMPROVEMENTS TO TRAINING

Building on the above analysis, all subsequent techniques and developments will be carried out under the *continuous-time* shortcut model with *linear paths*, so we choose MeanFlow with SiT-B/2 architecture as our baseline implementation with its default hyperparameters shown in Appendix D.2. Table 2 presents an ablation study that shows the effectiveness of our improvement techniques, where ESC as explicit&easier shortcut model is the CTSC with all the proposed techniques as follows.

**Plug-in velocity instead of conditional one.**  Since the marginal velocity is intractable, training relies on the conditional velocity, obtained by sampling $\boldsymbol{x}_0$ from the finite training set $\{\boldsymbol{y}^{(i)}\}_{i=1}^N$. Based on it, we derive $\boldsymbol{v}_t^*(\boldsymbol{x}_t|\{\boldsymbol{y}^{(i)}\}_{i=1}^N)$ as the marginal velocity under the empirical data distribution, which we refer to as the ideal velocity in the following:

**Proposition 4.1** (Marginal velocity of empirical distribution and bias-variance comparison). *Assume the data distribution is the empirical distribution, as $p_0(\boldsymbol{y}) = \frac{1}{N}\sum_{i=1}^N \mathbb{1}_{\boldsymbol{y}_i}(\boldsymbol{y})$, the marginal velocity reads*

$$\boldsymbol{v}_t^*(\boldsymbol{x}_t|\{\boldsymbol{y}^{(i)}\}_{i=1}^N) = \sum_i^N \frac{\mathcal{N}(\boldsymbol{x}_t; \alpha_t\boldsymbol{y}^{(i)}, \sigma_t^2\mathbf{I})}{\sum_j^N \mathcal{N}(\boldsymbol{x}_t; \alpha_t\boldsymbol{y}^{(j)}, \sigma_t^2\mathbf{I})}(\dot{\alpha}_t\boldsymbol{y}^{(i)} + \frac{\dot{\sigma}_t}{\sigma_t}(\boldsymbol{x}_t - \alpha_t\boldsymbol{y}^{(i)})), \qquad (12)$$

*where $\boldsymbol{x}_t = \alpha_t\boldsymbol{x}_0 + \sigma_t\varepsilon$. Specifically, under mild assumptions in Prop. C.13 in Appendix C.5.2, substituting $\boldsymbol{v}_t$ in $\mathcal{L}_{ctsc}$ with $\boldsymbol{v}_t^*$ significantly decreases Eq. 10's last term $\sigma_{\boldsymbol{v}_{t|0}} = \mathbb{E}\|\boldsymbol{v}_{t|0} - \boldsymbol{v}_t\|^2$, which reduces the variance by $\mathcal{O}(1 - 1/N)$ while increasing the bias by $\mathcal{O}(1/N)$.*

Replacing the conditional velocity $\boldsymbol{v}_{t|0}$ in Eq. 10 with the ideal velocity obtained from the full training set the variance of the velocity term to $\mathcal{O}(1/N)$. As a result, since $N$ is usually a large number, according to Prop. 2.2, employing the ideal velocity field can therefore provide more stable supervision during training and lower error in inference. However, its computation requires summing over the entire data set, which is infeasible for large-scale data such as ImageNet ($N = 1,281,167$). To address this limitation, we adopt the *plug-in velocity* during training instead, which reads $\boldsymbol{v}_t^*(\boldsymbol{x}_t|\{\boldsymbol{y}^{(i)}\}_{i=1}^B)$. The above computation is restricted to a mini-batch $\{\boldsymbol{y}^{(i)}\}_{i=1}^B$ with pseudocode implementation provided in Algorithm 1. This can be viewed as a mixture of conditional velocities from the mini-batch samples, reducing the level of variance $\sigma_{\boldsymbol{v}_{t|0}}$ in Eq. 10 to $\mathcal{O}(1/B)$, at the minor cost of increased bias. Theoretically, we give further details on the validity of the training objective employing plug-in velocity in Prop. C.15 in Appendix C.6.

**Plug-in velocity under guidance training.**  From the comparison between Fig. 2(b) and Fig. 2(c), it is evident that classifier-free guidance (CFG) is crucial for high-quality image generation (Geng et al.,

Table 2: Evaluation of training improvements under one-step generation with SiT-B/2 as $F^\theta$.

| | Training configuration | FID50k |
|---|---|---|
| | MeanFlow under CFG. (Baseline) | 6.09 |
| +A1 | Plug-in velocity ($p_{\text{plug-in}} = 1.0$) | 6.01 |
| +A2 | Plug-in velocity ($p_{\text{plug-in}} = 0.5$) | 5.98 |
| +B1 | Plug-in velocity ($p_{\text{plug-in}} = 1.0$) & class-consistent batching | 6.08 |
| +B2 | Plug-in velocity ($p_{\text{plug-in}} = 0.5$) & class-consistent batching | 5.96 |
| +C | Gradual time sampler | 5.99 |
| +D | sCM training techniques | 5.95 |
| | **ESC** (Baseline + B2 + C + D) | **5.77** |

**Algorithm 1** Calculation of Plug-in Velocity.

```
# x: training batch (B,D)
# t: sampled time
e = randn_like(x)
xt = (1- t) * x + t * e
x_ex, xt_ex = x[:,None,:], xt[None,:,:]
eps = (xt_ex - (1- t) * x_ex) / t

logp_fn = Normal(0, 1).log_prob
logp = sum(logp_fn(eps), dim=2)
weight = softmax(logp, dim=0)

v_cnd = eps - x_ex
v_plugin = matmul(weight.T, v_cnd)
```

2025a). With CFG, the class-conditional velocity $v_t(x_t|x_0, c)$ leverages instance-level supervision from the label $c$. In contrast, $v_t^*(x_t|\{(y^{(i)}, c^{(i)})\}_{i=1}^B)$, is computed by averaging over randomly drawn mini-batches, which is likely to dilute or erase the class-specific signal. To this end, we employ a plug-in probability $p_{\text{plug-in}}$ that substitutes the conditional velocity with the plug-in velocity, as a trade-off between lowering variance during training and retaining class guidance. The other trick is *class-consistent mini-batching*: When applying CFG during training, we ensure that each mini-batch is sampled within the same class. In multi-GPU training, the class labels of mini-batches across different processes are independent of each other.

**Gradual time sampler from sCT to MeanFlow.**    As discussed in Q.3 from Sec. 3, we design a time-sampling schedule that gradually evolves with training iterations. During the first $K_{\text{fix0}}$ iterations, the sampler selects $r = 0$ with probability $p_{\text{fix0}}$, and with probability $1 - p_{\text{fix0}}$ follows the MeanFlow sampler shown in Table 1. The value of $p_{\text{fix0}}$ decays from 1.0 to 0 under a cosine schedule at the beginning of the training, so that after $K_{\text{fix0}}$ iterations the sampler fully adopts the MeanFlow's strategy, where $K_{\text{fix0}}$ is usually set to 20k in practice.

**Adoption of training techniques.**    Moreover, since sCT can be regarded as a variant of CTSC, several training strategies have already been explored in its original work, such as variational adaptive loss weighting (Karras et al., 2024) and tangent warmup (Lu & Song, 2025). These techniques are also applicable to CTSC and bring performance improvements in the cases given in Appendix D.3.

## 5    SCALING-UP EVALUATION

**Setting.**    In this part, we evaluate the proposed ESC as an improved variant of CTSCs to illustrate its effectiveness at scale. We conduct a scaling-up experiment on ImageNet-256×256 in latent space, and employ SiT-XL/2 (∼676M param.) as the backbone model. We follow the training setting of MeanFlow with CFG, where the model is trained from scratch with 240 epochs (∼1.2M iterations). Furthermore, ESC+ is trained with 480 epochs (∼2.4M iterations). In addition, for CIFAR-10 (Krizhevsky, 2009), all the shortcut models use the same U-Net (Ronneberger et al., 2015) architecture from Song et al. (2020) (∼55M param.). The code repository is provided for reproducibility[1]. For further details on setting, please refer to Appendix D.3.

**Benchmark comparison.**    In Table 3, we compare our results with previous methods by benchmarking the FID50k under one-step generation (1-NFE). In the context of single-step generation, the proposed techniques bring more improvements with the large-scale network architecture (SiT-XL/2) than with the basic one (SiT-B/2), as ESC achieves state-of-the-art performance of an FID50k of 2.85 with 240 epochs and 2.53 with 480 epochs. This represents an improvement of 16.9% and 26.2% compared to the prior one-step result of 3.43 obtained by MeanFlow, respectively, and even better than the two-step generative fidelity of MeanFlow (FID50k 2.93). For visualization of images generated by ESC with different network architectures, please refer to Appendix D.4. Moreover,

---

[1]https://github.com/EDAPINENUT/ExplicitShortCut/

Table 4 gives unconditional generation results on CIFAR-10, showing that our improved models achieve competitive performance with prior approaches. For a full comparison including other families of methods, please refer to Appendix D.6. Notably, we find that the performance gains from ESC with SiT-XL/2 over MeanFlow baseline are much more significant than it with SiT-B/2, which we discuss in Appendix E.2.

Table 3: Evaluation on ImageNet-256×256. Values with underline denote the best except shortcut models, values in **bold** is the best shortcut diffusion model under one-step generation.

| Family | Method | Param. | NFE | FID50k |
|---|---|---|---|---|
| GAN | BigGAN (Brock et al., 2019) | 112M | 1 | 6.95 |
| | GigaGAN (Kang et al., 2023) | 569M | 1 | 3.45 |
| | StyleGAN-XL (Karras et al., 2019) | 166M | 1 | 2.30 |
| AR/Mask | AR w/ VQGAN (Esser et al., 2021) | 227M | 1024 | 26.52 |
| | MaskGIT (Chang et al., 2022) | 227M | 8 | 6.18 |
| | VAR-d30 (Tian et al., 2024) | 2B | 10×2 | 1.92 |
| | MAR-H (Li et al., 2024) | 943M | 256×2 | 1.55 |
| Diff/ Flow | ADM (Karras et al., 2024) | 554M | 250×2 | 10.94 |
| | LDM-4-G (Rombach et al., 2021) | 400M | 250×2 | 3.60 |
| | SimDiff (Hoogeboom et al., 2023) | 2B | 512×2 | 2.77 |
| | DiT-XL/2 (Peebles & Xie, 2022) | 675M | 250×2 | 2.27 |
| | SiT-XL/2 (Ma et al., 2024) | 675M | 250×2 | 2.06 |
| | SiT-XL/2+REPA (Yu et al., 2025) | 675M | 250×2 | 1.42 |
| Shortcut | iCT (Song & Dhariwal, 2023) | 675M | 1 | 34.24 |
| | SCD (Frans et al., 2025) | 675M | 1 | 10.60 |
| | IMM (Zhou et al., 2025) | 675M | 1×2 | 7.77 |
| | MeanFlow (Geng et al., 2025a) | 676M | 1 | 3.43 |
| | | | 2 | 2.93 |
| | **ESC** (w/o-class-consist.) | 676M | 1 | 2.92 |
| | **ESC** (w/-class-consist.) | 676M | 1 | 2.85 |
| | **ESC+** (w/-class-consist.) | 676M | 1 | **2.53** |

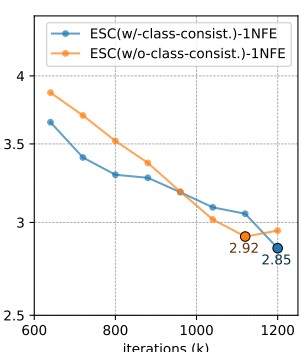

Figure 3: Convergence of FID50k.

Table 4: Uncond. CIFAR-10.

| method | NFE | FID |
|---|---|---|
| iCT | 1 | **2.83** |
| ECT | 1 | 3.60 |
| sCT | 1 | 2.97 |
| IMM | 1 | 3.20 |
| MeanFlow | 1 | 2.92 |
| **ESC** | 1 | **2.83** |

**The time cost of plug-in velocity is minimal.** Computing plug-in velocity involves an $\mathcal{O}(B^2)$ weighted operation within each mini-batch, but with DDP training, per-device batch size is small ($B = 16$ in our experiments). As a result, the extra overhead is negligible because profiling over 1M iterations shows 554 ms/iter vs. 558 ms/iter for conditional vs. plug-in velocity ($\approx 0.7\%$ increase). Despite a small batch size introducing larger estimation variance and bias relative to the ideal velocity, compared to the conditional velocity, it stabilizes training by theoretically reducing variance by $\mathcal{O}(1 - 1/B)$ at almost no additional computational cost and a minor increase in estimation bias.

**Class-consistent mini-batching brings faster convergence.** While the final reported results show comparable performance with and without class-consistent mini-batching, we observe from Fig. 3 that the convergence of FID50k during training is substantially faster with the technique, where Appendix D.7 gives full details. This suggests that the training technique is advantageous in scenarios requiring finetuning with limited training iterations. Exploring its broader applications will be a direction for future work.

## 6 CONCLUSION

We focus on one-step shortcut models trained from scratch and propose a general design framework with theoretical justification of its validity. Building on this, we elucidate the design space of shortcut models through theoretical analysis and empirical evidence, and further propose improvements for continuous-time shortcut model training. Our improved model achieves state-of-the-art performance in image synthesis. More broadly, our work lowers the barrier to innovation in one-step diffusion and enables more systematic exploration of their design, with limitations discussed in Appendix F.

## ACKNOWLEDGEMENTS

This work was supported by the National Science and Technology Major Project of China (No.2021YFA1301603), the Project (No. WU2025B006) from the SOE Dean Special Project

Fund (SOE-DSPF) Program of Westlake University, National Science and Technology Major Project (No. 2022ZD0115101), National Natural Science Foundation of China Project (No. 624B2115, No. U21A20427), Project (No. WU2022A009) from the Center of Synthetic Biology and Integrated Bioengineering of Westlake University, Project (No. WU2023C019) from the Westlake University Industries of the Future Research Funding and Hangzhou Postdoctoral Daily Funding Program (Grant No. 103140026582502, 2025). In addition, we gratefully acknowledge the continuous support from DP Technology, Beijing AI for Science Institute, and Westlake University Center for High-performance Computing, for their sustained provision of computational resources for this project.

## ETHICS STATEMENT

This work investigates one-step diffusion for generative modeling at the methodological level. The datasets used in this study are publicly available benchmark datasets and do not contain sensitive or personally identifiable information (e.g., ImageNet, CIFAR-10).

Potential risks include the possibility of misuse, such as generating misleading or harmful content, or propagating societal biases present in the training data. Our method itself does not explicitly address these issues, but we highlight that appropriate safeguards should be adopted in downstream applications, including content filtering, bias auditing, and domain-specific restrictions.

Overall, we believe the contributions of this work pose minimal ethical risks and can positively impact the community by advancing the efficiency and effectiveness of one-step generative modeling.

## REPRODUCIBILITY STATEMENT

We have made every effort to ensure the reproducibility of our results. All datasets used in this work are publicly available (e.g., ImageNet, CIFAR-10). The preprocessing steps, model architectures, training hyperparameters, and evaluation protocols are described in detail in Sections 3 and 5.

To further facilitate reproducibility, we release our source code through https://github.com/EDAPINENUT/ExplicitShortCut/, and will release trained model checkpoints and experiment scripts upon publication. This will allow researchers to reproduce all reported results and extend our approach in future work.

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

# Appendix

## Table of Contents

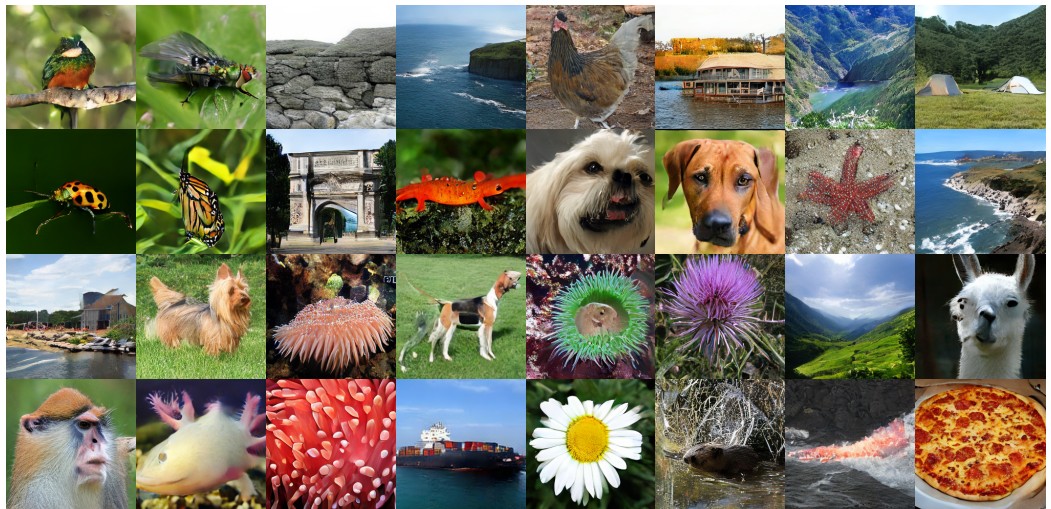

Figure 4: Images generated by ESC with SiT-B/2 trained on ImageNet-256×256, with FID50k 5.77.

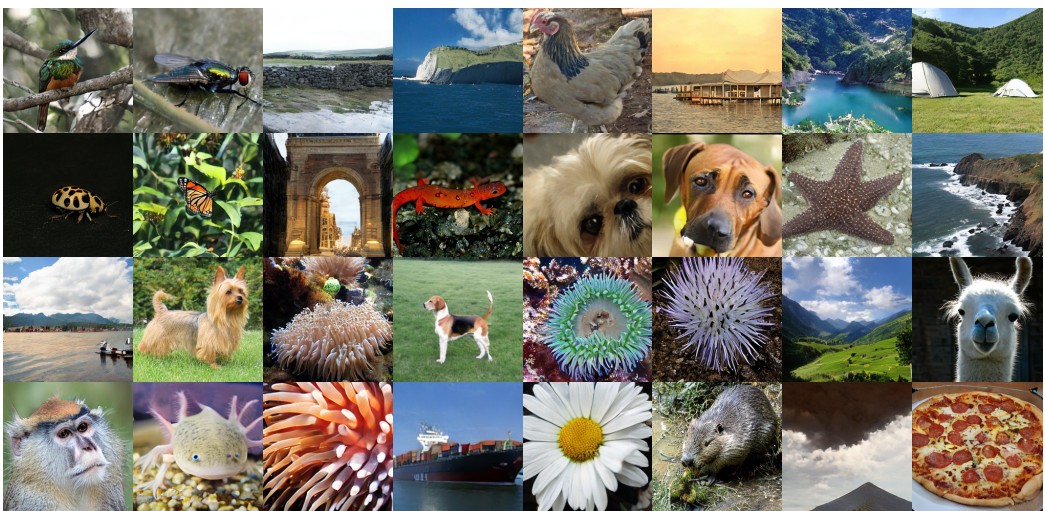

Figure 5: Images generated by ESC with SiT-XL/2 trained on ImageNet-256×256, with FID50k 2.85.

## A  BACKGROUND OF DIFFUSION MODELS

### A.1  STOCHASTIC INTERPOLANTS AND FLOW MAP

Here we give a more formal definition of stochastic interplants and flow map:

**Definition A.1** (Stochastic Interpolants (Albergo et al., 2023)). *The stochastic interpolant $\boldsymbol{I}_t$ between probability densities $q$ and $p_1 = \mathcal{N}(0, \boldsymbol{I})$ is the stochastic process given by*

$$\boldsymbol{x}_t = \alpha_t \boldsymbol{x}_0 + \sigma_t \boldsymbol{z}, \tag{13}$$

*where $\alpha_t, \sigma_t \in C^1([0,1])$ satisfy $\alpha_0 = \sigma_1 = 1$ and $\alpha_1 = \sigma_0 = 0$. We denote the distribution of $\boldsymbol{x}_t$ as $p_t$.*

**Proposition A.2** (Probability Flow). *For all $t \in [0,1]$, the probability density of $\boldsymbol{x}_t$ is the same as the probability density of the solution to*

$$\dot{\boldsymbol{x}}_t = \boldsymbol{v}_t(\boldsymbol{x}_t), \quad \boldsymbol{x}_0 \sim p_0(\boldsymbol{x}), \tag{14}$$

where $\boldsymbol{v} : [0,1] \times \mathbb{R}^d \to \mathbb{R}^d$ is the time-dependent velocity field (or drift) given by

$$\boldsymbol{v}_t(\boldsymbol{x}) = \mathbb{E}_{\boldsymbol{x}_0 \sim p_0, \boldsymbol{z} \sim \mathcal{N}(0, \boldsymbol{I})}[\dot{\boldsymbol{x}}_t \mid \boldsymbol{x}_t = \boldsymbol{x}]. \tag{15}$$

More specifically,

$$\boldsymbol{v}_t(\boldsymbol{x}) = \dot{\alpha}_t \, \mathbb{E}(\boldsymbol{x}_0 \mid \boldsymbol{x}_t = \boldsymbol{x}) + \dot{\sigma}_t \, \mathbb{E}(\boldsymbol{z} \mid \boldsymbol{x}_t = \boldsymbol{x}) \tag{16}$$

**Definition A.3** (Flow Map (Boffi et al., 2025b; Liu, 2025)). *The flow map $\boldsymbol{X}_{s,t} : \mathbb{R}^d \to \mathbb{R}^d$ for Eq. 14 is the unique map such that*

$$X_{s,t}(\boldsymbol{x}_s) = \boldsymbol{x}_t, \quad \text{for all } (s,t) \in [0,1]^2, \tag{17}$$

*where $(\boldsymbol{x}_t)_{t \in [0,1]}$ is any solution to the ODE Eq. 14.*

**Proposition A.4** (Consistency Property (Boffi et al., 2025b)). *The flow map $\boldsymbol{X}_{s,t}(\boldsymbol{x})$ satisfies the Consistency Property*

$$X_{s,r}(X_{t,s}(\boldsymbol{x})) = X_{t,r}(\boldsymbol{x}) \tag{18}$$

*for all $(t,s,r,\boldsymbol{x}) \in [0,1]^3 \times \mathbb{R}^d$. In particular, $X_{s,t}(X_{t,s}(\boldsymbol{x})) = \boldsymbol{x}$ for all $(s,t,\boldsymbol{x}) \in [0,1]^2 \times \mathbb{R}^d$.*

## A.2 FLOW MAP SOLVER

### A.2.1 EULER SOLVER

With a probability velocity field $\boldsymbol{v}_t(\boldsymbol{x})$ which can be derived from a pre-defined probability path or approximated by $\boldsymbol{v}_t^\theta(\boldsymbol{x})$ with a neural network, an Euler Solver can predict the Flow Map from $t$ to $r$ with

$$\boldsymbol{x}_r = X_{t,r}(\boldsymbol{x}_t) = \boldsymbol{x}_t + \int_t^r \boldsymbol{v}_\tau(\boldsymbol{x}) d\tau. \tag{19}$$

Besides, if the integral of $\boldsymbol{v}_\tau(\boldsymbol{x})$ is given as $\boldsymbol{u}_{t,r} = \frac{1}{r-t} \int_t^r \boldsymbol{v}_\tau d\tau$ or parameterized with $\boldsymbol{u}_{t,r}^\theta$, the flow map can be easily obtained with

$$\boldsymbol{x}_r = X_{t,r}(\boldsymbol{x}_t) = \boldsymbol{x}_t + (r-t)\boldsymbol{u}(t,r). \tag{20}$$

### A.2.2 DDIM SOLVER

Let the forward process be defined by

$$\boldsymbol{x}_r = \alpha(r)\boldsymbol{x}_0 + \sigma(r)\boldsymbol{\varepsilon}, \qquad \boldsymbol{\varepsilon} \sim \mathcal{N}(\boldsymbol{0}, \boldsymbol{I}), \tag{21}$$

so that at any $t$ we have

$$\boldsymbol{x}_t = \alpha(t)\boldsymbol{x}_0 + \sigma(t)\boldsymbol{\varepsilon}, \qquad \boldsymbol{v}_t = \dot{\alpha}(t)\boldsymbol{x}_0 + \dot{\sigma}(t)\boldsymbol{\varepsilon}. \tag{22}$$

**Conditional formulation.** Conditioned on $\boldsymbol{x}_t$, the posterior distribution of $(\boldsymbol{x}_0, \boldsymbol{\varepsilon})$ is Gaussian, and hence both $\boldsymbol{x}_r$ and $\boldsymbol{v}_t$ can be written as linear functions of $\boldsymbol{x}_0, \boldsymbol{\varepsilon}$. Taking conditional expectations yields

$$\mathbb{E}\begin{bmatrix} \boldsymbol{x}_t \\ \boldsymbol{v}_t \end{bmatrix} = \begin{bmatrix} \alpha_t & \sigma_t \\ \dot{\alpha}_t & \dot{\sigma}_t \end{bmatrix} \mathbb{E}\begin{bmatrix} \boldsymbol{x}_0 \\ \boldsymbol{\varepsilon} \end{bmatrix}, \tag{23}$$

and by inversion, we obtain

$$\mathbb{E}\begin{bmatrix} \boldsymbol{x}_0 \\ \boldsymbol{\varepsilon} \end{bmatrix} = \frac{1}{\alpha_t \dot{\sigma}_t - \sigma_t \dot{\alpha}_t} \begin{bmatrix} \dot{\sigma}_t & -\sigma_t \\ -\dot{\alpha}_t & \alpha_t \end{bmatrix} \mathbb{E}\begin{bmatrix} \boldsymbol{x}_t \\ \boldsymbol{v}_t \end{bmatrix}. \tag{24}$$

**DDIM update.** Substituting back into the expression for $\boldsymbol{x}_r$ gives

$$\mathbb{E}[\boldsymbol{x}_r \mid \boldsymbol{x}_t] = \alpha_r \, \mathbb{E}[\boldsymbol{x}_0 \mid \boldsymbol{x}_t] + \sigma_r \, \mathbb{E}[\boldsymbol{\varepsilon} \mid \boldsymbol{x}_t] \tag{25}$$

$$= \bar{\alpha}(t,r) \, \boldsymbol{x}_t + \bar{\beta}(t,r) \, \boldsymbol{v}_t, \tag{26}$$

where the coefficients are

$$\bar{\alpha}(t,r) = \frac{\alpha_r \dot{\sigma}_t - \sigma_r \dot{\alpha}_t}{\alpha_t \dot{\sigma}_t - \sigma_t \dot{\alpha}_t}, \qquad \bar{\beta}(t,r) = \frac{-\alpha_r \sigma_t + \sigma_r \alpha_t}{\alpha_t \dot{\sigma}_t - \sigma_t \dot{\alpha}_t}. \tag{27}$$

**Cosine Path.** In the cosine path, the schedule of $\alpha_t = \alpha(t)$ and $\sigma_t = \sigma(t)$ reads

$$\alpha(t) = \cos\left(\frac{\pi}{2}t\right), \quad \sigma(t) = \sin\left(\frac{\pi}{2}t\right)$$

$$\dot{\alpha}(t) = -\frac{\pi}{2}\sin\left(\frac{\pi}{2}t\right), \quad \dot{\sigma}(t) = \frac{\pi}{2}\cos\left(\frac{\pi}{2}t\right)$$

Then:

$$\bar{\alpha}(t,r) = \cos\left(\frac{\pi}{2}(r-t)\right), \quad \bar{\beta}(t,r) = \frac{2}{\pi}\sin\left(\frac{\pi}{2}(r-t)\right) \tag{28}$$

**Linear Schedule.** In the linear path, the schedule of $\alpha_t = \alpha(t)$ and $\sigma_t = \sigma(t)$ reads

$$\alpha(t) = 1 - t, \quad \sigma(t) = t \Rightarrow \dot{\alpha}(t) = -1, \quad \dot{\sigma}(t) = 1$$

Then:

$$\bar{\alpha}(t,r) = \frac{(1-r)(1) - r(-1)}{(1-t)(1) - t(-1)} = 1$$

$$\bar{\beta}(t,r) = \frac{-(1-r)t + r(1-t)}{1} = r - t$$

Therefore,

$$\bar{\alpha}(t,r) = 1, \quad \bar{\beta}(t,r) = r - t$$

### A.3 DERIVED FLOW PATH FROM PRECONDITIONER OF EDM

The original establishment of EDM is based on the score-based diffusion model, while in this part, we aim to demonstrate that although in EDM, $\alpha_t$ and $\sigma_t$ do not satisfy

$$\alpha_0 = 1, \alpha_1 = 0; \sigma_0 = 0, \sigma_1 = 1,$$

the preconditioner of EDM is equivalent to the cosine path in our paper, or namely TrigFlow in sCT, by using the change-of-variable. This part is mostly based on Appendix B of TrigFlow proposed by Lu & Song (2025), while we use a more unified view from stochastic interpolants (Albergo et al., 2023) and SiT (Ma et al., 2024).

#### A.3.1 SCORE-BASED VIEW OF EDM.

**EDM forward diffusion.** We draw $\boldsymbol{x}_0 \sim p_{\text{data}}$ and $\boldsymbol{\varepsilon} \sim \mathcal{N}(\boldsymbol{0}, \sigma_{\text{data}}\boldsymbol{I})$ and define

$$\boldsymbol{x}_t = \alpha_t \boldsymbol{x}_0 + \sigma_t \boldsymbol{\varepsilon}, \tag{29}$$

where $\alpha_t > 0$ and $\sigma_t > 0$ are schedule functions determined by a noise scale $\sigma(t)$:

$$\alpha_t = \frac{\sigma_{\text{data}}}{\sqrt{\sigma_{\text{data}}^2 + \sigma(t)^2}}, \qquad \sigma_t = \frac{\sigma(t)}{\sqrt{\sigma_{\text{data}}^2 + \sigma(t)^2}}. \tag{30}$$

Here $\sigma_{\text{data}}$ denotes the data standard deviation, and the EDM noise schedule is

$$\sigma(t) = \left(\sigma_{\max}^{1/\rho} + t(\sigma_{\min}^{1/\rho} - \sigma_{\max}^{1/\rho})\right)^{\rho}, \qquad t \in [0,1], \tag{31}$$

with typical choices $\sigma_{\min} \approx 2 \times 10^{-3}$, $\sigma_{\max} \approx 80.0$, and $\rho = 7$.

#### A.3.2 FROM EDM PRECONDITIONER TO COSINE PATH.

**Score parameterization.** We may interpret EDM as a score-based model. Specifically, define the score

$$\boldsymbol{s}_t(\boldsymbol{x}_t) := \nabla_{\boldsymbol{x}_t} \log p_t(\boldsymbol{x}_t), \tag{32}$$

and approximate it by a neural network

$$\varphi^{\theta}(\boldsymbol{x}_t, t) \approx \boldsymbol{s}_t(\boldsymbol{x}_t). \tag{33}$$

Since the Gaussian corruption satisfies $\boldsymbol{\varepsilon} = -\sigma(t)\boldsymbol{s}_t(\boldsymbol{x}_t)$, the EDM predictor is written as

$$D^\theta(\boldsymbol{x}_t, t) = c_{\text{skip}}(t)\,\boldsymbol{x}_t + c_{\text{out}}(t)\,\varphi^\theta\big(c_{\text{in}}(t)\boldsymbol{x}_t,\, c_{\text{noise}}(t)\big), \tag{34}$$

since $\frac{\sigma_{\text{data}}^2}{\sigma^2(t)+\sigma_{\text{data}}^2}(\boldsymbol{x}_0 + \sigma(t)\boldsymbol{\varepsilon}) = c_{\text{skip}}(t)\boldsymbol{x}_t$, and according to Eq. 29 and Eq. 30, with scaling coefficients ensuring unit-normalized training targets:

$$c_{\text{in}}(t) = \frac{1}{\sigma_{\text{data}}}, \qquad c_{\text{skip}}(t) = \alpha_t, \qquad c_{\text{out}}(t) = -\sigma_{\text{data}}\,\sigma_t. \tag{35}$$

and the denoiser reduces to

$$D^\theta(\boldsymbol{x}_t, t) = \alpha_t\,\hat{\boldsymbol{x}}_t - \beta_t\,\sigma_{\text{data}}\,\varphi^\theta\big(\hat{\boldsymbol{x}}_t/\sigma_{\text{data}},\, c_{\text{noise}}(t)\big). \tag{36}$$

**Cosine reparameterization.** Since $\alpha_t^2 + \beta_t^2 = 1$, we can introduce a cosine time variable $t' \in [0, 1]$ such that

$$\alpha_t = \cos\left(\frac{\pi}{2}t'\right), \qquad \beta_t = \sin\left(\frac{\pi}{2}t'\right), \qquad t' = \frac{2}{\pi}\arctan\left(\frac{\beta_t}{\alpha_t}\right) = \frac{2}{\pi}\arctan\left(\frac{\sigma(t)}{\sigma_{\text{data}}}\right). \tag{37}$$

On this cosine path, we may equivalently define

$$\alpha(t') = \cos\left(\frac{\pi}{2}t'\right), \qquad \sigma(t') = \sin\left(\frac{\pi}{2}t'\right), \tag{38}$$

which again satisfies $\alpha(t')^2 + \sigma(t')^2 = 1$. Moreover, since in EDM one typically samples $t \sim \log\mathcal{N}(P_{\text{mean}}, P_{\text{std}}^2)$, under the change of variables $t' = \frac{2}{\pi}\arctan(t)$, the resulting sampling matches the time parameterization used in CT and sCT (Table 1).

### A.3.3 FROM SCORE PARAMETERIZATION TO VELOCITY

In general, we can denote $t' \in [0, 1]$ as $t$, with Eq. 38, which leads the denoising predictor of Eq. 36 to

$$D^\theta(\boldsymbol{x}_t, t) = \cos(\frac{\pi}{2}t)\,\boldsymbol{x}_t - \sin(\frac{\pi}{2}t)\,\sigma_{\text{data}}\,\varphi^\theta\big(\boldsymbol{x}_t/\sigma_{\text{data}},\, c'_{\text{noise}}(t)\big), \tag{39}$$

Since $D^\theta(\boldsymbol{x}_t, t)$ aims to approximate $\boldsymbol{x}_0$, and if we write

$$\boldsymbol{v}^\theta(\boldsymbol{x}_t, t) = \frac{\pi}{2}\,\sigma_{\text{data}}\,\varphi^\theta\big(\boldsymbol{x}_t/\sigma_{\text{data}},\, c'_{\text{noise}}(t)\big).$$

, it reads

$$D^\theta(\boldsymbol{x}_t, t) = \cos(\frac{\pi}{2}t)\boldsymbol{x}_t + \frac{2}{\pi}\sin(-\frac{\pi}{2}t)\boldsymbol{v}^\theta(\boldsymbol{x}_t, t)$$

the coefficient $\cos(\frac{\pi}{2}t)$ and $\frac{2}{\pi}\sin(-\frac{\pi}{2}t)$ coincides to $\bar{\alpha}(t, 0)$ and $\bar{\beta}(t, 0)$ in Eq. 28, the parameterization of $\boldsymbol{v}_t^\theta(\boldsymbol{x}_t) = \frac{\pi}{2}\sigma_{\text{data}}F^\theta(\boldsymbol{x}_t, t)$ is equivalent to denoise the path of preconditioner in EDM schedule. The same evidence is also provided with Eq. 4 in Lu & Song (2025).

## B DERIVATION OF FLOW MAP CONSTRUCTION AND LOSS

### B.1 CONSISTENCY TRAINING

In CTs and CMs (Song et al., 2023), the original paper uses the EDM preconditioner as the components of the basic diffusion model. By using a neural network $\varphi^\theta$ to approximate the score function $\boldsymbol{s}_t(\boldsymbol{x})$, its target is to map any noised samples in $t$ to 0 which in the flow map notation, reads

$$\hat{X}_{t,0}^\theta(\boldsymbol{x}_t) = f^\theta(\boldsymbol{x}_t) = c_{\text{skip}}(t)\boldsymbol{x}_t + c_{\text{out}}\sigma_{\text{data}}\varphi^\theta(\boldsymbol{x}_t),$$

such that

$$l_{\text{cm}} = d(f^\theta(\boldsymbol{x}_t), \text{sg}(f^\theta(\hat{\boldsymbol{x}}_s))), \tag{40}$$

where

$$t = [\sigma_{\max}^{1/\rho} + \frac{\tau}{K}(\sigma_{\min}^{1/\rho} - \sigma_{\max}^{1/\rho})]^\rho$$

$$s = [\sigma_{\max}^{1/\rho} + \frac{\tau+1}{K}(\sigma_{\min}^{1/\rho} - \sigma_{\max}^{1/\rho})]^\rho \tag{41}$$

$$\tau \sim \mathcal{U}[0, 1, \ldots, K],$$

and $\hat{x}_s$ is on the same conditional flow path that generates $x_t$. By adopting the equivalence of EDM preconditioner to Cosine path, as shown in Appendix A.3, we can write $F^\theta$ as the approximator of $v_t$ under the change-of-variable, which reads

$$
\begin{aligned}
x_0 &\sim p_0, \quad \varepsilon \sim \mathcal{N}(0,1) \\
x_t &= \cos(\frac{\pi}{2}t)x_0 + \sin(\frac{\pi}{2}t)\varepsilon \\
v_{t|0} &= -\frac{\pi}{2}\sin(\frac{\pi}{2}t)x_0 + \frac{\pi}{2}\cos(\frac{\pi}{2}t)\varepsilon \\
\hat{x}_s &= \cos(\frac{\pi}{2}s)x_0 + \sin(\frac{\pi}{2}s)\varepsilon,
\end{aligned}
\tag{42}
$$

Then, following the derivation of Eq. 28, by substituting the coefficients $\bar{\alpha}_{t,s}$ and $\bar{\beta}_{t,s}$, we can equivalently express

$$
\begin{aligned}
\hat{X}_{t,s}(x_t) &= \hat{x}_s = \mathrm{DDIM}(x_t, v_{t|0}, t, s) \\
\hat{X}_{s,r}(\hat{x}_s) &= \hat{x}_r = \mathrm{DDIM}(\hat{x}_s, F^\theta(\hat{x}_s), s, r) \\
X^\theta_{t,r}(x_t) &= x^\theta_r = \mathrm{DDIM}(x_t, F^\theta(x_t), t, r)
\end{aligned}
\tag{43}
$$

Finally, by replacing $d$ in Eq. 40, it reads

$$
\begin{aligned}
l_{\mathrm{ct}}(x_t, r, s, t; \theta) &= \mathrm{LPIPS}(f^\theta(x_t), \mathrm{sg}(f^\theta(\hat{x}_s))), \\
&= \mathrm{LPIPS}(X^\theta_{t,r}(x_t), \mathrm{sg}(\hat{X}_{s,r}(\hat{x}_s))) \\
&= \mathrm{LPIPS}\Big(\mathrm{DDIM}(x_t, v^\theta_t(x_t), t, r), \ \mathrm{sg}\big(\mathrm{DDIM}(\hat{x}_s, v^\theta_s(\hat{x}_s), s, r)\big)\Big),
\end{aligned}
\tag{44}
$$

where $\hat{x}_s = \mathrm{DDIM}(x_t, v_{t|0}, t, s)$, which coincides the description of CTs in Sec. 2.3. Further, under the change-of-variable, the time sampler Eq. 41 will be of the form as described by Table 1.

## B.2 Shortcut Diffusion

In the original paper of SCD (Frans et al., 2025), it parameterizes the velocity with the neural network as

$$
F^\theta(x_t, t, r) = v^\theta(x_t, t, r),
$$

while we claim that the $v^\theta(x_t, t, r)$ is not the instantaneous one, because it requires the entries of both $t$ as the start point, and $r$ as the end point. Instead, we regard it as the average velocity, leading to our parameterization of

$$
F^\theta(x_t, t, r) = u^\theta_{t,r}(x_t).
$$

In this way, $x_t$ is first sampled from a linear path, as

$$
\begin{aligned}
x_0 &\sim p_0, \quad \varepsilon \sim \mathcal{N}(0,1) \\
x_t &= (1-t)x_0 + t\varepsilon \\
v_{t|0} &= -x_0 + \varepsilon,
\end{aligned}
\tag{45}
$$

Then, the flow map is constructed via

$$
\begin{aligned}
\hat{X}_{t,s}(x_t) &= \hat{x}_s = x_t - hu^\theta_{t,s}(x_t) \\
\hat{X}_{s,r}(x_s) &= \hat{x}_r = x_s - hu^\theta_{s,r}(\hat{x}_s) \\
X^\theta_{t,r}(x_t) &= \hat{x}^\theta_t = x_t - 2hu^\theta_{t,r}(x_t)
\end{aligned}
\tag{46}
$$

Finally, according to the consistency property of flow map shown in Prop. A.4, and by setting $w = h^2$ the loss term for the regularization of velocity can be rewritten as

$$
\begin{aligned}
l_{\mathrm{scd}}(x_t, r, s, t; \theta) &= \frac{1}{4h^2} \cdot \big\| x_t - 2hu^\theta_{t,r}(x_t) - \mathrm{sg}\big(x_t - hu^\theta_{t,s}(x_t) - hu^\theta_{s,r}(x_t - hu^\theta_{t,s}(x_t))\big) \big\|^2_2 \\
&= \frac{1}{4h^2} \cdot \big\| x_t - 2hu^\theta_{t,r}(x_t) - \mathrm{sg}\big(x_t - hu^\theta_{t,s}(x_t) - hu^\theta_{s,r}(\hat{x}_s)\big) \big\|^2_2 \\
&= \Big\| u^\theta_{t,r}(x_t) - \frac{1}{2}\mathrm{sg}\big(u^\theta_{t,s}(x_t) + u^\theta_{s,r}(\hat{x}_s)\big) \Big\|^2_2,
\end{aligned}
\tag{47}
$$

where $r, s, t$ are equi-spaced, such that $t - s = s - r = h$, which coincides Eq. 7 in Sec. 2.3. Specifically, in the time sampler, it defines the total step $K$ with $T = \lfloor \log_2 K \rfloor$. For each sample, it draws $2h \in \{2^{-0}, 2^{-1}, \dots, 2^{-(T-1)}\}$ and $e \sim \mathcal{U}(0, 1)$, then computes

$$t = \tfrac{1}{K} \left\lfloor 2hK + e \cdot (K - 2hK + 1) \right\rfloor, \qquad r = t - 2h, \qquad s = t - h, \tag{48}$$

as the time sampler for $\{r, s, t\}$. We denote the time sampler as $(t, h) \sim \text{Uniform} \log_2(t, h)$, and $s = t - h, r = t - 2h$.

## B.3 INDUCTIVE MOMENT MATCHING

Given a mini-batch of size $B$, IMM first draws $\{(\boldsymbol{x}_0^{(i)}, \boldsymbol{\varepsilon}^{(i)})\}_{i=1}^B$, and partition them into groups of size $M$. Within each group, a triplet $(r, s, t)$ is sampled, where $(r, t)$ are drawn uniformly from $[0, 1]$ with $s$ separated by a fixed difference, as

$$
\begin{aligned}
t &\sim \mathcal{U}[0, 1] \\
n_s &= \frac{1}{1 - t} - \frac{1}{2^{\gamma}} \\
s &= \frac{n_s}{n_s + 1} \\
r &\sim \mathcal{U}[0, t],
\end{aligned}
\tag{49}
$$

where $\gamma$ is usually set as 12 according to its code implementation. Each group thus defines $M$ correlated particle samples that share the same flow interpolation times. For each group of $M$ particles, IMM constructs an $M \times M$ kernel matrix based on a positive definite kernel function as metrics $d(\cdot, \cdot)$ (e.g., RBF). The objective is

$$\mathcal{L}_{\text{imm}}(\theta) = \mathbb{E}_{\boldsymbol{x}_t, \boldsymbol{x}_t', \boldsymbol{x}_r, \boldsymbol{x}_r', r, s, t} \left[ w(r, t) \left( \ker(\boldsymbol{z}_{t,r}, \boldsymbol{z}_{t,r}') + \ker(\boldsymbol{z}_{s,r}, \boldsymbol{z}_{s,r}') - \ker(\boldsymbol{z}_{t,r}, \boldsymbol{z}_{s,r}') - \ker(\boldsymbol{z}_{t,r}', \boldsymbol{z}_{s,r}) \right) \right], \tag{50}$$

where

$$
\begin{aligned}
\boldsymbol{z}_{t,r} &= \text{DDIM}(\boldsymbol{x}_t, \boldsymbol{v}_t^{\theta}(\boldsymbol{x}_t), t, r), \\
\boldsymbol{z}_{t,r}' &= \text{sg}(\text{DDIM}(\boldsymbol{x}_t, \boldsymbol{v}_t^{\theta}(\boldsymbol{x}_t), t, r)),
\end{aligned}
$$

and $w(r, t)$ is a prior weighting. Intuitively, samples are repelled by intra-group pairs (e.g. $\boldsymbol{z}_{t,r}$ vs. $\boldsymbol{z}_{t,r}'$) while attracted towards inter-group matches ($\boldsymbol{z}_{t,r}$ vs. $\boldsymbol{z}_{s,r}'$). This ensures both intra-sample diversity and inter-sample alignment.

In practice, a batch of size $B$ is divided into $B/M$ groups, and the IMM loss is computed as an average over these groups. For kernels, RBF and negative pseudo-Huber kernels are common choices for $\ker(\cdot, \cdot)$, which guarantee moment matching up to all orders.

Further, we bridge the IMM loss with the common flow map learning objective in the following. In Eq. 50, it gives the group kernel function, according to Appendix. C.3 in Zhou et al. (2025), we can write the loss here as

$\mathcal{L}_{\text{imm}}(\theta)$

$$= \mathbb{E}_{\boldsymbol{x}_t, \boldsymbol{x}_t', \boldsymbol{x}_r, \boldsymbol{x}_r', r, s, t} \left[ w(r, t) \left( \ker(\boldsymbol{z}_{t,r}, \boldsymbol{z}_{t,r}') + \ker(\boldsymbol{z}_{s,r}, \boldsymbol{z}_{s,r}') - \ker(\boldsymbol{z}_{t,r}, \boldsymbol{z}_{s,r}') - \ker(\boldsymbol{z}_{t,r}', \boldsymbol{z}_{s,r}) \right) \right],$$

$$= \mathbb{E}_{r, s, t} \left[ w(r, t) (\mathbb{E}_{\boldsymbol{x}_t, \boldsymbol{x}_t', \boldsymbol{x}_r, \boldsymbol{x}_r'} \left[ \langle \ker(\boldsymbol{z}_{t,r}, \cdot), \ker(\boldsymbol{z}_{t,r}', \cdot) \rangle + \langle \ker(\boldsymbol{z}_{s,r}, \cdot), \ker(\boldsymbol{z}_{s,r}', \cdot) \rangle \right. \right.$$

$$\left. \left. - \langle \ker(\boldsymbol{z}_{t,r}, \cdot), \ker(\boldsymbol{z}_{s,r}', \cdot) \rangle - \langle \ker(\boldsymbol{z}_{t,r}', \cdot), \ker(\boldsymbol{z}_{s,r}, \cdot) \rangle \right] \right) \right]$$

$$= \mathbb{E}_{r, s, t} \left[ w(r, t) \left\langle \mathbb{E}_{\boldsymbol{x}_t} \left[ \ker(X_{t,r}^{\theta}(\boldsymbol{x}_t), \cdot) - \ker(\text{sg}(X_{s,r}^{\theta}(\hat{\boldsymbol{x}}_s)), \cdot) \right], \right. \right.$$

$$\left. \left. \mathbb{E}_{\boldsymbol{x}_t'} \left[ \ker(X_{t,r}^{\theta}(\boldsymbol{x}_t'), \cdot) - \ker(\text{sg}(X_{s,r}^{\theta}(\hat{\boldsymbol{x}}_s')), \cdot) \right] \right\rangle \right]$$

$$= \mathbb{E}_{r, s, t} \left[ w(r, t) \left\| \mathbb{E}_{\boldsymbol{x}_t} \left[ \ker(X_{t,r}^{\theta}(\boldsymbol{x}_t), \cdot) - \ker(\text{sg}(X_{s,r}^{\theta}(\hat{\boldsymbol{x}}_s)), \cdot) \right] \right\|_{\mathcal{H}}^2 \right]$$

$$\tag{51}$$

where $\hat{\boldsymbol{x}}_s = \text{DDIM}(\boldsymbol{x}_t, \boldsymbol{v}_{t|0}, t, s)$ is estimated with conditional velocity, and $\left\|\mathbb{E}_{\boldsymbol{x}}\left[\ker(X_{t,r}^{\theta}(\boldsymbol{x}_t), \cdot) - \ker(\text{sg}(X_{s,r}^{\theta}(\hat{\boldsymbol{x}}_s)), \cdot)\right]\right\|_{\mathcal{H}}^2$ is the Maximum Mean Discrepancy commonly defined on Reproducing Kernel Hilbert Space (RKHS) $\mathcal{H}$ with a positive definite kernel in IMM. Then, according to Jensen's inequality,

$$\mathbb{E}_{r,s,t}\left[w(r,t)\left\|\mathbb{E}_{\boldsymbol{x}_t}\left[\ker(X_{t,r}^{\theta}(\boldsymbol{x}_t), \cdot) - \ker(\text{sg}(X_{s,r}^{\theta}(\hat{\boldsymbol{x}}_s)), \cdot)\right]\right\|_{\mathcal{H}}^2\right]$$

$$\leq \mathbb{E}_{r,s,t,\boldsymbol{x}_t}\left[w(r,t)\left\|\ker(X_{t,r}^{\theta}(\boldsymbol{x}_t), \cdot) - \ker(\text{sg}(X_{s,r}^{\theta}(\hat{\boldsymbol{x}}_s)), \cdot)\right\|_{\mathcal{H}}^2\right] \tag{52}$$

In this way, we define $d(\boldsymbol{x}, \boldsymbol{y})$ as RKHS discrepancy $\left\|\ker(\boldsymbol{x}, \cdot) - \ker(\boldsymbol{y}, \cdot)\right\|_{\mathcal{H}}^2$, it reads

$$\mathcal{L}_{\text{imm}}(\theta) \leq \mathbb{E}_{r,s,t\sim p(\tau),\, \boldsymbol{x}_t \sim p_t}\left[w(r,t) \cdot d(X_{t,r}^{\theta}(\boldsymbol{x}_t), \text{sg}(\hat{X}_{s,r} \circ \hat{X}_{t,s}(\boldsymbol{x}_t)))\right] \tag{53}$$

Therefore, minimizing Eq. 5 is equivalent to upper-bounding the IMM loss.

## B.4 MEANFLOW

As MeanFlow takes the Linear flow path, we can easily obtain

$$\text{DDIM}(\boldsymbol{x}_t, \boldsymbol{v}_t, t, s) = \boldsymbol{x}_t + (s - t)\boldsymbol{v}_t, \tag{54}$$

which means the DDIM solver and Euler solver are the same. With the flow map construction, we can write the corresponding terms into $\|\cdot\|^2$ of $d(\cdot, \cdot)$ as follows:

$$\left(\boldsymbol{x}_t + (r - t)\boldsymbol{u}_{t,r}^{\theta}(\boldsymbol{x}_t)\right) - \left(\boldsymbol{x}_t + (s - t)\boldsymbol{v}_t + (r - s)\boldsymbol{u}_{s,r}^{\theta}(\boldsymbol{x}_s)\right)$$

$$= (r - t)\boldsymbol{u}_{t,r}^{\theta}(\boldsymbol{x}_t) - (s - t)\boldsymbol{v}_t - (r - s)\boldsymbol{u}_{s,r}^{\theta}(\boldsymbol{x}_s). \tag{55}$$

By substituting $s = t - dt$ and normalized by $dt$, we get

$$((r - t)\boldsymbol{u}_{t,r}^{\theta}(\boldsymbol{x}_t) + dt \cdot \boldsymbol{v}_t - (r - t + dt)\boldsymbol{u}_{t-dt,r}^{\theta}(\boldsymbol{x}_{t-dt}))/dt$$

$$= dt \cdot \left(\boldsymbol{v}_t + \frac{d\left[(r - t)\boldsymbol{u}_{t,r}^{\theta}(\boldsymbol{x}_t)\right]}{dt}\right)/dt \tag{56}$$

$$= \boldsymbol{v}_t - \boldsymbol{u}_{t,r}^{\theta}(\boldsymbol{x}_t) - (t - r)\frac{d}{dt}\boldsymbol{u}_{t,r}^{\theta}(\boldsymbol{x}_t).$$

However, the marginal velocity $\boldsymbol{v}_t$ is inaccessible in training, so it can be replaced by the conditional velocity $\boldsymbol{v}_{t|0}$. From Eq. 6 in Geng et al. (2025a), by adding the adaptive loss term $w$, this loss coincides with the training objective of MeanFlow in Eq. 8, as

$$l(\boldsymbol{x}_t, r, t - dt, t; \theta) = w \cdot \left\|\boldsymbol{u}_{t,r}^{\theta}(\boldsymbol{x}_t) - \text{sg}\left(\boldsymbol{v}_{t|0} + (r - t)\frac{d\boldsymbol{u}_{t,r}^{\theta}(\boldsymbol{x}_t)}{dt}\right)\right\|^2 \tag{57}$$

Further, MeanFlow adopts an adaptively weighted squared L2 loss. Given the regression error $\Delta = u_{\theta} - u_{\text{tgt}}$, where $u_{\text{tgt}} = \text{sg}\left(\boldsymbol{v}_{t|0} + (r - t)\frac{d\boldsymbol{u}_{t,r}^{\theta}(\boldsymbol{x}_t)}{dt}\right)$, the squared L2 loss is $\|\Delta\|_2^2$. To stabilize training, MeanFlow reweights $\|\Delta\|_2^2$ with

$$w = \frac{1}{(\|\Delta\|_2^2 + c)^p}, \tag{58}$$

where $c > 0$ avoids division by zero and $p$ controls the weighting ($p = 0.5$ recovers a Pseudo-Huber style loss). The final loss is defined as $\text{sg}(w) \cdot \mathcal{L}$, where $\text{sg}(\cdot)$ denotes the stop-gradient operator.

## B.5 s-CONSISTENCY TRAINING

We here simplify the derivation with $\sigma_{\text{data}} = 1$. According to the Eq. 44, we can write the corresponding terms with squared $l_2$-distance with $s = t - \Delta t$ and $l(\boldsymbol{x}_t, r, s, t; \theta)$ normalized by $\Delta t$, as the

following:

$$w(t) \lim_{\substack{s=t-\Delta t \\ \Delta t \to 0}} \frac{1}{\Delta t} \|\mathrm{DDIM}(\boldsymbol{x}_t, \boldsymbol{v}_t^\theta, t, r) - \mathrm{sg}(\mathrm{DDIM}\left(\mathrm{DDIM}(\boldsymbol{x}_t, \boldsymbol{v}_t, t, s), \boldsymbol{v}_s^\theta, s, r)\right)\|^2$$

$$=w(t) \lim_{\substack{s=t-\Delta t \\ \Delta t \to 0}} \frac{1}{\Delta t} \|\mathrm{DDIM}(\boldsymbol{x}_t, \boldsymbol{v}_t^\theta, t, r) - \mathrm{sg}(\mathrm{DDIM}(\hat{\boldsymbol{x}}_s, \boldsymbol{v}_s^\theta, s, r))\|^2$$

$$=w(t) \lim_{\substack{s=t-\Delta t \\ \Delta t \to 0}} \left(\mathrm{DDIM}\left(\boldsymbol{x}_t, \boldsymbol{v}_t^\theta, t, r\right) - \mathrm{sg}(\mathrm{DDIM}(\hat{\boldsymbol{x}}_s, \boldsymbol{v}_s^\theta, s, r))\right)^\mathsf{T} \cdot$$
$$\frac{\mathrm{DDIM}\left(\boldsymbol{x}_t, \boldsymbol{v}_t^\theta, t, r\right) - \mathrm{sg}(\mathrm{DDIM}(\hat{\boldsymbol{x}}_s, \boldsymbol{v}_s^\theta, s, r))}{\Delta t} \tag{59}$$

$$\approx w(t)(\mathrm{DDIM}\left(\boldsymbol{x}_t, \boldsymbol{v}_t^\theta, t, r\right) - \mathrm{sg}(\mathrm{DDIM}(\hat{\boldsymbol{x}}_s, \boldsymbol{v}_s^\theta, s, r)))^\mathsf{T}\mathrm{sg}\left(\frac{d\mathrm{DDIM}\left(\boldsymbol{x}_t, \boldsymbol{v}_t^\theta, t, r\right)}{dt}\right)$$

By fixing $r = 0$, from Eq. 28, it can be obtain that the gradient of Eq. 59 w.r.t. $\theta$ is

$$w(t)\nabla_\theta \left[(\mathrm{DDIM}\left(\boldsymbol{x}_t, \boldsymbol{v}_t^\theta, t, r\right) - \mathrm{sg}(\mathrm{DDIM}(\hat{\boldsymbol{x}}_s, \boldsymbol{v}_s^\theta, s, r)))^\mathsf{T}\mathrm{sg}\left(\frac{d\mathrm{DDIM}\left(\boldsymbol{x}_t, \boldsymbol{v}_t^\theta, t, r\right)}{dt}\right)\right]$$

$$=w(t)\nabla_\theta \left[\mathrm{DDIM}\left(\boldsymbol{x}_t, \boldsymbol{v}_t^\theta, t, r\right)^\mathsf{T}\mathrm{sg}\left(\frac{d\mathrm{DDIM}\left(\boldsymbol{x}_t, \boldsymbol{v}_t^\theta, t, r\right)}{dt}\right)\right]$$

$$=w(t)\nabla_\theta \left[\left(\cos(\frac{\pi}{2}t)\boldsymbol{x}_t - \sin(\frac{\pi}{2}t)\boldsymbol{v}_t^\theta\right)^\mathsf{T}\mathrm{sg}\left(\frac{d\mathrm{DDIM}\left(\boldsymbol{x}_t, \boldsymbol{v}_t^\theta, t, r\right)}{dt}\right)\right]$$

$$=\nabla_\theta \left(\boldsymbol{v}_t^\theta\right)^\mathsf{T}\mathrm{sg}\left(-\sin(\frac{\pi}{2}t)w(t)\frac{d\mathrm{DDIM}\left(\boldsymbol{x}_t, \boldsymbol{v}_t^\theta, t, r\right)}{dt}\right)$$

$$=\nabla_\theta \|\boldsymbol{v}_t^\theta - \mathrm{sg}\left(\boldsymbol{v}_t^\theta + w'(t)\frac{d\mathrm{DDIM}\left(\boldsymbol{x}_t, \boldsymbol{v}_t^\theta, t, r\right)}{dt}\right)\|^2$$

$$\tag{60}$$

where $w(t) = \frac{1}{\tan(\frac{\pi}{2}t)}$, and $w'(t) = -\sin(\frac{\pi}{2}t)w(t) = -\cos\left(\frac{\pi}{2}t\right)$, we prove that this flow map construction corresponds to the original loss with the specific time sampler, and the derived loss is in the same form as Eq. 9 where we rewrite $w'$ by $w$.

## C   PROOF OF THEOREMS AND PROPOSITIONS

### C.1   PROOF OF EQUIVARIANCE OF MEANFLOW AND SCT-LINEAR (REMARK. 2.1)

*Sketch of proof.* First note that under linear paths, $\hat{X}_{t,0}^\theta(\boldsymbol{x}_t) = \mathrm{DDIM}(\boldsymbol{x}_t, \boldsymbol{v}_t^\theta(\boldsymbol{x}_t), t, 0) = \boldsymbol{x}_t - t\boldsymbol{v}_t^\theta(\boldsymbol{x}_t)$. As for the training objective, with $w(t) = 1$ and linear path, Eq. 9 can be easily written as $l(\boldsymbol{x}_t, r, t - dt, t; \theta)\big|_{r=0} = \|\boldsymbol{v}_t^\theta(\boldsymbol{x}_t) - \mathrm{sg}(\boldsymbol{v}_t + (r-t)\frac{d}{dt}\boldsymbol{v}_t^\theta(\boldsymbol{x}_t))\|_2^2\big|_{r=0}$. Since in sCT, $r$ is fixed to 0, parameterization of the neural network can be invariant to $r$, leading to $\boldsymbol{v}_t^\theta(\boldsymbol{x}_t) = F^\theta(\boldsymbol{x}_t, t, 0)$. Thus, in sampling, by replacing $\boldsymbol{u}_{t,0}$ and $\boldsymbol{v}_t$ with $F^\theta$ in Eq. 2 and 3, respectively, the sampling processes are the same when following linear paths.

### C.2   PROOF OF ERROR BOUND (THEOREM 2.2)

In this section, we aim to prove the error bound of DTSC&CTSC. Specifically, the theorem is stated as follows.

**Theorem C.1** (Error bound of DTSC&CTSC). *Assume the marginal velocity of the flow path satisfies the one-sided Lipschitz condition, where*

$$\exists\, C_t \in L^1[0,1] \,:\, \big(\boldsymbol{v}_t(\boldsymbol{x}) - \boldsymbol{v}_t(\boldsymbol{y})\big)\cdot(\boldsymbol{x}-\boldsymbol{y}) \geq -C_t\|\boldsymbol{x}-\boldsymbol{y}\|^2, \quad \text{for all } (t,\boldsymbol{x},\boldsymbol{y}) \in [0,1]\times\mathbb{R}^d\times\mathbb{R}^d.$$

*Assume $X_{t,s}^\theta$ are twice continuously differentiable with bounded second derivatives, the weighting function $w(r,s,t)$ is non-negative and bounded. For DTSC, also assume $p(r=0) > 0$, 1st-step satisfies $\hat{X}_{t,t}(\boldsymbol{x}_t) = \boldsymbol{x}_t$, and $\exists t_1 \leq \cdots \leq t_N$ s.t $p(0,t_n,t_{n+1}) > 0, w(0,t_n,t_{n+1}) > 0$.*
*Under $d(\boldsymbol{x},\boldsymbol{y}) = \|\boldsymbol{x}-\boldsymbol{y}\|_2^2$, given $\boldsymbol{x}_1 \sim p_1$, let $p_0$ the density of $\boldsymbol{x}_0$, and $p_0^\theta$ the density of $\boldsymbol{x}_0^\theta = X_{1,0}^\theta(\boldsymbol{x}_1)$ that is estimated by neural network with parameter $\theta$, then*

$$W_2^2(p_0, p_0^\theta) \leq C_1^1 \mathcal{L}_{dtsc}(\theta) + C_2^1(t-s),$$
$$W_2^2(p_0, p_0^\theta) \leq C_1^2 \mathcal{L}_{ctsc}(\theta),$$

*where we write the training objective in Eq. 5 as $\mathcal{L}_\bullet(\theta) = \mathbb{E}_{r,s,t\sim p(\tau),\, \boldsymbol{x}_t\sim p_t}[l_\bullet(\boldsymbol{x}_t,r,s,t;\theta)]$, with $\bullet \in \{ctsc, dtsc\}$, and $W_2(\cdot,\cdot)$ is the Wasserstein-2 distance.*

We note that MeanFlow loss and sCT loss are all $\mathcal{L}_{\text{ctsc}}(\theta)$, and CT loss and SCD loss are all $\mathcal{L}_{\text{dtsc}}(\theta)$. IMM's loss is calculated across different conditional paths, as finally bounded by the $\mathcal{L}_{\text{dtsc}}(\theta)$ as shown in Appendix B.3. The mentioned previous methods all satisfy the assumptions about $w(r,s,t)$, $p(r,s,t)$, and $\hat{X}_{t,t}(\boldsymbol{x}_t) = \boldsymbol{x}_t$. As for the assumption of $d(\boldsymbol{x},\boldsymbol{y}) = \|\boldsymbol{x}-\boldsymbol{y}\|_2^2$, it holds for all the mentioned methods except CT, which takes LPIPS as the metric function. The convergence of $\mathcal{L}_{\text{ct}}$ has already been proved by Song et al. (2023).

We prove the theorem in three steps: (i) establish the error bound for DTSC; (ii) derive the start point differential CTSC bound; and (iii) further derive the end point differential CTSC bound.

### C.2.1 ERROR BOUND OF DTSC

**Lemma C.2.** *Assume $d$ and $X_{t,s}^\theta$ are both twice continuously differentiable with bounded second derivatives, the weighting function $w(r,s,t)$ is non-negative and bounded, and 1st-step satisfies $\hat{X}_{t,t}(\boldsymbol{x}_t) = \boldsymbol{x}_t$. We define a loss $\mathcal{L}_1$ as follows:*

$$\mathcal{L}_1(\theta) := \mathbb{E}\left[ w(r,s,t)\, d\big( X_{t,r}^\theta(\boldsymbol{x}_t), \text{sg}(X_{s,r}^\theta(\boldsymbol{x}_s)) \big) \right]. \tag{61}$$

*Then,*

$$\mathcal{L}_{dtsc}(\theta) = \mathcal{L}_1(\theta) + \mathcal{O}(t-s),$$

*where $\mathcal{L}_{dtsc}$ is the discrete-time shortcut models' loss.*

*Proof.* As

$$\boldsymbol{x}_t = \boldsymbol{x}_s + (t-s)\boldsymbol{v}_s + \mathcal{O}(t-s),$$

and here we define the $\theta^-$ as the parameters in the model which stop-grad operates, for notational simplicity. By using Taylor expansion, we can get that

$$\begin{aligned}
\mathcal{L}_1(\theta) &= \mathbb{E}\left[ w(r,s,t)\, d\big(X_{t,r}^\theta(\boldsymbol{x}_t), X_{s,r}^{\theta^-}(\boldsymbol{x}_s)\big) \right] \\
&= \mathbb{E}\Big[ w(r,s,t)\, d\Big( X_{s,r}^\theta(\boldsymbol{x}_s) + \partial_s X_{s,r}^\theta(\boldsymbol{x}_s)(t-s) + \partial_x X_{s,r}^\theta(\boldsymbol{x}_s)(t-s)\boldsymbol{v}_s + o(t-s) \\
&\qquad\qquad , X_{s,r}^{\theta^-}(\boldsymbol{x}_s) \Big) \Big] \\
&= \mathbb{E}\left[ w(r,s,t)\, d\Big( X_{s,r}^\theta(\boldsymbol{x}_s) + \partial_s X_{s,r}^\theta(\boldsymbol{x}_s)(t-s) + \mathcal{O}(t-s), X_{s,r}^{\theta^-}(\boldsymbol{x}_s)\Big) \right] \\
&= \mathbb{E}\left[ w(r,s,t) \Big( d\big(X_{s,r}^\theta(\boldsymbol{x}_s), X_{s,r}^{\theta^-}(\boldsymbol{x}_s)\big) + \partial_1 d\big(X_{s,r}^\theta(\boldsymbol{x}_s), X_{s,r}^{\theta^-}(\boldsymbol{x}_s)\big)\partial_s X_{s,r}^{\theta^-}(\boldsymbol{x}_s)(t-s)\Big) \right] \\
&\quad + \mathcal{O}(t-s).
\end{aligned}$$

As for $\hat{X}_{t,s}(\boldsymbol{x}_t)$, there are three ways to calculate it as stated in Sec. 2. Ways in Eq. 1 and Eq. 3 are numerical solvers, so we have $\hat{X}_{t,s}(\boldsymbol{x}_s) = \boldsymbol{x}_s + \mathcal{O}(t-s)$. Eq. 2 also satisfies this equation since we

have the assumption that $\hat{X}_{t,t}(\boldsymbol{x}_t) = \boldsymbol{x}_t$ and $X_{t,s}$ is twice continuously differentiable with bounded second derivative.

With a similar derivation, we also have

$$
\begin{aligned}
\mathcal{L}_{\mathrm{dtsc}}(\theta) &= \mathbb{E}\left[w(r,s,t)\,d\big(X_{s,r}^{\theta^-}(\hat{X}_{t,s}(\boldsymbol{x}_t)),\, X_{t,r}^\theta(\boldsymbol{x}_t)\big)\right] \\
&= \mathbb{E}\left[w(r,s,t)\,d\Big(X_{s,r}^{\theta^-}(\boldsymbol{x}_s), X_{s,r}^\theta(\boldsymbol{x}_s) + \partial_s X_{s,r}^\theta(\boldsymbol{x}_s)(t-s) + \mathcal{O}(t-s)\Big)\right] \\
&= \mathbb{E}\left[w(r,s,t)\left(d\big(X_{s,r}^{\theta^-}(\boldsymbol{x}_s),\, X_{s,r}^\theta(\boldsymbol{x}_s)\big) + \partial_1 d\big(X_{s,r}^{\theta^-}(\boldsymbol{x}_s),\, X_{s,r}^\theta(\boldsymbol{x}_s)\big)\partial_s X_{s,r}^{\theta^-}(\boldsymbol{x}_s)(t-s)\right)\right] \\
&\quad + \mathcal{O}(t-s).
\end{aligned}
$$

By subtracting the two equations, we obtain

$$
\mathcal{L}_{\mathrm{dtsc}}(\theta) = \mathcal{L}_1(\theta) + \mathcal{O}(t-s).
$$

$\square$

**Theorem C.3.** *Assume $X_{t,s}^\theta$ is twice continuously differentiable with bounded second derivatives, the weighting function $w(r,s,t)$ is non-negative and bounded. Also, assume $p(r=0) > 0$, 1st-step satisfies $\hat{X}_{t,t}(\boldsymbol{x}_t) = \boldsymbol{x}_t$, and $\exists t_1 \leq \cdots \leq t_N$ s.t $p(0, t_n, t_{n+1}) > 0, w(0, t_n, t_{n+1}) > 0$. Under $d(\boldsymbol{x}, \boldsymbol{y}) = \|\boldsymbol{x} - \boldsymbol{y}\|_2^2$,*

$$
W_2^2(p_1, p_1^\theta) \leq C_1 \mathcal{L}_{dtsc}(\theta) + C_2(t-s), \tag{62}
$$

*where $C_1 = \sum_{n=1}^{N-1} \frac{1}{p(0,t_n,t_{n+1})w(0,t_n,t_{n+1})}$, and $W_2(\cdot, \cdot)$ is the Wasserstein-2 distance.*

*Proof.* Since we have proved that the difference between $\mathcal{L}_{\mathrm{dtsc}}$ and $\mathcal{L}_1$ is $\mathcal{O}(s-t)$, we only need to prove

$$
W_2^2(p_1, p_1^\theta) \leq C\mathcal{L}_1(\theta).
$$

Because for $r = 0, s_0, t_0$ s.t $p(0, s_0, t_0) > 0, w(0, s_0, t_0) > 0$,

$$
\begin{aligned}
\mathcal{L}_1(\theta) &= \mathbb{E}\big[w(r,s,t)d\big(X_{s,r}^{\theta^-}(\boldsymbol{x}_s), X_{t,r}^\theta(\boldsymbol{x}_t)\big)\big] \\
&\geq p(0,s_0,t_0)w(0,s_0,t_0)d\big(X_{s_0,0}^{\theta^-}(\boldsymbol{x}_{s_0}), X_{t_0,0}^\theta(\boldsymbol{x}_{t_0})\big),
\end{aligned}
$$

we have

$$
\mathbb{E}\left[d\big(X_{s_0,0}^{\theta^-}(\boldsymbol{x}_{s_0}), X_{t_0,0}^\theta(\boldsymbol{x}_{t_0})\big)\right] \leq \frac{1}{p(0,s_0,t_0)w(0,s_0,t_0)}\mathcal{L}_1(\theta). \tag{63}
$$

We define

$$
e_{t,0} := X_{t,0}(\boldsymbol{x}_t) - X_{t,0}^\theta(\boldsymbol{x}_t).
$$

Then,

$$
\begin{aligned}
e_{t_n,0} &= X_{t_n,0}(\boldsymbol{x}_{t_n}) - X_{t_n,0}^\theta(\boldsymbol{x}_{t_n}) \\
&= X_{t_{n+1},0}(\boldsymbol{x}_{t_{n+1}}) - X_{t_{n+1},0}^\theta(\boldsymbol{x}_{t_{n+1}}) + X_{t_{n+1},0}^\theta(\boldsymbol{x}_{t_{n+1}}) - X_{t_n,0}^\theta(\boldsymbol{x}_{t_n}) \\
&= e_{t_{n+1},0} + \left(X_{t_{n+1},0}^\theta(\boldsymbol{x}_{t_{n+1}}) - X_{t_n,0}^\theta(\boldsymbol{x}_{t_n})\right)
\end{aligned}
$$

Consequently,

$$
e_{1,0} = e_{0,0} + \sum_{n=1}^{N-1}\left(X_{t_{n+1},0}^\theta(\boldsymbol{x}_{t_{n+1}}) - X_{t_n,0}^\theta(\boldsymbol{x}_{t_n})\right),
$$

where $e_{0,0} = 0$. Using Eq. 63, we can get

$$W_2^2(p_0, p_0^\theta) \le \mathbb{E}\|e_{n,0}\|_2^2$$

$$\le \sum_{n=1}^{N-1} \mathbb{E}\|X_{t_{n+1},0}^\theta(\boldsymbol{x}_{t_{n+1}}) - X_{t_n,0}^\theta(\boldsymbol{x}_{t_n})\|_2^2$$

$$= \sum_{n=1}^{N-1} \mathbb{E}\left[d\left(X_{t_{n+1},0}^\theta(\boldsymbol{x}_{t_{n+1}}), X_{t_n,0}^\theta(\boldsymbol{x}_{t_n})\right)\right]$$

$$\le \sum_{n=1}^{N-1} \frac{\mathcal{L}_1(\theta)}{p(0, t_n, t_{n+1})w(0, t_n, t_{n+1})}$$

$$= \left(\sum_{n=1}^{N-1} \frac{1}{p(0, t_n, t_{n+1})w(0, t_n, t_{n+1})}\right)\mathcal{L}_1(\theta).$$

With Eq. 61, we finally have the Theorem $\qquad\square$

### C.2.2 ERROR BOUND OF START POINT DIFFERENTIAL CTSC

Next, we prove the error bound of the start point differential CTSC. The derivation is adopted from Boffi et al. (2025b).

**Theorem C.4.** *When $s \to t$, the CTSC loss can be written as*

$$\mathcal{L}_{\text{ctsc-s-to-t}} = \mathbb{E}\|\partial_t X_{t,r}^\theta(\boldsymbol{x}_t) + \boldsymbol{v}_t \nabla X_{t,r}^\theta(\boldsymbol{x}_t)\|_2^2$$

*Under $d(\boldsymbol{x}, \boldsymbol{y}) = \|\boldsymbol{x} - \boldsymbol{y}\|_2^2$, then*

$$W_2^2(p_0, p_0^\theta) \le C_3 \mathcal{L}_{\text{ctsc-s-to-t}}(\theta),$$

*where $C_3 = e$, and $W_2(\cdot, \cdot)$ is the Wasserstein-2 distance.*

*Proof.* Firstly, from the chain rule,

$$\frac{d}{dt}X_{t,r}^\theta(\boldsymbol{x}_t) = \partial_t X_{t,r}^\theta(\boldsymbol{x}_t) + \boldsymbol{v}_t \cdot \nabla X_{t,r}^\theta(\boldsymbol{x}_t),$$

we can simply use $\boldsymbol{u}_{t,r}^\theta$ as the model output, and write the term into the expectation of $\mathcal{L}_{\text{ctsc-s-to-t}}$ as

$$\|\partial_t X_{t,r}^\theta(\boldsymbol{x}_t) + \boldsymbol{v}_t \cdot \nabla X_{t,r}^\theta(\boldsymbol{x}_t)\|_2^2$$

$$=\|\frac{d}{dt}X_{t,r}^\theta(\boldsymbol{x}_t)\|_2^2$$

$$=\|\boldsymbol{v}_t + \frac{d}{dt}(r-t)\boldsymbol{u}_{t,r}^\theta(\boldsymbol{x}_t)\|_2^2$$

$$=\|\boldsymbol{u}_{t,r}(\boldsymbol{x}_t) - \boldsymbol{v}_t - (r-t)\frac{d\boldsymbol{u}_{t,r}(\boldsymbol{x}_t)}{dt}\|_2^2$$

which coincides to the MeanFlow loss $l_{\text{mf}}$ in Eq. 8. While in Remark. 2.1, sCT loss is equivalent to MeanFlow loss in linear paths, the CTSC loss when $s \to r$ is also of the same form as claimed. The cosine path version of sCT loss is a variant, so we did not include it in this stage, and will consider it as our future work. Then, we first define that

$$E_{t,r} := \mathbb{E}\|X_{t,r}(\boldsymbol{x}_t) - X_{t,r}^\theta(\boldsymbol{x}_t)\|_2^2.$$

After differentiation, we can get

$$-\frac{dE_{t,r}}{dt} = -\mathbb{E}\left[2\left(X_{t,r}(\boldsymbol{x}_t) - X_{t,r}^\theta(\boldsymbol{x}_t)\right)\left(\frac{dX_{t,r}(\boldsymbol{x}_t)}{dt} - \frac{dX_{t,r}^\theta(\boldsymbol{x}_t)}{dt}\right)\right]$$

$$= \mathbb{E}\left[2\left(X_{t,r}(\boldsymbol{x}_t) - X_{t,r}^\theta(\boldsymbol{x}_t)\right)\left(\frac{dX_{t,r}^\theta(\boldsymbol{x}_t)}{dt}\right)\right]$$

$$= \mathbb{E}\left[2\left(X_{t,r}(\boldsymbol{x}_t) - X_{t,r}^\theta(\boldsymbol{x}_t)\right)\left(\partial_t X_{t,r}^\theta(\boldsymbol{x}_t) + \boldsymbol{v}_t \cdot \nabla X_{t,r}^\theta(\boldsymbol{x}_t)\right)\right]$$

$$\le E_{t,r} + \mathbb{E}\|\partial_t X_{t,r}^\theta(\boldsymbol{x}_t) + \boldsymbol{v}_t \cdot \nabla X_{t,r}^\theta(\boldsymbol{x}_t)\|_2^2,$$

So,

$$-e^t \partial_t E_{t,r} - e^t E_{t,r} \le e^t \mathbb{E}\|\partial_t X_{t,r}^\theta(\boldsymbol{x}_t) + \boldsymbol{v}_t \cdot \nabla X_{t,r}^\theta(\boldsymbol{x}_t)\|_2^2$$

$$-\partial_t e^t E_{t,r} \le e^t \mathbb{E}\|\partial_t X_{t,r}^\theta(\boldsymbol{x}_t) + \boldsymbol{v}_t \cdot \nabla X_{t,r}^\theta(\boldsymbol{x}_t)\|_2^2$$

With $E_{r,r} = 0$, we have

$$E_{t,r} \le \int_r^t e^{\tau - t} \mathbb{E}\|\partial_t X_{t,r}^\theta(\boldsymbol{x}_t) + \boldsymbol{v}_t \cdot \nabla X_{t,r}^\theta(\boldsymbol{x}_t)\|_2^2 d\tau$$

$$\le e^1 \int_r^t \mathbb{E}\|\partial_t X_{t,r}^\theta(\boldsymbol{x}_t) + \boldsymbol{v}_t \cdot \nabla X_{t,r}^\theta(\boldsymbol{x}_t)\|_2^2 d\tau$$

$$\le e\mathcal{L}_{\text{ctsc-s-to-r}}.$$

By setting $C_3 = e$, the theorem is proved. $\qquad\square$

### C.2.3 Error Bound of End Point Differential CTSC

Finally, we provide the proof of the error bound of the endpoint differential CTSC.

**Theorem C.5.** *Assume the marginal velocity of the flow path satisfies the one-sided Lipschitz condition, where*

$$\exists C_t \in L^1[0,1] \; : \; \big(\boldsymbol{v}_t(\boldsymbol{x}) - \boldsymbol{v}_t(\boldsymbol{y})\big) \cdot (\boldsymbol{x} - \boldsymbol{y}) \ge -C_t\|\boldsymbol{x} - \boldsymbol{y}\|^2, \quad \text{for all } (t, \boldsymbol{x}, \boldsymbol{y}) \in [0,1] \times \mathbb{R}^d \times \mathbb{R}^d.$$

*When $s \to r$, the CTSC loss can be written as*

$$\mathcal{L}_{\text{ctsc-s-to-r}}(\theta) = \mathbb{E}\|\boldsymbol{v}(X_{t,\tau}^\theta(\boldsymbol{x}_t)) - \partial_\tau X_{t,\tau}^\theta(\boldsymbol{x}_t)\|_2^2$$

*Under $d(\boldsymbol{x}, \boldsymbol{y}) = \|\boldsymbol{x} - \boldsymbol{y}\|_2^2$, then*

$$W_2^2(p_0, p_0^\theta) \le C_3 \mathcal{L}_{\text{ctsc-s-to-r}}(\theta),$$

*where $C_3 = e^{1 + 2\int_0^1 |C_t| dt}$, and $W_2(\cdot, \cdot)$ is the Wasserstein-2 distance.*

*Proof.* Using the one-sided Lipschitz condition, we can get

$$-(X_{t,r}(\boldsymbol{x}) - X_{t,r}(\boldsymbol{y}))(\boldsymbol{v}_r(X_{t,r}(\boldsymbol{x})) - \boldsymbol{v}_r(X_{t,r}(\boldsymbol{y}))) \le 2C_t\|X_{t,r}(\boldsymbol{x}) - X_{t,r}(\boldsymbol{y})\|_2^2.$$

We then define

$$E_{t,r} := \mathbb{E}_{\boldsymbol{x}}\|X_{t,r}(\boldsymbol{x}_t) - X_{t,r}^\theta(\boldsymbol{x}_t)\|_2^2.$$

With differentiation, we have

$$-\frac{dE_{t,r}}{dr} = -2\mathbb{E}_{\boldsymbol{x}}\left[\left(X_{t,r}(\boldsymbol{x}_t) - X_{t,r}^\theta(\boldsymbol{x}_t)\right)\left(\frac{dX_{t,r}(\boldsymbol{x}_t)}{dr} - \frac{dX_{t,r}^\theta(\boldsymbol{x}_t)}{dr}\right)\right]$$

$$= -2\mathbb{E}_{\boldsymbol{x}}\left[\left(X_{t,r}(\boldsymbol{x}_t) - X_{t,r}^\theta(\boldsymbol{x}_t)\right)\left(\boldsymbol{v}(X_{t,r}(\boldsymbol{x}_t)) - \partial_r X_{t,r}^\theta(\boldsymbol{x}_t)\right)\right]$$

$$= -2\mathbb{E}_{\boldsymbol{x}}\left[\left(X_{t,r}(\boldsymbol{x}_t) - X_{t,r}^\theta(\boldsymbol{x}_t)\right)\left(\boldsymbol{v}(X_{t,r}^\theta(\boldsymbol{x}_t)) - \partial_r X_{t,r}^\theta(\boldsymbol{x}_t)\right)\right]$$

$$\quad - 2\mathbb{E}_{\boldsymbol{x}}\left[\left(X_{t,r}(\boldsymbol{x}_t) - X_{t,r}^\theta(\boldsymbol{x}_t)\right)\left(\boldsymbol{v}(X_{t,r}(\boldsymbol{x}_t)) - \boldsymbol{v}(X_{t,r}^\theta(\boldsymbol{x}_t))\right)\right]$$

$$\le \mathbb{E}_{\boldsymbol{x}}\|X_{t,r}(\boldsymbol{x}_t) - X_{t,r}^\theta(\boldsymbol{x}_t)\|_2^2 + \mathbb{E}_{\boldsymbol{x}}\|\boldsymbol{v}(X_{t,r}^\theta(\boldsymbol{x}_t)) - \partial_r X_{t,r}^\theta(\boldsymbol{x}_t)\|_2^2$$

$$\quad - 2\mathbb{E}_{\boldsymbol{x}}\left[\left(X_{t,r}(\boldsymbol{x}_t) - X_{t,r}^\theta(\boldsymbol{x}_t)\right)\left(\boldsymbol{v}(X_{t,r}(\boldsymbol{x}_t)) - \boldsymbol{v}(X_{t,r}^\theta(\boldsymbol{x}_t))\right)\right]$$

$$\le E_{t,r} + \mathbb{E}_{\boldsymbol{x}}\|\boldsymbol{v}(X_{t,r}^\theta(\boldsymbol{x}_t)) - \partial_r X_{t,r}^\theta(\boldsymbol{x}_t)\|_2^2 - 2C_t E_{t,r}$$

$$= (1 - 2C_t)E_{t,r} + \mathbb{E}_{\boldsymbol{x}}\|\boldsymbol{v}(X_{t,r}^\theta(\boldsymbol{x}_t)) - \partial_r X_{t,r}^\theta(\boldsymbol{x}_t)\|_2^2.$$

So,

$$\partial_r(-e^{r - 2\int_t^r C_\tau d\tau} E_{t,r}) \le e^{r - 2\int_t^r C_\tau d\tau} \mathbb{E}_{\boldsymbol{x}}\|\boldsymbol{v}(X_{t,r}^\theta(\boldsymbol{x}_t)) - \partial_r X_{t,r}^\theta(\boldsymbol{x}_t)\|_2^2.$$

With $E_{t,t} = 0$, we have

$$E_{t,r} \le \int_r^t e^{-r+\tau+2\int_\tau^r C_\gamma d\gamma} \mathbb{E}_{\boldsymbol{x}} \|\boldsymbol{v}(X_{t,\tau}^\theta(\boldsymbol{x}_t)) - \partial_\tau X_{t,\tau}^\theta(\boldsymbol{x}_t)\|_2^2 d\tau$$

$$\le e^{-r+1+2\int_r^t |C_\tau| d\tau} \int_r^t \mathbb{E}_{\boldsymbol{x}} \|\boldsymbol{v}(X_{t,\tau}^\theta(\boldsymbol{x}_t)) - \partial_\tau X_{t,\tau}^\theta(\boldsymbol{x}_t)\|_2^2 d\tau.$$

Therefore, when $t = 1, r = 0$, we have

$$E_{1,0} \le e^{1+2\int_0^1 |C_t| dt} \int_0^1 \mathbb{E}_{\boldsymbol{x}} \|\boldsymbol{v}(X_{t,\tau}^\theta(\boldsymbol{x}_t)) - \partial_\tau X_{t,\tau}^\theta(\boldsymbol{x}_t)\|_2^2 d\tau.$$

Finally, due to

$$W_2^2(p_0, p_0^\theta) \le \mathbb{E}\|X_{1,0}(x_1) - X_{1,0}^\theta(x_1)\|_2^2$$

and

$$\mathcal{L}_{\text{ctsc-s-to-r}}(\theta) = \mathbb{E}\|\boldsymbol{v}(X_{t,\tau}^\theta(\boldsymbol{x}_t)) - \partial_\tau X_{t,\tau}^\theta(\boldsymbol{x}_t)\|_2^2$$

$$= \int_{[0,1]^2} w(t,t,r) \mathbb{E}_{\boldsymbol{x}} \|\boldsymbol{v}(X_{t,\tau}^\theta(\boldsymbol{x}_t)) - \partial_\tau X_{t,\tau}^\theta(\boldsymbol{x}_t)\|_2^2 dt dr,$$

we obtain

$$W_2^2(p_0, p_0^\theta) \le \mathbb{E}\|X_{1,0}(\boldsymbol{x}_1) - X_{1,0}^\theta(\boldsymbol{x}_1)\|_2^2$$

$$\le e^{1+2\int_0^1 |C_t| dt} \mathcal{L}_{\text{ctsc-s-to-r}}(\theta).$$

By setting $C_3 = e^{1+2\int_0^1 |C_t| dt}$, the theorem is proved. $\qquad\square$

## C.3 Optimal Path of Shortcut Model (Q.1. in Sec. 3)

Previous works have claimed that the cosine path is optimal for diffusion models from the perspective of Fisher information metric (Santos & Lin, 2023). Here, we provide the analysis of the optimal path for one-step models under the Fisher information metric.

We first briefly introduce the Fisher information metric. Treating probability distributions $p(\gamma)$ as a smooth manifold, the Fisher information metric defines a Riemannian geometry that enables the computation of distances between them. Specifically, the definition is

$$I(\gamma)_{ij} = \mathbb{E}_{X \sim p_\gamma} \left[ \frac{\partial}{\partial \gamma_i} \log p_\gamma(X) \frac{\partial}{\partial \gamma_j} \log p_\gamma(X) \right].$$

When the distribution family is exponential as

$$p(\boldsymbol{x}|\gamma) = h(\boldsymbol{x}) \exp(\eta(\gamma)^T T(\boldsymbol{x}) - \psi(\gamma)),$$

the Fisher information metric becomes (Karczewski et al., 2025)

$$\mathcal{I}_\gamma = \left( \frac{\partial \eta(\gamma)}{\partial \gamma} \right)^\top \left( \frac{\partial \mu(\gamma)}{\partial \gamma} \right),$$

where

$$\mu(\gamma) = \mathbb{E}[T(x) \mid \gamma] = \int T(x) p(x \mid \gamma) dx$$

is the expectation parameter. Then we can naturally define the optimal schedule as the geodesic between two distributions, which leads to the theorem below.

**Theorem C.6.** *(Zhang, 2025) The optimal schedule under the metric $I(\gamma) \in \mathbb{R}$ is generated by $\varphi^*$ of the form*

$$\varphi^*(\gamma) = \Lambda^{-1}(\Lambda\gamma), \quad \text{where} \quad \Lambda(s) = \int_0^s \sqrt{I(r)} dr.$$

With these preparations, we now turn to the one-step diffusion. We point out that, for the one-step diffusion, since our goal becomes modeling the average velocity, we no longer consider the manifold of $p(\boldsymbol{x}_t)$, but rather that of $p(\boldsymbol{u}_{0,t}(\boldsymbol{x}_0)) = p(\boldsymbol{x}_t - \boldsymbol{x}_0)$. Then, we claim that the linear path is the optimal conditional schedule as follows.

**Theorem C.7.** *For $\forall \boldsymbol{x}_0$, the linear schedule is the optimal schedule considering $\{p(\boldsymbol{u}_{0,t} \mid \boldsymbol{x}_0, t)\}$, i.e,*

$$\gamma = \Lambda^{-1}(\Lambda\gamma), \quad where \quad \gamma = (\boldsymbol{x}_0, t).$$

*Proof.*

$$p(\boldsymbol{u}_{0,t} \mid \boldsymbol{x}_0, t) = p(\frac{\boldsymbol{x}_t - \boldsymbol{x}_0}{t} \mid \boldsymbol{x}_0, t)$$

$$= p(\frac{\alpha_t \boldsymbol{x}_0 - \boldsymbol{x}_0 + \sigma_t \epsilon}{t} \mid \boldsymbol{x}_0, t)$$

$$= \mathcal{N}(\boldsymbol{u}_{0,t}; \frac{\alpha_t \boldsymbol{x}_0 - \boldsymbol{x}_0}{t}, \frac{\sigma_t^2}{t^2}I)$$

$$= \frac{t}{(2\pi)^{d/2}\sigma_t^d} \exp\Big(-\frac{t^2}{2\sigma_t^2}\big(\|\boldsymbol{u}_{0,t}\|^2 - \frac{2\alpha_t - 2}{t}\boldsymbol{u}_{0,t}^T\boldsymbol{x}_0 + (\frac{\alpha_t - 1}{t})^2\|\boldsymbol{x}_0\|^2\big)\Big)$$

So $p(\boldsymbol{u}_{0,t} \mid \boldsymbol{x}_0, t)$ is exponential, and we have

$$\eta(\boldsymbol{x}_0, t) = -\frac{t^2}{2\sigma_t^2}\big(-\frac{2\alpha_t - 2}{t}\boldsymbol{x}_0, 1\big)$$

$$= \big(\frac{t(\alpha_t - 1)}{\sigma_t^2}\boldsymbol{x}_0, -\frac{t^2}{2\sigma_t^2}\big),$$

and

$$T(\boldsymbol{u}_{0,t}) = (\boldsymbol{u}_{0,t}, \|\boldsymbol{u}_{0,t}\|^2).$$

Then

$$\mu(\boldsymbol{x}_0, t) = \mathbb{E}[T(\boldsymbol{u}_{0,t}) \mid \boldsymbol{x}_0, t]$$

$$= \int T(\boldsymbol{u}_{0,t})p(\boldsymbol{u}_{0,t} \mid \boldsymbol{x}_0, t)d\boldsymbol{u}_{0,t}$$

$$= \int (\boldsymbol{u}_{0,t}, \|\boldsymbol{u}_{0,t}\|^2)\mathcal{N}(\boldsymbol{u}_{0,t}; \frac{\alpha_t \boldsymbol{x}_0 - \boldsymbol{x}_0}{t}, \frac{\sigma_t^2}{t^2}I)d\boldsymbol{u}_{0,t}$$

$$= \big(\frac{\alpha_t \boldsymbol{x}_0 - \boldsymbol{x}_0}{t}, \big(\frac{\alpha_t \boldsymbol{x}_0 - \boldsymbol{x}_0}{t}\big)^2 + \frac{\sigma_t^2}{t^2}\big).$$

So

$$\frac{\partial \eta(\boldsymbol{x}_0, t)}{\partial(\boldsymbol{x}_0, t)} = \begin{pmatrix} \dfrac{t(\alpha_t - 1)}{\sigma_t^2} & \dfrac{(\alpha_t - 1 + t\alpha_t)\sigma_t^2 - 2t(\alpha_t - 1)\dot{\sigma}_t\sigma_t}{\sigma_t^4}\boldsymbol{x}_0 \\ 0 & -\dfrac{4t\sigma_t^2 - 4\dot{\sigma}_t\sigma_t t^2}{4\sigma_t^4} \end{pmatrix},$$

$$\frac{\partial \mu(\boldsymbol{x}_0, t)}{\partial(\boldsymbol{x}_0, t)} = \begin{pmatrix} \dfrac{\alpha_t - 1}{t} & \dfrac{\dot{\alpha}_t t - \alpha_t + 1}{t^2}\boldsymbol{x}_0 \\ \big(\dfrac{\alpha_t - 1}{t}\big)^2 \cdot 2\boldsymbol{x}_0 & \dfrac{\dot{\alpha}_t t - \alpha_t + 1}{t^2} \cdot 2\dfrac{\alpha_t - 1}{t}\boldsymbol{x}_0 + \dfrac{2\dot{\alpha}_t\alpha_t t^2 - 2t\sigma_t^2}{t^4} \end{pmatrix}$$

$$= \begin{pmatrix} \dfrac{\alpha_t - 1}{t} & \dfrac{\dot{\alpha}_t t - \alpha_t + 1}{t^2}\boldsymbol{x}_0 \\ 2\big(\dfrac{\alpha_t - 1}{t}\big)^2\boldsymbol{x}_0 & \dfrac{2(\dot{\alpha}_t t - \alpha_t + 1)(\alpha_t - 1)}{t^3}\boldsymbol{x}_0 + \dfrac{2\dot{\alpha}_t\alpha_t t^2 - 2t\sigma_t^2}{t^4} \end{pmatrix}.$$

Substituting the linear schedule $\alpha_t = 1 - t$ and $\sigma_t = t$, we obtain

$$\frac{\partial \eta(\boldsymbol{x}_0, t)}{\partial(\boldsymbol{x}_0, t)} = \begin{pmatrix} \dfrac{t(-t)}{t^2} & \dfrac{-2t \cdot t^2 - 2t(1-t)t + 2t^2}{t^4} \boldsymbol{x}_0 \\ 0 & \dfrac{t^3 - t^3}{t^4} \end{pmatrix}$$

$$= \begin{pmatrix} -1 & \dfrac{-2t^3 - 2t^2 + 2t^3 + 2t^2}{t^4} \boldsymbol{x}_0 \\ 0 & 0 \end{pmatrix}$$

$$= \begin{pmatrix} -1 & 0 \\ 0 & 0 \end{pmatrix}$$

and

$$\frac{\partial \mu(\boldsymbol{x}_0, t)}{\partial(\boldsymbol{x}_0, t)} = \begin{pmatrix} \dfrac{1 - t - 1}{t} & \dfrac{-t - 1 + t + 1}{t^2} \boldsymbol{x}_0 \\ 2\boldsymbol{x}_0 & \dfrac{-t - 1 + t + 1}{t^2} \cdot 2 \dfrac{1 - t - 1}{t} \boldsymbol{x}_0 + \dfrac{2t^3 - 2t^3}{t^4} \end{pmatrix}$$

$$= \begin{pmatrix} -1 & 0 \\ 2\boldsymbol{x}_0 & 0 \end{pmatrix}.$$

Based on these two equations, we get

$$I(\boldsymbol{x}_0, t) = \begin{pmatrix} -1 & 0 \\ 0 & 0 \end{pmatrix}^T \begin{pmatrix} -1 & 0 \\ 2\boldsymbol{x}_0 & 0 \end{pmatrix}$$

$$= \begin{pmatrix} 1 & 0 \\ 0 & 0 \end{pmatrix},$$

which means the metric under the linear path is uniform. So the probability $p(\boldsymbol{u}_{0,t}|\boldsymbol{x}_0, t)$ travels at a constant rate, leading to the optimum. □

## C.4 PROOF OF INFERENCE ERROR ANALYSIS (PROP. 3.1)

We first give a detailed version of Prop. 3.1 as following,

**Proposition C.8** (Inference error analysis). *Under mild regularity conditions shown in Appendix C.4.1, the Wasserstein-2 distance of the shortcut model with one-step generation is bounded as:*

$$W_2^2(p_0, p_0^\theta) \le 2 \left( \text{BV}_{ctsc} + 8\text{Var}\left[\frac{d}{dt}\boldsymbol{u}_{t,r}^\theta(\boldsymbol{x}_t)\right] + 8\sigma_{\boldsymbol{v}_{t|0}}^2 \right)\bigg|_{r=0, t=1}, \tag{64}$$

$$W_2^2(p_0, p_0^\theta) \le 2 \left( \text{BV}_{dtsc} + 8\delta_2^2 \, \text{Var}\left[\boldsymbol{u}_{s,r}^\theta(\boldsymbol{x}_t)\right] + 8(1 + \ell^2\delta_2^2)\delta_1^2 \, \sigma_{dtsc}^2 \right)\bigg|_{r=0, t=1}, \tag{65}$$

*where* $\text{BV}_\bullet = \text{Bias}_{\bullet\text{-}tgt}^2 + \text{Bias}_{\bullet\text{-}loss}^2 + 2\text{Var}[\boldsymbol{u}^\theta(\boldsymbol{x}_1, t, r)]$ *with* $\bullet \in \{\text{ctsc, dtsc}\}$, *and* $\text{Bias}_{\bullet\text{-}tgt}^2$ *and* $\text{Bias}_{\bullet\text{-}loss}^2$ *are defined as*

$$\text{Bias}_{\bullet\text{-}tgt}^2 = \mathbb{E}[\|\boldsymbol{u}_{t,r}(\boldsymbol{x}_t) - Y_\bullet\|_2^2]$$
$$\text{Bias}_{\bullet\text{-}loss}^2 = \mathbb{E}[l_\bullet(\boldsymbol{x}_t, t, r; \theta)] \tag{66}$$

$Y_\bullet$ *is the two flow map target when* $Y_{ctsc}$ *in each model, i.e.*

$$Y_{ctsc} = \frac{d}{dt}\boldsymbol{u}^\theta(\boldsymbol{x}_t, t, r) - \boldsymbol{v}_{t|0}(\boldsymbol{x}_t|\boldsymbol{x}_0),$$
$$Y_{dtsc} = \delta_1 \, \boldsymbol{v}_{t|0}(\boldsymbol{x}_t|\boldsymbol{x}_0) + \delta_2 \, \boldsymbol{u}_{s,r}^\theta(\boldsymbol{x}_t + \delta_1 \, \boldsymbol{v}_{t|0}(\boldsymbol{x}_t|\boldsymbol{x}_0)) \quad \text{if use CT loss},$$
$$Y_{dtsc} = \delta_1 \, \boldsymbol{v}_{t|0}(\boldsymbol{x}_t|\boldsymbol{x}_0) + \delta_2 \, \boldsymbol{u}^\theta(\boldsymbol{x}_t + \delta_1 \, \boldsymbol{u}_{s,r}^\theta(\boldsymbol{x}_t, t, s)) \quad \text{if use SCD loss}.$$

$l_\bullet(\boldsymbol{x}_t, r, s, t; \theta)$ *is the term in the expectation of different training objectives as given in Sec. 2.3;* $\delta_1 = t - s$, $\delta_2 = s - r$; $\ell$ *is local Lipschitz constant of* $\boldsymbol{u}^\theta$; $\sigma_{\boldsymbol{v}_{t|0}}^2$ *is the variance of the conditional velocity, defined by* $\sigma_{\boldsymbol{v}_{t|0}}^2 := \text{Var}(\boldsymbol{v}_{t|0}(\boldsymbol{x}_t|\boldsymbol{x}_0)|)$; $\sigma_{dtsc}^2 = \sigma_{\boldsymbol{v}_{t|0}}^2$ *when using CT's flow map targets, or* $\sigma_{dtsc}^2 = \text{Var}[\boldsymbol{u}_{t,s}^\theta(\boldsymbol{x}_t)]$ *when using SCD's targets.*

### C.4.1 ASSUMPTIONS FOR PROP. 3.1

We state the regularity conditions required for the theorem.

**Assumption C.9** (Velocity variance). *The conditional velocity $\boldsymbol{v}_{t|0}$ approximates the ground-truth velocity $\boldsymbol{v}_t$, such that we can write*

$$\boldsymbol{v}_{t|0}(\boldsymbol{x}_t|\boldsymbol{x}_0) = \boldsymbol{v}_t(\boldsymbol{x}_t) + \boldsymbol{\eta}_t,$$

*where we assume the discrepancy $\boldsymbol{\eta}_t$ has variance $\sigma^2_{\boldsymbol{v}_{t|0}}$. Because $E[\boldsymbol{\eta}_t|\boldsymbol{x}_0] = \boldsymbol{0}$ (take expectation on both sides), so $\mathrm{Var}(\boldsymbol{v}_{t|0}) = \sigma^2_{\boldsymbol{v}_{t|0}}$*

**Assumption C.10** (Lipschitz continuity). *There exist constants $\ell$ such that for any $\boldsymbol{h}$,*

$$\|\boldsymbol{u}^\theta(\boldsymbol{x}_t + \boldsymbol{h}, r, s) - \boldsymbol{u}^\theta(\boldsymbol{x}_t, r, s)\| \leq \ell\|\boldsymbol{h}\|,$$

*with $\ell$ independent of $(\boldsymbol{x}_t, t, r, s)$.*

### C.4.2 LEMMA USED FOR PROOF

**Lemma C.11** (Bias–variance–covariance(BV-CV) decomposition). *For random vectors $A, B \in \mathbb{R}^d$ with finite second moments,*

$$\mathbb{E}\|A - B\|^2 = \|\mathbb{E}A - \mathbb{E}B\|^2 + \mathrm{tr}\,\mathrm{Cov}[A] + \mathrm{tr}\,\mathrm{Cov}[B] - 2\,\mathrm{tr}\,\mathrm{Cov}[A, B]. \tag{67}$$

*Proof.* Denote $A$ and $B$'s expectations by $\mu_A = \mathbb{E}A$ and $\mu_B = \mathbb{E}B$. We start by expanding

$$\mathbb{E}\|A - B\|^2 = \mathbb{E}\big[(A - B)^\top(A - B)\big] = \mathbb{E}\|A\|^2 + \mathbb{E}\|B\|^2 - 2\,\mathbb{E}[A^\top B].$$

Each term can be decomposed as follows:

$$\begin{aligned}
\mathbb{E}\|A\|^2 &= \mathrm{tr}(\mathbb{E}[AA^\top]) = \mathrm{tr}\big(\mathrm{Cov}[A] + \mu_A\mu_A^\top\big) \\
&= \mathrm{tr}\,\mathrm{Cov}[A] + \|\mu_A\|^2, \\
\mathbb{E}\|B\|^2 &= \mathrm{tr}(\mathbb{E}[BB^\top]) = \mathrm{tr}\big(\mathrm{Cov}[B] + \mu_B\mu_B^\top\big) \\
&= \mathrm{tr}\,\mathrm{Cov}[B] + \|\mu_B\|^2, \\
\mathbb{E}[A^\top B] &= \mathrm{tr}(\mathbb{E}[AB^\top]) = \mathrm{tr}\big(\mathrm{Cov}[A, B] + \mu_A\mu_B^\top\big) \\
&= \mathrm{tr}\,\mathrm{Cov}[A, B] + \mu_A^\top\mu_B.
\end{aligned}$$

Substituting these expressions back, we obtain

$$\mathbb{E}\|A - B\|^2 = \|\mu_A - \mu_B\|^2 + \mathrm{tr}\,\mathrm{Cov}[A] + \mathrm{tr}\,\mathrm{Cov}[B] - 2\,\mathrm{tr}\,\mathrm{Cov}[A, B].$$

This establishes the bias–variance–covariance identity. $\qquad\square$

**Lemma C.12** (Variance lower bound under local bi-Lipschitz). *Let $f(\boldsymbol{x}) := \boldsymbol{u}^\theta(\boldsymbol{x}, r, s)$ and assume local bi-Lipschitz: there exists $c > 0$ such that for all sufficiently small $\boldsymbol{h}$,*

$$\|f(\boldsymbol{x} + \boldsymbol{h}) - f(\boldsymbol{x})\| \geq c\|\boldsymbol{h}\|.$$

*Fix $\boldsymbol{x}_t$ and let $W$ be a random vector. Define*

$$Z = f(\boldsymbol{x}_t + \delta_1 W) - f(\boldsymbol{x}_t).$$

*Then, conditioning on the $\sigma$-field that renders $\boldsymbol{x}_t$ deterministic,*

$$\mathrm{tr}\,\mathrm{Cov}(Z|\boldsymbol{x}_t) \geq c^2\,\delta_1^2\,\mathrm{tr}\,\mathrm{Cov}(W|\boldsymbol{x}_t).$$

*Consequently,*

$$\mathrm{tr}\,\mathrm{Cov}(Z) \geq c^2\,\delta_1^2\,\mathbb{E}\big[\mathrm{tr}\,\mathrm{Cov}(W|\boldsymbol{x}_t)\big].$$

*Proof.* Set $\bar{W} := \mathbb{E}[W|\boldsymbol{x}_t]$ and write

$$Z = \underbrace{f(\boldsymbol{x}_t + \delta_1 W) - f(\boldsymbol{x}_t + \delta_1 \bar{W})}_{Z'} + \underbrace{f(\boldsymbol{x}_t + \delta_1 \bar{W}) - f(\boldsymbol{x}_t)}_{\text{constant given } \boldsymbol{x}_t}.$$

Adding a constant does not change variance, hence $\operatorname{tr} \operatorname{Cov}(Z|\boldsymbol{x}_t) = \operatorname{tr} \operatorname{Cov}(Z'|\boldsymbol{x}_t)$. By the bi-Lipschitz lower bound,

$$\|Z'\| = \|f(\boldsymbol{x}_t + \delta_1(W - \bar{W})) - f(\boldsymbol{x}_t)\| \geq c\,\delta_1 \|W - \bar{W}\|.$$

Squaring and taking the conditional expectation,

$$\mathbb{E}[\|Z'\|^2|\boldsymbol{x}_t] \geq c^2 \delta_1^2 \mathbb{E}[\|W - \bar{W}\|^2|\boldsymbol{x}_t] = c^2 \delta_1^2 \operatorname{tr} \operatorname{Cov}(W|\boldsymbol{x}_t).$$

Since $\operatorname{tr} \operatorname{Cov}(Z'|\boldsymbol{x}_t) \leq \mathbb{E}[\|Z'\|^2|\boldsymbol{x}_t]$, we obtain $\operatorname{tr} \operatorname{Cov}(Z|\boldsymbol{x}_t) = \operatorname{tr} \operatorname{Cov}(Z'|\boldsymbol{x}_t) \geq c^2 \delta_1^2 \operatorname{tr} \operatorname{Cov}(W|\boldsymbol{x}_t)$. Taking expectation in $\boldsymbol{x}_t$ and using the law of total variance gives the second claim. $\square$

### C.4.3 PROOF FOR THEOREM 3.1

Write $\delta_1 = s - t$, $\delta_2 = r - s$ and note that in inequalities below we only use $\delta_1, \delta_2$.

**Step 1. Upper bound for CTSC (MeanFlow and sCT with linear path).** Here we consider the sampling error from $t$ to $r$, as

$$E_{t,r} = \mathbb{E}[\|X_{t,r}(\boldsymbol{x}_t) - X_{t,r}^\theta(\boldsymbol{x}_t)\|_2^2].$$

It can be written as

$$\mathbb{E}[\|(r-t)\boldsymbol{u}_{t,r}(\boldsymbol{x}_t) - (r-t)\boldsymbol{u}_{t,r}^\theta(\boldsymbol{x}_t)\|_2^2]$$
$$=\mathbb{E}[\|(r-t)\boldsymbol{u}_{t,r}(\boldsymbol{x}_t) - (r-t)\boldsymbol{u}_{t,r}^\theta(\boldsymbol{x}_t) - (r-t)Y_{\text{ctsc}} + (r-t)Y_{\text{ctsc}}\|_2^2]$$
$$\leq 2(\delta_1 + \delta_2)^2\mathbb{E}\|\boldsymbol{u}_{t,r}(\boldsymbol{x}_t) - Y_{\text{ctsc}}\|_2^2 + 2(\delta_1 + \delta_2)^2\mathbb{E}\|\boldsymbol{u}_{t,r}^\theta(\boldsymbol{x}_t) - Y_{\text{ctsc}}\|_2^2$$

where $Y_{\text{ctsc}} = (\delta_1 + \delta_2) \frac{d}{dt}\boldsymbol{u}_{t,r}^\theta(\boldsymbol{x}_t) - \boldsymbol{v}_{t|0}(\boldsymbol{x}_t|\boldsymbol{x}_0)$.

First, consider the first term take

$$A = \boldsymbol{u}_{t,r}(\boldsymbol{x}_t), \qquad B = Y_{\text{ctsc}} = (\delta_1 + \delta_2) \frac{d}{dt}\boldsymbol{u}_{t,r}^\theta(\boldsymbol{x}_t) - \boldsymbol{v}_{t|0}(\boldsymbol{x}_t|\boldsymbol{x}_0).$$

Applying Eq. 67,

$$2(\delta_1 + \delta_2)^2\mathbb{E}\|A - B\|_2^2 = 2(\delta_1 + \delta_2)^2\big(\underbrace{\|\mathbb{E}A - \mathbb{E}B\|^2}_{\text{Bias}_{\text{ctsc-tgt}}^2} + \operatorname{Var}[A] + \operatorname{Var}[B] - 2\operatorname{Cov}(A, B)\big).$$

According to $-2\operatorname{Cov}(A, B) \leq \operatorname{Var}[A] + \operatorname{Var}[B]$, we can get

$$\mathbb{E}\left[\|A - B\|_2^2\right] \tag{68}$$
$$\leq \operatorname{Bias}_{\text{ctsc-tgt}}^2 + 2\operatorname{Var}[\boldsymbol{u}_{t,r}] + 2\operatorname{Var}[Y_{\text{ctsc}}] \tag{69}$$
$$= \operatorname{Bias}_{\text{ctsc-tgt}}^2 + 2\operatorname{Var}[Y_{\text{ctsc}}], \tag{70}$$

because $\operatorname{Var}[\boldsymbol{u}_{t,r}] = 0$.

Then, by Assumption C.9 ($\operatorname{Var}[\boldsymbol{v}_{t|0}] = \sigma_{\boldsymbol{v}_{t|0}}^2$),

$$\operatorname{Var}[Y_{\text{ctsc}}] = \operatorname{Var}\left[(\delta_1 + \delta_2) \frac{d}{dt}\boldsymbol{u}_{t,r}^\theta - \boldsymbol{v}_{t|0}\right]$$
$$\leq 2\operatorname{Var}\left[(\delta_1 + \delta_2) \frac{d}{dt}\boldsymbol{u}_{t,r}^\theta\right] + 2\operatorname{Var}[\boldsymbol{v}_{t|0}]$$
$$= 2(\delta_1 + \delta_2)^2\operatorname{Var}\left[\frac{d}{dt}\boldsymbol{u}_{t,r}^\theta\right] + 2\sigma_{\boldsymbol{v}_{t|0}}^2,$$

Therefore,

$$\mathbb{E}\|\boldsymbol{u}_{t,r}(\boldsymbol{x}_t) - Y_{\text{ctsc}}\|^2$$

$$\leq \text{Bias}^2_{\text{ctsc-tgt}} + 4(\delta_1 + \delta_2)^2 \text{Var}\Big[\frac{d}{dt}\boldsymbol{u}^\theta_{t,r}\Big] + 4\sigma^2_{\boldsymbol{v}_{t|0}}$$

Secondly, take

$$A = \boldsymbol{u}^\theta_{t,r}(\boldsymbol{x}_t), \qquad B = Y_{\text{ctsc}} = (\delta_1 + \delta_2)\frac{d}{dt}\boldsymbol{u}^\theta_{t,r}(\boldsymbol{x}_t) - \boldsymbol{v}_{t|0}(\boldsymbol{x}_t|\boldsymbol{x}_0).$$

and it coincides that $\mathbb{E}\|\boldsymbol{u}^\theta_{t,r}(\boldsymbol{x}_t) - Y_{\text{ctsc}}\|^2_2 = \mathbb{E}[l_{\text{ctsc}}]$. Applying Eq. 67, we have

$$\mathbb{E}[l_{\text{ctsc}}] = \underbrace{\|\mathbb{E}A - \mathbb{E}B\|^2}_{\text{Bias}^2_{\text{ctsc-loss}}} + \text{Var}[A] + \text{Var}[B] - 2\,\text{Cov}(A, B).$$

It can also be easily to obtain

$$\mathbb{E}[l_{\text{ctsc}}] \leq \text{Bias}^2_{\text{ctsc-loss}} + 2\text{Var}[\boldsymbol{u}^\theta_{t,r}(\boldsymbol{x}_t)] + 4(\delta_1 + \delta_2)^2 \text{Var}\Big[\frac{d}{dt}\boldsymbol{u}^\theta_{t,r}(\boldsymbol{x}_t)\Big] + 4\sigma^2_{\boldsymbol{v}_{t|0}},$$

In summary,

$$E_{t,r}$$

$$\leq 2(\delta_1 + \delta_2)^2 \left(\text{Bias}^2_{\text{ctsc-tgt}} + \text{Bias}^2_{\text{ctsc-loss}} + 2\text{Var}[\boldsymbol{u}^\theta_{t,r}(\boldsymbol{x}_t)] + 8(\delta_1 + \delta_2)^2 \text{Var}\Big[\frac{d}{dt}\boldsymbol{u}^\theta_{t,r}(\boldsymbol{x}_t)\Big] + 8\sigma^2_{\boldsymbol{v}_{t|0}}\right)$$

Specifically, when $t = 1, r = 0$,

$$E_{1,0} \leq 2\left(\text{Bias}^2_{\text{ctsc-tgt}} + \text{Bias}^2_{\text{ctsc-loss}} + 2\text{Var}[\boldsymbol{u}^\theta_{t,r}(\boldsymbol{x}_1)] + 8\text{Var}\Big[\frac{d}{dt}\boldsymbol{u}^\theta_{t,r}(\boldsymbol{x}_t)\Big] + 8\sigma^2_{\boldsymbol{v}_{t|0}}\right)\Big|_{r=0,t=1}$$

**Step 2. Upper bound for CT.** Then, let's still consider

$$E_{t,r} = \mathbb{E}[\|X_{t,r}(\boldsymbol{x}_t) - X^\theta_{t,r}(\boldsymbol{x}_t)\|^2_2],$$

by setting $Y_{\text{ct}} = \frac{1}{\delta_1 + \delta_2}\big(\delta_1\,\boldsymbol{v}_{t|0}(\boldsymbol{x}_t|\boldsymbol{x}_0) + \delta_2\,\boldsymbol{u}^\theta_{s,r}(\boldsymbol{x}_t + \delta_1\,\boldsymbol{v}_{t|0}(\boldsymbol{x}_t|\boldsymbol{x}_0))\big)$, which equals to

$$\mathbb{E}[\|(r-t)\boldsymbol{u}_{t,r}(\boldsymbol{x}_t) - (r-t)\boldsymbol{u}^\theta_{t,r}(\boldsymbol{x}_t)\|^2_2]$$

$$= \mathbb{E}[\|(r-t)\boldsymbol{u}_{t,r}(\boldsymbol{x}_t) - (r-t)\boldsymbol{u}^\theta_{t,r}(\boldsymbol{x}_t) - (r-t)Y_{\text{ct}} + (r-t)Y_{\text{ct}}\|^2_2]$$

$$\leq 2(\delta_1 + \delta_2)^2 \mathbb{E}\|\boldsymbol{u}_{t,r}(\boldsymbol{x}_t) - Y_{\text{ct}}\|^2_2 + 2(\delta_1 + \delta_2)^2 \mathbb{E}\|\boldsymbol{u}^\theta_{t,r}(\boldsymbol{x}_t) - Y_{\text{ct}}\|^2_2$$

Firstly, set

$$A = \boldsymbol{u}_{t,r}(\boldsymbol{x}_t), \qquad B = Y_{\text{ct}} = \frac{1}{\delta_1 + \delta_2}\big(\delta_1\,\boldsymbol{v}_{t|0}(\boldsymbol{x}_t|\boldsymbol{x}_0) + \delta_2\,\boldsymbol{u}^\theta_{s,r}(\boldsymbol{x}_t + \delta_1\,\boldsymbol{v}_{t|0}(\boldsymbol{x}_t|\boldsymbol{x}_0))\big).$$

By Cauchy-Schwarz, $-2\,\text{Cov}(A, B)$ is again absorbed into the $\lesssim$ notation. So we have

$$\mathbb{E}[\|A - B\|^2_2] \tag{71}$$

$$\leq \text{Bias}^2_{\text{ct-tgt}} + 2\text{Var}[\boldsymbol{u}_{t,r}] + 2\text{Var}[Y_{\text{ct}}] \tag{72}$$

$$= \text{Bias}^2_{\text{ct-tgt}} + 2\text{Var}[Y_{\text{ct}}], \tag{73}$$

For $\text{Var}[Y_{\text{ct}}]$, expand the second term as

$$\boldsymbol{u}^\theta_{s,r}(\boldsymbol{x}_t + \delta_1\,\boldsymbol{v}_{t|0}) = \boldsymbol{u}^\theta_{s,r}(\boldsymbol{x}_t) + \Big(\boldsymbol{u}^\theta_{s,r}(\boldsymbol{x}_t + \delta_1\,\boldsymbol{v}_{t|0}) - \boldsymbol{u}^\theta_{s,r}(\boldsymbol{x}_t)\Big).$$

Apply Lemma C.12 with $f(\boldsymbol{x}) = \boldsymbol{u}^\theta_{s,r}(\boldsymbol{x})$ and $W = \boldsymbol{v}_{t|0}(\boldsymbol{x}_t|\boldsymbol{x}_0)$. Conditioning on $\boldsymbol{x}_t$ and using the Lipschitz upper constant $\ell > 0$, we obtain

$$\text{Var}\Big(\boldsymbol{u}^\theta_{s,r}(\boldsymbol{x}_t + \delta_1\,\boldsymbol{v}_{t|0}) - \boldsymbol{u}^\theta_{s,r}(\boldsymbol{x}_t)\,\Big|\,\boldsymbol{x}_t\Big) \leq \ell^2\,\delta^2_1\,\text{Var}\big(\boldsymbol{v}_{t|0}(\boldsymbol{x}_t|\boldsymbol{x}_0)\,\big|\,\boldsymbol{x}_t\big).$$

Taking expectation in $\boldsymbol{x}_t$ yields

$$\mathrm{Var}\Big[\boldsymbol{u}_{s,r}^{\theta}(\boldsymbol{x}_t + \delta_1 \, \boldsymbol{v}_{t|0}) - \boldsymbol{u}_{s,r}^{\theta}(\boldsymbol{x}_t)\Big] \leq \ell^2 \, \delta_1^2 \, \sigma_{\boldsymbol{v}_{t|0}}^2.$$

Therefore,

$$\mathrm{Var}[Y_{\mathrm{ct}}] \leq \frac{1}{(\delta_1 + \delta_2)^2}\Big(2\delta_1^2 \, \sigma_{\boldsymbol{v}_{t|0}}^2 \; + \; 2\delta_2^2 \, \mathrm{Var}\big[\boldsymbol{u}_{s,r}^{\theta}(\boldsymbol{x}_t)\big] \; + \; 2l^2 \, \delta_1^2 \, \delta_2^2 \, \sigma_{\boldsymbol{v}_{t|0}}^2\Big).$$

Hence, we get

$$\mathbb{E}\|\boldsymbol{u}_{t,r}(\boldsymbol{x}_t) - Y_{\mathrm{ct}}\|_2^2 \leq \mathrm{Bias}_{\mathrm{ct\text{-}tgt}}^2 + \frac{4}{(\delta_1 + \delta_2)^2}\Big(\delta_2^2 \, \mathrm{Var}\big[\boldsymbol{u}_{s,r}^{\theta}(\boldsymbol{x}_t)\big] \; + \; (1 + \ell^2\delta_2^2)\delta_1^2 \, \sigma_{\boldsymbol{v}_{t|0}}^2\Big).$$

Secondly, set

$$A \; = \; \boldsymbol{u}_{t,r}^{\theta}(\boldsymbol{x}_t), \qquad B \; = \; Y_{\mathrm{ct}} \; = \; \frac{1}{\delta_1 + \delta_2}\big(\delta_1 \, \boldsymbol{v}_{t|0}(\boldsymbol{x}_t|\boldsymbol{x}_0) \; + \; \delta_2 \, \boldsymbol{u}_{s,r}^{\theta}(\boldsymbol{x}_t + \delta_1 \, \boldsymbol{v}_{t|0}(\boldsymbol{x}_t|\boldsymbol{x}_0)))\big).$$

Then $\mathbb{E}\|\boldsymbol{u}_{t,r}^{\theta}(\boldsymbol{x}_t) - Y_{\mathrm{ct}}\|_2^2 = \mathbb{E}[l_{\mathrm{ct}}]$. Easily following the above derivation, we can obtain

$$\mathbb{E}[l_{\mathrm{ct\text{-}loss}}] \leq \mathrm{Bias}_{\mathrm{ct\text{-}loss}}^2 + 2\mathrm{Var}\big[\boldsymbol{u}_{t,r}^{\theta}(\boldsymbol{x}_t)\big] + \frac{4}{(\delta_1 + \delta_2)^2}\Big(\delta_2^2 \, \mathrm{Var}\big[\boldsymbol{u}_{s,r}^{\theta}(\boldsymbol{x}_t)\big] + (1 + \ell^2\delta_2^2)\delta_1^2 \, \sigma_{\boldsymbol{v}_{t|0}}^2\Big).$$

To sum up, we have

$$E_{t,r} \leq 2(\delta_1 + \delta_2)^2\Big(\mathrm{Bias}_{\mathrm{ct\text{-}tgt}}^2 + \mathrm{Bias}_{\mathrm{ct\text{-}loss}}^2 + 2\mathrm{Var}\big[\boldsymbol{u}_{t,r}^{\theta}(\boldsymbol{x}_t)\big]$$
$$+ \frac{8}{(\delta_1 + \delta_2)^2}\Big(\delta_2^2 \, \mathrm{Var}\big[\boldsymbol{u}_{s,r}^{\theta}(\boldsymbol{x}_t)\big] + (1 + \ell^2\delta_2^2)\delta_1^2 \, \sigma_{\boldsymbol{v}_{t|0}}^2\Big)\Big).$$

When $t = 0, r = 1$, the inequality becomes

$$E_{1,0} \leq 2\Big(\mathrm{Bias}_{\mathrm{ct\text{-}tgt}}^2 + \mathrm{Bias}_{\mathrm{ct\text{-}loss}}^2 + 2\mathrm{Var}\big[\boldsymbol{u}_{t,r}^{\theta}(\boldsymbol{x}_t)\big]$$
$$+ 8\delta_2^2 \, \mathrm{Var}\big[\boldsymbol{u}_{s,r}^{\theta}(\boldsymbol{x}_t)\big] + 8(1 + \ell^2\delta_2^2)\delta_1^2 \, \sigma_{\boldsymbol{v}_{t|0}}^2\Big)\Big|_{r=0,t=1}.$$

**Step 3. Upper bound for SCD.**    In this case, consider

$$E_{t,r} = \mathbb{E}[\|X_{t,r}(\boldsymbol{x}_t) - X_{t,r}^{\theta}(\boldsymbol{x}_t)\|_2^2],$$

by setting $Y_{\mathrm{scd}} \; = \; \frac{1}{\delta_1 + \delta_2}\big(\delta_1 \, \boldsymbol{u}_{t,s}^{\theta}(\boldsymbol{x}_t) \; + \; \delta_2 \, \boldsymbol{u}_{s,r}^{\theta}\big(\boldsymbol{x}_t + \delta_1 \, \boldsymbol{u}_{t,s}^{\theta}(\boldsymbol{x}_t)\big)\big)$, which equals to

$$\mathbb{E}[\|(r - t)\boldsymbol{u}_{t,r}(\boldsymbol{x}_t) - (r - t)\boldsymbol{u}_{t,r}^{\theta}(\boldsymbol{x}_t)\|_2^2]$$
$$= \mathbb{E}[\|(r - t)\boldsymbol{u}_{t,r}(\boldsymbol{x}_t) - (r - t)\boldsymbol{u}_{t,r}^{\theta}(\boldsymbol{x}_t) - (r - t)Y_{\mathrm{scd}} + (r - t)Y_{\mathrm{scd}}\|_2^2]$$
$$\leq 2(\delta_1 + \delta_2)^2\mathbb{E}\|\boldsymbol{u}_{t,r}(\boldsymbol{x}_t) - Y_{\mathrm{scd}}\|_2^2 + 2(\delta_1 + \delta_2)^2\mathbb{E}\|\boldsymbol{u}_{t,r}^{\theta}(\boldsymbol{x}_t) - Y_{\mathrm{scd}}\|_2^2$$

We can find that the only difference between SCD and CT is the second term in $Y_{\mathrm{scd}}$. So we only need to analyze this term:

$$\boldsymbol{u}_{s,r}^{\theta}(\boldsymbol{x}_t + \delta_1 \, \boldsymbol{u}_{t,s}^{\theta}(\boldsymbol{x}_t)) = \boldsymbol{u}_{s,r}^{\theta}(\boldsymbol{x}_t) \; + \; \Big(\boldsymbol{u}_{s,r}^{\theta}(\boldsymbol{x}_t + \delta_1 \, \boldsymbol{u}_{t,s}^{\theta}(\boldsymbol{x}_t)) - \boldsymbol{u}_{s,r}^{\theta}(\boldsymbol{x}_t)\Big).$$

By Assumption C.10 (Lipschitz continuity), this term contributes a variance of order $\ell^2 \, \delta_1^2 \, \mathrm{Var}[\boldsymbol{u}_{t,s}^{\theta}(\boldsymbol{x}_t)]$. Hence,

$$\mathrm{Var}[Y_{\mathrm{scd}}] \; \leq \; \delta_1^2 \, \mathrm{Var}\big[\boldsymbol{u}_{t,s}^{\theta}(\boldsymbol{x}_t)\big] \; + \; \delta_2^2 \, \mathrm{Var}\big[\boldsymbol{u}_{s,r}^{\theta}(\boldsymbol{x}_t)\big] \; + \; \ell^2 \, \delta_1^2\delta_2^2 \, \mathrm{Var}\big[\boldsymbol{u}_{t,s}^{\theta}(\boldsymbol{x}_t)\big].$$

Similar to the derivation of CT, we can get

$$E_{t,r} \leq 2(\delta_1 + \delta_2)^2\Big(\mathrm{Bias}_{\mathrm{scd\text{-}tgt}}^2 + \mathrm{Bias}_{\mathrm{scd\text{-}loss}}^2 + 2\mathrm{Var}\big[\boldsymbol{u}_{t,r}^{\theta}(\boldsymbol{x}_t)\big]$$
$$+ \frac{8}{(\delta_1 + \delta_2)^2}\Big(\delta_2^2 \, \mathrm{Var}\big[\boldsymbol{u}_{s,r}^{\theta}(\boldsymbol{x}_t)\big] + (1 + \ell^2\delta_2^2)\delta_1^2\mathrm{Var}\big[\boldsymbol{u}_{t,s}^{\theta}(\boldsymbol{x}_t)\big]\Big)\Big).$$

When $t = 0, r = 1$, the inequality becomes

$$E_{1,0} \leq 2\Big(\mathrm{Bias}_{\mathrm{scd\text{-}tgt}}^2 + \mathrm{Bias}_{\mathrm{scd\text{-}loss}}^2 + 2\mathrm{Var}\big[\boldsymbol{u}_{t,r}^{\theta}(\boldsymbol{x}_t)\big]$$
$$+ 8\delta_2^2 \, \mathrm{Var}\big[\boldsymbol{u}_{s,r}^{\theta}(\boldsymbol{x}_t)\big] + 8(1 + \ell^2\delta_2^2)\delta_1^2\mathrm{Var}\big[\boldsymbol{u}_{t,s}^{\theta}(\boldsymbol{x}_t)\big]\Big)\Big|_{r=0,t=1}.$$

**Step 4. Conclusion** Since

$$W_2^2(p_0, p_0^\theta) \le \mathbb{E}\|X_{1,0}(\boldsymbol{x}_1) - X_{1,0}^\theta(\boldsymbol{x}_1)\|_2^2 = E_{1,0}$$

The proposition is proved.

## C.5 PROOF OF IDEAL VELOCITY AND ITS BIAS-VARIANCE ANALYSIS (PROP. 4.1)

### C.5.1 THE FORM OF IDEAL VELOCITY

*Proof.* By definition,

$$\boldsymbol{v}_t = \int \boldsymbol{v}_t(\boldsymbol{x}_t|\boldsymbol{x}_0)\frac{p_t(\boldsymbol{x}_t|\boldsymbol{x}_0)}{p_t(\boldsymbol{x}_t)}p_0(\boldsymbol{x}_0)d\boldsymbol{x}_0$$

We aim to rewrite

$$\boldsymbol{v}_t(\boldsymbol{x}_t) = \int \boldsymbol{v}_t(\boldsymbol{x}_t \mid \boldsymbol{x}_0)\frac{p_t(\boldsymbol{x}_t \mid \boldsymbol{x}_0)\,p_0(\boldsymbol{x}_0)}{p_t(\boldsymbol{x}_t)}\,d\boldsymbol{x}_0 = \mathbb{E}_{p(\boldsymbol{x}_0|\boldsymbol{x}_t)}[\boldsymbol{v}_t(\boldsymbol{x}_t \mid \boldsymbol{x}_0)]. \tag{74}$$

The forward (noising) process is linear Gaussian:

$$\boldsymbol{x}_t = \alpha_t\boldsymbol{x}_0 + \sigma_t\boldsymbol{\varepsilon}, \qquad \boldsymbol{\varepsilon} \sim \mathcal{N}(\boldsymbol{0}, \boldsymbol{I}), \tag{75}$$

so that

$$p_t(\boldsymbol{x}_t \mid \boldsymbol{x}_0) = \mathcal{N}\big(\boldsymbol{x}_t;\, \alpha_t\boldsymbol{x}_0,\, \sigma_t^2\boldsymbol{I}\big). \tag{76}$$

Assume the conditional velocity has the form

$$\boldsymbol{v}_t(\boldsymbol{x}_t \mid \boldsymbol{x}_0) = \hat{\alpha}_t\boldsymbol{x}_0 + \hat{\sigma}_t\boldsymbol{\varepsilon}. \tag{77}$$

Since $\boldsymbol{\varepsilon} = (\boldsymbol{x}_t - \alpha_t\boldsymbol{x}_0)/\sigma_t$, we can eliminate the noise and write

$$\boldsymbol{v}_t(\boldsymbol{x}_t \mid \boldsymbol{x}_0) = \tfrac{\hat{\sigma}_t}{\sigma_t}\boldsymbol{x}_t + \left(\hat{\alpha}_t - \tfrac{\hat{\sigma}_t\alpha_t}{\sigma_t}\right)\boldsymbol{x}_0 \tag{78}$$

$$\triangleq a_t\boldsymbol{x}_t + b_t\boldsymbol{x}_0. \tag{79}$$

Suppose the prior $p_0$ is empirical:

$$p_0(\boldsymbol{y}) = \frac{1}{N}\sum_{i=1}^{N}\mathbb{1}_{\boldsymbol{y}_i}(\boldsymbol{y}). \tag{80}$$

Then the marginal and posterior are finite mixtures:

$$p_t(\boldsymbol{x}_t) = \tfrac{1}{N}\sum_{i=1}^{N}\mathcal{N}\big(\boldsymbol{x}_t;\, \alpha_t\boldsymbol{y}_i,\, \sigma_t^2\boldsymbol{I}\big), \tag{81}$$

$$p(\boldsymbol{x}_0 = \boldsymbol{y}_i \mid \boldsymbol{x}_t) = \frac{w_i(\boldsymbol{x}_t)}{\sum_{j=1}^{N}w_j(\boldsymbol{x}_t)}, \qquad w_i(\boldsymbol{x}_t) \triangleq \mathcal{N}\big(\boldsymbol{x}_t;\, \alpha_t\boldsymbol{y}_i,\, \sigma_t^2\boldsymbol{I}\big). \tag{82}$$

Taking the expectation of the linear form yields

$$\boldsymbol{v}_t(\boldsymbol{x}_t) = a_t\boldsymbol{x}_t + b_t\,\mathbb{E}[\boldsymbol{x}_0 \mid \boldsymbol{x}_t]. \tag{83}$$

From Bayes' rule,

$$\mathbb{E}[\boldsymbol{x}_0 \mid \boldsymbol{x}_t] = \sum_{i=1}^{N}\pi_i(\boldsymbol{x}_t)\,\boldsymbol{y}_i, \qquad \pi_i(\boldsymbol{x}_t) = \frac{w_i(\boldsymbol{x}_t)}{\sum_{j=1}^{N}w_j(\boldsymbol{x}_t)}. \tag{84}$$

In conclusion, under the empirical prior, $\boldsymbol{v}_t(\boldsymbol{x}_t)$ is obtained as a posterior-weighted average of the conditional velocities associated with each training sample $\boldsymbol{y}_i$:

$$\boldsymbol{v}_t^*(\boldsymbol{x}_t) = \sum_{i=1}^{N}p_0(\boldsymbol{x}_0 = \boldsymbol{y}_i \mid \boldsymbol{x}_t)\,\boldsymbol{v}_t(\boldsymbol{x}_t \mid \boldsymbol{y}_i). \tag{85}$$

$\square$

### C.5.2 The Bias and Variance of Plug-in Velocity

Under mild assumptions and with the Bias-Variance Decomposition, we can analyze $\mathbb{E}\|v_t^* - v_t\|^2$, which consists of the bias term and variance term. The proposition below shows that although there is an increase in bias of order $\mathcal{O}(1/N)$, the variance is significantly reduced by $\mathcal{O}(1 - 1/N)$.

**Proposition C.13** (Bias-Variance Decomposition of Ideal Velocity). *Assume there are the empirical distribution $p_{emp}$ on any $\{y^{(i)}\}_{i=1}^N$ and the data distribution $p_0$ has the finite normalization constant*

$$Z(p_{emp} \mid \{y\}_{i=1}^N), Z(p_0) \geq z_0 > 0.$$

*Suppose $\exists M_1, M_2 > 0, s > \frac{d}{2}, s.t. \|v_t(x_t \mid x_0)\|_{\mathcal{C}^s} \leq M_1$, and $\|v_t(x_t)\| \leq M_2$. Then we have*

$$\mathbb{E}\|v_t^* - v_t\|^2 \leq C\left(\frac{M_1^2 + 2M_2^2}{N} + \frac{4\,\mathrm{Var}\left[v_t(x_t|x_0)\right]}{N}\right).$$

*Proof.* According to Eq. 67

$$\mathbb{E}\|A - B\|^2 = \|\mu_A - \mu_B\|^2 + \mathrm{Var}[A] + \mathrm{Var}[B] - 2\,\mathrm{Cov}[A, B],$$

and the Cauchy-Schwarz inequality

$$-2\,\mathrm{Cov}(A, B) \leq \mathrm{Var}[A] + \mathrm{Var}[B],$$

we have

$$\mathbb{E}\|v_t - v_t^*\|^2 \leq (\mathbb{E}v_t - \mathbb{E}v_t^*)^2 + 2\,\mathrm{Var}[v_t^*] + 2\,\mathrm{Var}[v_t]$$

So we analyze the bias and variance of $v_t^*$ below.

**Bias of plug-in velocity.**

$$|\mathbb{E}[v^*] - \mathbb{E}[v_t]| \leq \frac{1}{z_0}|\mathbb{E}[p_{emp}(x_0)p(x_t \mid x_0)v_t(x_t \mid x_0) - p_0(x_0)p(x_t \mid x_0)v_t(x_t \mid x_0)]|$$
$$\leq M_1\mathbb{E}[\|p_{emp} - p_0\|_{C_1^s}]$$

where

$$\|\nu\|_{C_1^s} := \sup\left\{\int f\,d\nu : f \in C^s(\Omega),\ \|f\|_{C^s} \leq 1\right\}.$$

The previous work (Kloeckner, 2018) has proven that

$$\mathbb{E}[\|p_{emp} - p_0\|_{C_1^s}] \leq \frac{C}{\sqrt{N}},$$

so we finally get

$$|\mathbb{E}[v^*] - \mathbb{E}[v_t]|^2 \leq \frac{CM_1^2}{N}.$$

**Variance of plug-in velocity.** Let

$$Z(x_t, \{y^{(i)}\}_{i=1}^N, t) := \int p(x_t \mid x_0)p_{emp}(x_0)dx = \frac{1}{N}\sum_{i=1}^N p(x_t \mid y^{(i)}).$$

Then, under the empirical distribution, we can write

$$v_t^* = \frac{1}{NZ(x_t, \{y^{(i)}\}_{i=1}^N, t)}\sum_{i=1}^N v_t(x_t|y^{(i)})p(x_t|y^{(i)})$$

which leads to the variance of $v_t^*$ as

$$
\begin{aligned}
\mathrm{Var}[v_t^*] &\leq \frac{C}{N^2} \sum_{i=1}^N \mathrm{Var}\left[v_t(x_t|y^{(i)})p(x_t|y^{(i)})\right] \\
&= \frac{C}{N^2} \sum_{i=1}^N \left(\mathbb{E}\left[v_t(x_t|y^{(i)})^2 p(x_t|y^{(i)})^2\right] - \left(\mathbb{E}\left[v_t(x_t|y^{(i)})p(x_t|y^{(i)})\right]\right)^2\right) \\
&\leq \frac{C}{N^2} \sum_{i=1}^N \mathbb{E}\left[v_t(x_t|y^{(i)})^2 p(x_t|y^{(i)})^2\right] \\
&\leq \frac{C'}{N^2} \sum_{i=1}^N \mathbb{E}\left[v_t(x_t|y^{(i)})^2\right] \\
&\leq \frac{C'}{N^2} \sum_{i=1}^N \left(\mathrm{Var}\left[v_t(x_t|y^{(i)})\right] + \left(\mathbb{E}\left[v_t(x_t|y^{(i)})\right]\right)^2\right) \\
&= \frac{C'}{N^2} \sum_{i=1}^N \left(\mathrm{Var}\left[v_t(x_t|y^{(i)})\right] + v_t(x_t)^2\right) \\
&= \frac{C'}{N} \left(\mathrm{Var}\left[v_t(x_t|x_0)\right] + M_2^2\right).
\end{aligned}
$$

$\square$

## C.6    The Convergence of CTSC Loss Employing Plug-in Velocity (Sec. 4)

**Lemma C.14.** *Define the normalization constant as*

$$
Z(q) := \int p(x_t \mid y)q(y)dy.
$$

*For the weight function*

$$
w_q(y) := \frac{q(y)p(x_t \mid y)}{Z(q)},
$$

*if there are two distribution $q$ and $r$ with finite normalization constant*

$$
Z(q), Z(r) \geq z_0 > 0,
$$

*the following inequalities holds*

$$
|w_q(y) - w_r(y)| \leq \left(\frac{1}{(2\pi)^{d/2}\sigma_t^d z_0} + \frac{L}{(2\pi)^{d/2}\sigma_t^d z_0^2}\right) W_1(q, r)
$$

$$
|w_q(y)^2 - w_r(y)^2| \leq \left(\frac{1}{(2\pi)^d \sigma_t^2 d z_0^2} + \frac{L}{(2\pi)^d \sigma_t^2 d z_0^3}\right) W_1(q, r).
$$

*Proof.* First, since $p(x_t \mid y) = \mathcal{N}(x_t; y, \sigma_t^2 I)$ There exist constants $L > 0$ such that

$$
0 < p(x_t \mid y) \leq \frac{1}{(2\pi)^{d/2}\sigma_t^d}, \qquad |p(x_t \mid y) - p(x_t \mid y')| \leq L\|y - y'\|.
$$

Denote $p(x_t \mid y)$ as $K_t(y)$. We decompose

$$
w_q(y) - w_r(y) = K_t(y)\left(\frac{q(y)}{Z(q)} - \frac{r(y)}{Z(r)}\right) = K_t(y)\left(\frac{q(y) - r(y)}{Z(q)} + r(y)\frac{Z(r) - Z(q)}{Z(q)Z(r)}\right).
$$

Then, by Kantorovich–Rubinstein duality,

$$
|Z(q) - Z(r)| = \left|\int K_t \, d(q - r)\right| \leq L W_1(q, r).
$$

Also, using $Z(q), Z(r) \geq z_0$ and $K_t(\boldsymbol{y}) \leq \frac{1}{(2\pi)^{d/2}\sigma_t^d}$, we have

$$
\begin{aligned}
|w_q(\boldsymbol{y}) - w_r(\boldsymbol{y})| &\leq \frac{1}{(2\pi)^{d/2}\sigma_t^d z_0}|q(\boldsymbol{y}) - r(\boldsymbol{y})| + \frac{1}{(2\pi)^{d/2}\sigma_t^d z_0^2}|Z(q) - Z(r)| \\
&\leq \Big( \frac{1}{(2\pi)^{d/2}\sigma_t^d z_0} + \frac{L}{(2\pi)^{d/2}\sigma_t^d z_0^2} \Big) W_1(q,r).
\end{aligned}
$$

Next, since $0 \leq w_q(\boldsymbol{y}) \leq \frac{1}{\sqrt{2\pi}\sigma_t}/z_0$, we bound the squared difference:

$$
\begin{aligned}
|w_q(\boldsymbol{y})^2 - w_r(\boldsymbol{y})^2| &= |w_q(\boldsymbol{y}) - w_r(\boldsymbol{y})|(w_q(\boldsymbol{y}) + w_r(\boldsymbol{y})) \\
&\leq \frac{1}{(2\pi)^{d/2}\sigma_t^d z_0} \Big( \frac{1}{(2\pi)^{d/2}\sigma_t^d z_0} + \frac{L}{(2\pi)^{d/2}\sigma_t^d z_0^2} \Big) W_1(q,r) \\
&= \Big( \frac{1}{(2\pi)^d\sigma_t^2 d z_0^2} + \frac{L}{(2\pi)^d\sigma_t^2 d z_0^3} \Big) W_1(q,r).
\end{aligned}
$$

$\square$

**Proposition C.15.** *Denote the empirical distribution as $p_{emp}$, then the difference between the training loss employing plug-in velocity and marginal velocity can be bounded by the Wasserstein distance between $p_{emp}$ and $p_0$ as*

$$
\big| \mathcal{L}_{plug\text{-}in}(\theta) - \mathcal{L}_{marginal}(\theta) \big| \leq CW(p_{emp}, p_0).
$$

*Proof.* Substituting the form of plug-in velocity in Eq. 12, we have

$$
\begin{aligned}
\mathcal{L}_{\text{plug-in}}(\theta) &= \mathbb{E}\big[ \|\boldsymbol{u}_{t,r}(\boldsymbol{x}_t) - (r-t)\frac{d}{dt}\boldsymbol{u}_{t,r}^\theta(\boldsymbol{x}_t) - \boldsymbol{v}_t^*(\boldsymbol{x}_t|\{\boldsymbol{y}^{(i)}\}_{i=1}^N)\|^2 \big] \\
&= \mathbb{E}\Big[ \|\boldsymbol{u}_{t,r}(\boldsymbol{x}_t) - (r-t)\frac{d}{dt}\boldsymbol{u}_{t,r}^\theta(\boldsymbol{x}_t) \\
&\qquad - \sum_i^N \frac{\mathcal{N}(\boldsymbol{x}_t; \alpha_t\boldsymbol{y}^{(i)}, \sigma_t^2\mathbf{I})}{\sum_j^N \mathcal{N}(\boldsymbol{x}_t; \alpha_t\boldsymbol{y}^{(j)}, \sigma_t^2\mathbf{I})}(\dot{\alpha}_t\boldsymbol{y}^{(i)} + \frac{\dot{\sigma}_t}{\sigma_t}(\boldsymbol{x}_t - \alpha_t\boldsymbol{y}^{(i)}))\|^2 \Big] \\
&= \mathbb{E}\Big[ \sum_i^N \Big( \Big( \frac{\mathcal{N}(\boldsymbol{x}_t; \alpha_t\boldsymbol{y}^{(i)}, \sigma_t^2\mathbf{I})}{\sum_j^N \mathcal{N}(\boldsymbol{x}_t; \alpha_t\boldsymbol{y}^{(j)}, \sigma_t^2\mathbf{I})} \Big)^2 \cdot \\
&\qquad \|\boldsymbol{u}_{t,r}(\boldsymbol{x}_t) - (r-t)\frac{d}{dt}\boldsymbol{u}_{t,r}^\theta(\boldsymbol{x}_t) - (\dot{\alpha}_t\boldsymbol{y}^{(i)} + \frac{\dot{\sigma}_t}{\sigma_t}(\boldsymbol{x}_t - \alpha_t\boldsymbol{y}^{(i)}))\|^2 \Big) \Big].
\end{aligned}
$$

We denote

$$
w(\boldsymbol{x}_t, \boldsymbol{y}, \{\boldsymbol{y}^{(i)}\}_{i=1}^N, t) := \frac{\mathcal{N}(\boldsymbol{x}_t; \alpha_t\boldsymbol{y}, \sigma_t^2\mathbf{I})}{\sum_j^N \mathcal{N}(\boldsymbol{x}_t; \alpha_t\boldsymbol{y}^{(j)}, \sigma_t^2\mathbf{I})},
$$

$$
l(\boldsymbol{x}_t, \boldsymbol{y}, t, r, \theta) := \|\boldsymbol{u}_{t,r}(\boldsymbol{x}_t) - (r-t)\frac{d}{dt}\boldsymbol{u}_{t,r}^\theta(\boldsymbol{x}_t) - (\dot{\alpha}_t\boldsymbol{y} + \frac{\dot{\sigma}_t}{\sigma_t}(\boldsymbol{x}_t - \alpha_t\boldsymbol{y}))\|^2,
$$

then

$$
\mathcal{L}_{\text{plug-in}}(\theta) = \mathbb{E}\Big[ \sum_{i=1}^N \Big( w^2(\boldsymbol{x}_t, \boldsymbol{y}^{(i)}, \{\boldsymbol{y}^{(i)}\}_{i=1}^N, t) l(\boldsymbol{x}_t, \boldsymbol{y}^{(i)}, t, r, \theta) \Big) \Big].
$$

Recall that $p_{\text{emp}}(\boldsymbol{y}) = \frac{1}{N}\sum_{i=1}^N \mathbb{1}_{\boldsymbol{y}_i}(\boldsymbol{y})$, then we can obtain

$$
w(\boldsymbol{x}_t, \boldsymbol{y}, \{\boldsymbol{y}^{(i)}\}_{i=1}^N, t) = w_{p_{\text{emp}}}(\boldsymbol{x}_t, \boldsymbol{y}).
$$

And the training loss with marginal velocity is

$$
\mathcal{L}_{\text{marginal}}(\theta) = \mathbb{E}\Big[ \sum_{i=1}^N \Big( w_{p_0}^2(\boldsymbol{x}_t, \boldsymbol{y}) l(\boldsymbol{x}_t, \boldsymbol{y}^{(i)}, t, r, \theta) \Big) \Big].
$$

Finally, by the boundedness of $\ell$ and Lemma C.14, we get

$$
\begin{aligned}
\left| \mathcal{L}_{\text{plug-in}}(\theta) - \mathcal{L}_{\text{marginal}}(\theta) \right| &= \mathbb{E}\Big[ \sum_{i=1}^{N} \big( w_q^2 - w_r^2 \big) \ell(x_t, y^{(i)}, t, r, \theta) \Big] \\
&\leq C \sup_{y} |w_q^2 - w_r^2| \\
&\leq C W_1(p_{\text{emp}}, p_0).
\end{aligned}
$$

$\square$

**Remark C.16.** *We point out that the Wasserstein-1 distance between the empirical distribution and the true distribution decreases as the number of data samples increases, which has been established in previous literature (Fournier & Guillin, 2015).*

## D  EXPERIMENTAL DETAILS

### D.1  DETAILS FOR EMPIRICAL ANALYSIS OF FIG. 2

We conduct experiments on unconditional generation on CIFAR-10, conditional generation on ImageNet-256×256 with or without the classifier-free-guidance setting. We here give more details of each method's setting of implementation.

**Uncond. CIFAR-10.**  We use a unified setting with batch size as 512 and iteration number as 800k (~8000 epochs). For stability, we adopt exponential moving average (EMA) to update the model parameters, with decay ratio set to either 0.99995 or 0.9999. We find that 0.9999 ema decay usually performs better under 800k iteration with batchsize 512. We report results using the best-performing EMA setting. For all the experimental trials, we trained them with Nvidia-A100×8. The detailed hyperparameter configurations for each model are as follows:

- **CT** and **CT-linear**. Both variants adopt LPIPS as the loss metric, where the difference lies in the choice of time path: the former uses a cosine path while the latter employs a linear path. We set the learning rate in training to 2e-4. Following the official JAX implementation, we adopt a progressive time sampler such that the scale $K_{\min}$ is initialized at 2 and gradually increased to a maximum of $K_{\max} = 150$. This implies that the interval $[\sigma_{\min}, \sigma_{\max}]$ is partitioned according to

$$
\left\{ [\sigma_{\max} + \tfrac{h}{K}\big(\sigma_{\min}^{1/\rho} - \sigma_{\max}^{1/\rho}\big)]^{\rho} \right\}_{h=1}^{K},
$$

  with $\sigma_{\min} = 0.002$, $\sigma_{\max} = 80.0$. After the change-of-variable, a $\frac{2}{\pi}\arctan()$ is operated to scale the time from [0.002, 80] to [0,1] in cosine path, while in linear path, the sampled time is normalized by $\frac{t}{t+1} \in [0, 1]$. In addition, a curriculum learning strategy is introduced to regulate the evolution of $K$ with respect to the training iterations. When updating the model inside $\text{sg}(\cdot)$ via EMA, a decay rate $r_{\text{ema}}$ is employed to further stabilize training. In detail, at training step $j \in \{1, \ldots, J\}$ with total steps $J = 800$k, the progressive scale $K(j)$ and the corresponding EMA decay rate $r_{\text{ema}}(j)$ are computed as

$$
K(j) = \left\lceil \sqrt{\frac{j}{J}\big((K_{\max} + 1)^2 - K_{\min}^2\big) + K_{\min}^2} - 1 \right\rceil + 1,
$$

$$
r_{\text{ema}}(j) = \exp\left( -\frac{-\log(r_{\text{ema-min}})\, K_{\min}}{K(J)} \right).
$$

  Here $K(j)$ is lower bounded by 1, and $r_{\text{ema}}(j)$ smoothly interpolates between $r_{\text{ema-min}} = 0.9$ and 1 as training progresses.

- **SCD**. For SCD, since the official release does not include the configuration for training on CIFAR, we use the same hyperparameter settings as those used for ImageNet in the

official release. $K$ defined as the total number of steps that we divide the time interval into is set as 128, and the $p_{\text{teq}} = 0.25$ as the probability of training the average velocity with instantaneous conditional velocity supervision as described in Sec. 2.3. We set the learning rate in training as 1e-4.

- **IMM**. Unlike the summary in Table 1, IMM here employs a cosine path with an EDM preconditioner. $M$ as the group size is set as 4 and $\gamma = 12$ for calculating the difference between $s$ and $t$, as its default configuration. For the grouped kernel function, it is implemented by the RBF kernel. We set the learning rate in training to 1e-4.

- **sCT** and **sCT-linear**. In time sampler, $P_{\text{mean}} = -1$ and $P_{\text{std}} = 1.4$. Tangent warmup iteration for gradient ratio is set as 10000. We set the learning rate in training to 1e-4. Besides, the variational adaptive weighting techniques are not employed for better understanding the modularized contribution of each models, while the tangent normalization is employed in the sCT for stabler training, but not implemented in sCT-linear.

- **MeanFlow**. In time sampler, $P_{\text{mean}} = -2.0$ and $P_{\text{std}} = 2.0$. The $p_{\text{teq}} = 0.25$ as the probability of training the average velocity with instantaneous conditional velocity supervision. We set the learning rate in training to 6e-4. The power for adaptive weighting is 0.75.

Moreover, for CIFAR-10, to enable a fairer comparison, we keep the models identical except for the time sampler. Specifically, we disable adaptive loss in MeanFlow, variational adaptive weighting in sCT, and tangent warmup, and instead use a squared $l_2$ loss with a learning rate of 2e-4. Under this setting, with $p_{\text{teq}} = 0.25$, we obtain FID50k of 4.64 for MeanFlow and 4.81 for sCT-linear on CIFAR-10, which also validates our conclusion in the Sec. 3.

**Cond. ImageNet.** In this setting, we include the class label as part of the network input for conditional training. Since CTs require LPIPS as their loss metric, replacing it with a squared $l_2$ loss on latents causes training to diverge. For all the experimental trials, we trained them with Nvidia-A100×8. Therefore, we do not report CTs results in the latent space. For the other models, the settings are as follows:

- **SCD**. The configuration is identical to that used for CIFAR-10.

- **IMM**. It is implemented with a linear path in latent space. $M$ as the group size is set as 4 and $\gamma = 12$. We observed that **IMM fails to converge** (FID does not decrease to a reasonable range) on SiT-B/2 when the $B \in \{512, 1024\}$. Convergence appears only when we increase batch size $B$ to 2048, at which point the model begins to generate valid images. This phenomenon is consistent with IMM's grouped loss: with group size $M = 4$, each mini-batch provides only $B/M$ *independent group-level supervision signals* for backpropagation. Consequently, $B = 2048$ yields $2048/4 = 512$ effective signals, which seems to be a practical threshold for stable training in our setup. Therefore, in Fig. 2(b), we report IMM with bsz $= 2048$; the corresponding training epochs are scaled by the grouping factor, i.e., $4 \times 240 = 960$ epochs, to match the effective number of parameter updates. Others are the same as the setting for CIFAR-10.

- **sCT** and **sCT-linear**. We use the same hyperparameter setting as for CIFAR-10, since the original paper uses a U-Net in the pixel space, we cannot use the provided official configuration.

- **MeanFlow**. In time sampler, $P_{\text{mean}} = -0.4$ and $P_{\text{std}} = 1.0$. The $p_{\text{teq}} = 0.75$. We set the learning rate in training to 1e-4. The power for adaptive weighting is 1.0.

**CFG. ImageNet.** For one-step generation, our training setting with CFG follows MeanFlow, as it introduces a mixing scale $\kappa$ and defines the velocity under CFG as

$$\boldsymbol{v}^{\text{cfg}}(\boldsymbol{x}_t, t \mid c) = \omega \boldsymbol{v}_{t|0}(\boldsymbol{x}_t \mid c) + \kappa \boldsymbol{u}_{t,t}^{\text{cfg}}(\boldsymbol{x}_t \mid c) + (1 - \omega - \kappa)\boldsymbol{u}_{t,t}^{\text{cfg}}(\boldsymbol{x}_t). \tag{86}$$

This satisfies the original CFG formulation with an effective guidance scale $\kappa$. As it is proposed to bridge the instantaneous velocity and average velocity under classifier-free guidance, applying this technique directly to sCT, which models the instantaneous velocity, is not entirely straightforward. However, for sCT-linear, since we have shown its near equivalence to MeanFlow, the CFG training technique can be directly adopted. In addition, for IMM, applying CFG requires two NFEs during

inference to compute $\boldsymbol{v}^{\text{cfg}}$. As our focus is on one-step generation, *i.e.*, 1-NFE, we therefore do not include IMM in the comparisons.

In addition, we adopt the best hyperparameter configuration recommended in the official MeanFlow implementation with DiT-B/2 while our network is changed into SiT-B/2, *i.e.*, $\omega = 1.0$, $\kappa = 0.5$, class-dropout$= 0.1$ and CFG triggered if $t$ in $[0, 1]$, while keeping all other settings identical to those used for Cond. ImageNet. As an improved variant of SiT over DiT, it leads the FID50k to 6.09, better than 6.17 as reported to the original paper.

## D.2 DETAILS FOR EMPIRICAL ANALYSIS OF TABLE 2

Here, we adopt the exact same parameter setting as MeanFlow with SiT-B/2.

**sCM training techniques.** In addition, in ESC, we, following sCM (Lu & Song, 2025) and EDMv2 (Karras et al., 2024), introduce a variational weighting output head, where the output of the time embedder is passed through a linear layer to a one-dimensional scalar as the adaptive weighting function, which reads $w_{\text{adpt}}^{\psi}(t, r)$ and is then used to reweight the original loss in Eq. 8. We keep the the SiT architecture blocks untouched, while architectural improvements are orthogonal and possible. Moreover, a ratio $r_{\text{grad}} = \min\{\frac{\text{iter}}{K_{\text{grad}}}, 1\}$ for tangent warmup is implemented for mitigating some gradient spikes during training, where $K_{\text{grad}}$ is set as 10k, the same as sCT training.

$$l_{\text{esc}}(\boldsymbol{x}_t, r, t - dt, t; \theta) = \frac{e^{w_{\text{adpt}}^{\psi}(t,r)}}{D} \cdot w \cdot \left\| \boldsymbol{u}_{t,r}^{\theta}(\boldsymbol{x}_t) - \text{sg}\left( \boldsymbol{v}_{t|0} + r_{\text{grad}} \cdot (r - t)\frac{d\boldsymbol{u}_{t,r}^{\theta}(\boldsymbol{x}_t)}{dt} \right) \right\|_2^2$$
$$- w_{\text{adpt}}^{\psi}(t, r),$$

In this way, we gives the full hyper-parameter setting for Table 2, as conclude in left column of Table 5. For all the experimental trials with network architecture SiT-B/2, we trained them with Nvidia-A100×8.

## D.3 DETAILS FOR SCALING-UP EVALUATION IN SEC. 5

**CIFAR-10.** We conduct class-unconditional generation experiments on CIFAR-10. Following the official MeanFlow setting, we adopt the Adam optimizer with a learning rate of $6 \times 10^{-4}$, batch size 1024, and momentum parameters $(\beta_1, \beta_2) = (0.9, 0.999)$. We use a dropout rate of 0.2, no weight decay, and an EMA decay factor of 0.99995. Training is performed for 800k iterations, including a 10k warm-up phase. For time sampling, we draw $(r, t) \sim \text{LogNorm}(-2.0, 2.0)$, with probability 75% that $r \neq t$. The adaptive weighting exponent is set to $p = 0.75$. Data augmentation follows the protocol of Karras et al. (2022), except that vertical flipping and rotation are disabled.

Regarding our proposed improvements, we observed that variational adaptive weighting from EDM2 did not yield further gains and was therefore not adopted. Instead, we found that setting the plug-in probability $p_{\text{plug-in}} \in [0.2, 0.5]$ improved training stability, although a performance gap remained. Moreover, we set $K_{\text{fix0}} = 20$k and $K_{\text{grad}} = 10$k. All CIFAR-10 experiments with U-Net architectures were conducted on 8 Nvidia A100 GPUs.

**ImageNet-256×256.** For large-scale evaluation, we adopt SiT-XL/2 as the backbone of our improved CTSC variant, denoted as ESC. The hyperparameters follow the default configuration recommended by MeanFlow under the CFG setting, with details provided in the right column of Table 5. In practice, we find that the tangent normalization technique does not further brings performance improvements in the continuous-time shortcut model with linear path in training SiT-XL/2. All ImageNet experiments with SiT-XL/2 were trained on 16 Nvidia A100 GPUs.

## D.4 VISUALIZATION EXAMPLES FOR ESC

We provide visualization results of ESC-generated images on ImageNet-256×256 with different network architectures: SiT-B/2 (Figure 4) and SiT-XL/2 (Figure 5). All samples are generated in a single step using the same noise initialization from the latent prior and identical class labels for classifier guidance. Additional CIFAR-10 examples generated by ESC at different training epochs are shown in Figure 10.

Table 5: Configurations on ImageNet $256 \times 256$ for Table 2, (w/-cc) means 'with-class-consistent' and (w/o-cc) means 'without-class-consistent'.

| Experiment | | | | Sec. 4 | | | | | Sec. 5 | |
|---|---|---|---|---|---|---|---|---|---|---|
| Configs | MeanFlow | A1 | A2 | B1 | B2 | C | D | ESC | ESC(w/-cc) | ESC(w/o-cc) |
| **Architecture** | | | | B/2 | | | | | XL/2 | |
| params (M) | | | | 131 | | | | | 676 | |
| FLOPs (G) | | | | 23.1 | | | | | 119.0 | |
| depth | | | | 12 | | | | | 28 | |
| hidden dim | | | | 768 | | | | | 1152 | |
| heads | | | | 12 | | | | | 16 | |
| patch size | | | | 2×2 | | | | | 2×2 | |
| **Training and optimization** | | | | | | | | | | |
| epochs | | | | 240 | | | | | 240 | |
| batch size | | | | 512 | | | | | 256 | |
| dropout | | | | 0.0 | | | | | 0.0 | |
| optimizer | | | | Adam (Kingma & Ba, 2017) | | | | | Adam | |
| lr schedule | | | | constant | | | | | constant | |
| lr | | | | 0.0001 | | | | | 0.0001 | |
| Adam $(\beta_1, \beta_2)$ | | | | (0.9, 0.95) | | | | | (0.9, 0.95) | |
| weight decay | | | | 0.0 | | | | | 0.0 | |
| ema decay | | | | 0.9999 | | | | | 0.9999 | |
| **Time sampler** | | | | | | | | | | |
| $p_{\text{teq}}$ | | | | 0.75 | | | | | 0.75 | |
| $(r, t)$ sampler | | | | lognorm(-0.4, 1.0) | | | | | lognorm(-0.4, 1.0) | |
| power for adaptive weight $w$ | | | | 1.0 | | | | | 1.0 | |
| **CFG settings** | | | | | | | | | | |
| $\omega$ in Eq. 86 | | | | 1.0 | | | | | 0.2 | |
| $\kappa$ in Eq. 86 | | | | 0.5 | | | | | 0.92 | |
| cls-cond drop | | | | 0.1 | | | | | 0.1 | |
| triggered if $t$ is in | | | | [0.0,1.0] | | | | | [0.0,0.75] | |
| **ESC improvments** | | | | | | | | | | |
| $p_{\text{plug-in}}$ | 0.0 | 1.0 | 0.5 | 1.0 | 0.5 | 0.0 | 0.0 | 0.5 | 0.2 | 0.2 |
| $K_{\text{grad}}$ | 1 | 1 | 1 | 1 | 1 | 1 | 10k | 10k | 00k | 00k |
| $K_{\text{fix0}}$ | 1 | 1 | 1 | 1 | 1 | 20k | 1 | 20k | 20k | 20k |
| class-consistent batching | No | No | No | Yes | Yes | No | No | Yes | No | Yes |
| variational adaptive weighting | No | No | No | No | No | No | Yes | Yes | Yes | Yes |

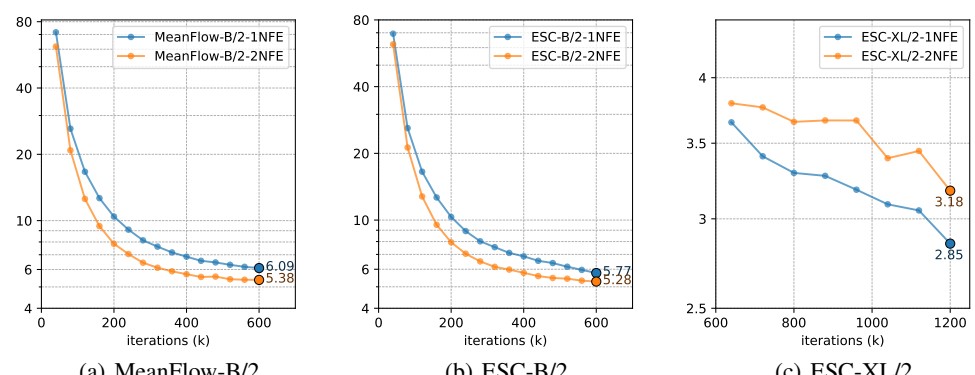

Figure 6: Comparison of FID50k under 1-NFE and under 2-NFE during training among different methods.

## D.5 ALGORITHM FOR PLUG-IN VELOCITY CALCULATION.

---

**Algorithm 2** Calculation of Plug-in Velocity

---

**Require:** Training batch $\boldsymbol{x} \in \mathbb{R}^{B \times D}$, sampled time $t$
1: Sample noise $\boldsymbol{e} \sim \mathcal{N}(0, I)$
2: Compute noised samples: $\boldsymbol{x}_t = (1 - t)\boldsymbol{x} + t\boldsymbol{e}$
3: For all sample pairs $(i, j)$ in the batch, compute

$$\boldsymbol{\varepsilon}_{i,j} = \frac{\boldsymbol{x}_{t,j} - (1-t)\boldsymbol{x}_i}{t}$$

4: Evaluate log-probabilities:

$$\log p_{i,j} = \sum_{d=1}^{D} \log \mathcal{N}(\varepsilon_{i,j,d}; 0, 1)$$

5: Compute normalized weights along index $i$:

$$w_{i,j} = \frac{\exp(\log p_{i,j})}{\sum_{i'} \exp(\log p_{i',j})}$$

6: Compute conditional velocity:

$$\boldsymbol{v}_{\mathrm{cnd},i,j} = \boldsymbol{\varepsilon}_{i,j} - \boldsymbol{x}_i$$

7: Aggregate to obtain plug-in velocity:

$$\boldsymbol{v}_{\mathrm{plug\text{-}in},j} = \sum_i w_{i,j}\, \boldsymbol{v}_{\mathrm{cnd},i,j}$$

**Ensure:** $\boldsymbol{v}_{\mathrm{plug\text{-}in}} = \{\boldsymbol{v}_{\mathrm{plug\text{-}in},j}\}_{j=1}^B$

---

### D.6 FULL COMPARISON OF ESC VS. OTHER SOTA BENCHMARKS

We further include comparisons with the current state-of-the-art diffusion and autoregressive models for completeness, as shown in Table 6 for ImageNet 256×256, and Table 7 for CIFAR10.

Table 6: Evaluation of ESC and other benchmarks under one/few-step generation on ImageNet-256×256. Underline means overall the best, while bold means the best in shortcut models.

| Family | Method | Param. | NFE | FID50k |
|---|---|---|---|---|
| GAN | BigGAN (Brock et al., 2019) | 112M | 1 | 6.95 |
| | GigaGAN (Kang et al., 2023) | 569M | 1 | 3.45 |
| | StyleGAN-XL (Karras et al., 2019) | 166M | 1 | 2.30 |
| AR/Mask | AR w/ VQGAN (Esser et al., 2021) | 227M | 1024 | 26.52 |
| | MaskGIT (Chang et al., 2022) | 227M | 8 | 6.18 |
| | VAR-d30 (Tian et al., 2024) | 2B | 10×2 | 1.92 |
| | MAR-H (Li et al., 2024) | 943M | 256×2 | 1.55 |
| Diff/ Flow | ADM (Karras et al., 2024) | 554M | 250×2 | 10.94 |
| | LDM-4-G (Rombach et al., 2021) | 400M | 250×2 | 3.60 |
| | SimDiff (Hoogeboom et al., 2023) | 2B | 512×2 | 2.77 |
| | DiT-XL/2 (Peebles & Xie, 2022) | 675M | 250×2 | 2.27 |
| | SiT-XL/2 (Ma et al., 2024) | 675M | 250×2 | 2.06 |
| | SiT-XL/2+REPA (Yu et al., 2025) | 675M | 250×2 | 1.42 |
| Shortcut | iCT (Song & Dhariwal, 2023) | 675M | 1 | 34.24 |
| | SCD (Frans et al., 2025) | 675M | 1 | 10.60 |
| | IMM (Zhou et al., 2025) | 675M | 1×2 | 7.77 |
| | MeanFlow (Geng et al., 2025a) | 676M | 1 | 3.43 |
| | | | 2 | 2.93 |
| | **ESC (w/-o-class-consist.)** | 676M | 1 | 2.92 |
| | **ESC (w/-class-consist.)** | 676M | 1 | 2.85 |
| | **ESC+ (w/-class-consist.)** | 676M | 1 | **2.53** |

Table 7: Full comparison on unconditional generation on. CIFAR-10.

| Family | method | NFE | FID |
|---|---|---|---|
| Distill | Diff-Instruct (Luo et al., 2024) | 1 | 4.53 |
| Distill | DMD (Yin et al., 2024b) | 1 | 2.66 |
| Distill | SID (Zhou et al., 2024) | 1 | **1.92** |
| Shortcut | iCT (Song & Dhariwal, 2023) | 1 | 2.83 |
| Shortcut | ECT (Geng et al., 2025b) | 1 | 3.60 |
| Shortcut | sCT (Lu & Song, 2025) | 1 | 2.97 |
| Shortcut | IMM (Zhou et al., 2025) | 1 | 3.20 |
| Shortcut | MeanFlow (Geng et al., 2025a) | 1 | 2.92 |
| Shortcut | **ESC** | 1 | 2.83 |

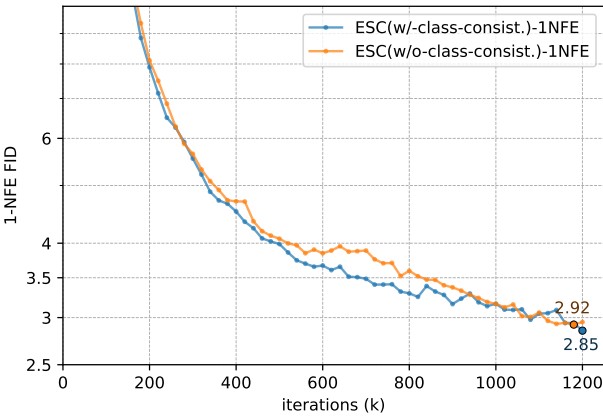

Figure 7: Convergence of FID with ESC-XL with or without class-consistent.

### D.7 DETAILS OF ESC-XL/2 CONVERGENCE WITH AND WITHOUT CLASS-CONSISTENT MINI-BATCHING

Here we give the convergence of FID with ESC-XL with or without class-consistent in the complete training process, as shown in Figure 7.

## E FURTHER ANALYSIS

### E.1 PLUG-IN VELOCITY STABILIZES THE TRAINING

To figure out whether the plug-in velocity helps to stabilize the training of shortcut models, here we give the training loss vs. iteration steps for MeanFlow and MeanFlow with Plug-in Velocity. We show the comparison of the first 200k iteration in Figure 8, where all the training setting are the same in our paper with batch size set as 512. It further illustrates that incorporating the plug-in velocity significantly stabilizes the training of MeanFlow.

### E.2 LARGE MODELS GAIN MORE PERFORMANCE FROM LOW VARIANCE TRAINING

As shown, performance improvement for SiT-XL/2 over the MeanFlow is 16.9%, while it is 5.3% for SiT-B/2 architecture. We attribute the performance gap to two key factors:

- **Optimization Dynamics.** In larger networks (e.g., XL/2), the representational capacity increases substantially, amplifying the impact of optimization stability. As shown in Figure 8, MeanFlow exhibits higher variance and less stable loss behavior during training, whereas ESC maintains stable optimization and is therefore more likely to converge to a better solution. In smaller models (e.g., B/2), the representational capacity is nearly saturated, leaving

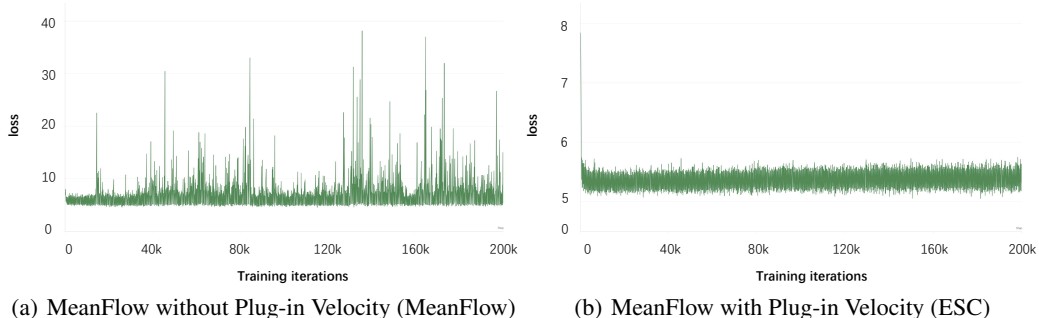

(a) MeanFlow without Plug-in Velocity (MeanFlow)   (b) MeanFlow with Plug-in Velocity (ESC)

Figure 8: Stable loss fluctuation with plug-in velocity.

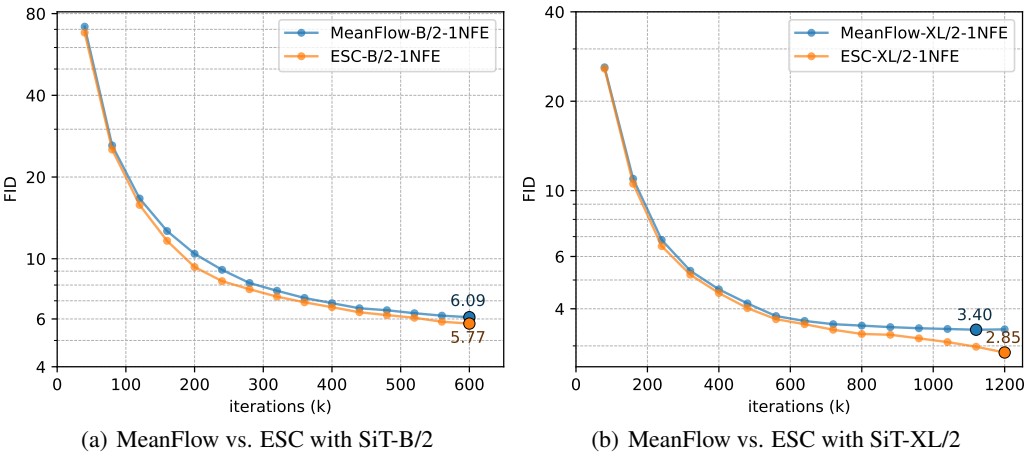

(a) MeanFlow vs. ESC with SiT-B/2   (b) MeanFlow vs. ESC with SiT-XL/2

Figure 9: Convergence of FID with different model architectures.

limited room for further improvement. In contrast, for larger models, ESC's improved stability enables it to better exploit the additional capacity, resulting in more noticeable performance gains.

- **Statistical Generalization.** As the parameter space dimensionality increases, gradient noise also grows, making variance-reduction mechanisms—such as EMA, momentum, gradient clipping, or the proposed plug-in velocity—more beneficial. This observation aligns with the theoretical intuition in Kaplan et al. (2020), where the generalization gap (or overfitting) is linked to the variance term scaling. Within the scaling law framework, bias dominates in smaller models, while variance becomes the main factor as the model scales up. To illustrate this, we compare the FID convergence curves of ESC-B/2 vs. MeanFlow-B/2 (trained for 600k iterations) and ESC-XL/2 vs. MeanFlow-XL/2 (trained for 1.2M iterations), as shown in Figure 9. Empirically, in the smaller B/2 setting, both methods converge rapidly to similar FID values. However, in the larger XL/2 model, MeanFlow's FID curve plateaus in the later training stages, while ESC continues to improve and reaches 2.85. This suggests that in large-scale models, variance dominates generalization behavior, and the variance reduction introduced by plug-in velocity significantly enhances final performance.

## F   LIMITATIONS AND FUTURE WORKS

- **Slow convergence in few-step generation.** An interesting phenomenon we observed is that, under the proposed improvements of ESC, employing two-step generation, i.e.,

$x_0 = X_{0.5,0}^\theta \circ X_{1,0.5}^\theta(x_1)$, led to slower FID convergence compared to one-step generation. This effect is particularly evident under the SiT-XL/2 architecture, whereas for B/2, the two-step scheme still achieves better performance, as shown in Fig. 6. Although MeanFlow also exhibits relatively slow convergence with two-step generation, it still outperforms one-step (2.93 vs. 3.43). One possible explanation is that, in variational adaptive weighting, predictions from 0 to 1 are inherently more difficult. With the stronger expressivity of the XL/2 architecture, the training naturally allocates higher weights to $u_{0,1}^\theta$, while the simpler sub-task $u_{0.5,0}^\theta$ receives less weight. In contrast, for the more capacity-limited B/2 architecture, fitting the easier task like $u_{0.5,0}^\theta$ proves beneficial for the overall convergence. We leave a deeper investigation of this phenomenon as future work.

- **Inflexibility in training with CFG.** We observe that introducing CFG leads to a relative improvement of $(33.05 - 6.09)/33.05 = 81.5\%$, indicating that training with CFG is essential. However, the current approach follows Eq. 86, which inevitably introduces two additional hyperparameters, $\omega$ and $\kappa$. As shown in Table 5 and in Table 4 of the original work, the optimal values of these parameters, as well as the triggered intervals, vary significantly across architectures. This greatly complicates hyperparameter tuning, and for large models such as XL/2, results in substantial computational overhead. Therefore, we argue that alternative approaches, such as *representation alignment* (Yu et al., 2025), *representation entanglement* (Wu et al., 2025), or RL-guided generation (Zheng et al., 2025), may offer promising replacements by injecting class-related semantic information into training or enabling CFG-free diffusion generation. We leave the exploration of these directions for future work.

- **Approximation for fast JVP.** Since computing JVP is required, techniques such as FlashAttention cannot be directly applied in architectures like SiT. Although this does not incur a significant time overhead, it leads to substantial memory consumption. Moreover, the computation of JVP itself is relatively expensive and introduces additional memory usage. In future work, we plan to explore numerical approximations of JVP to reduce reliance on explicit differential operators.

- **Generalization to downstream tasks and more models.** Our current work focuses purely on generative modeling. An important future direction is to extend the proposed framework to downstream tasks where generation is conditioned on additional modalities, such as text-to-image synthesis, image editing, or molecule design. Incorporating cross-modal alignment mechanisms and scalable conditioning strategies would allow the model to generalize beyond unconditional settings, making it applicable to a wider range of real-world scenarios. In particular, extending the framework to text-to-image generation represents a natural and promising step, enabling richer semantic control and practical applications. Furthermore, the proposed techniques like plug-in velocity, should be regarded as a general training technique. Since our paper includes extensive modular decomposition and performance comparisons across a wide range of methods, it is difficult to perform with/without plug-in velocity evaluations for all models under limited computational resources. We will consider extending the proposed techniques for evaluation to a broader set of models as part of our future work.

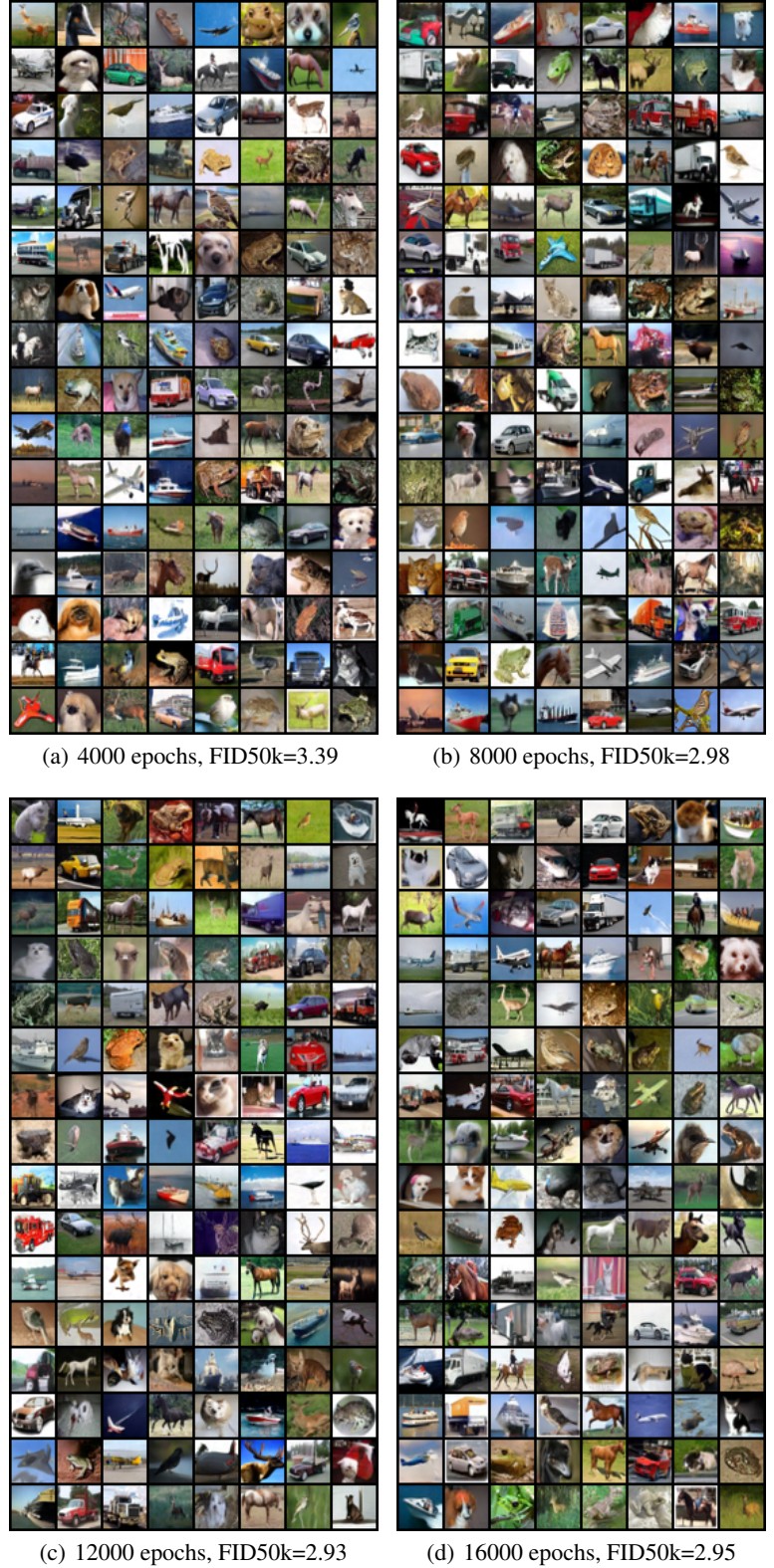

(a) 4000 epochs, FID50k=3.39

(b) 8000 epochs, FID50k=2.98

(c) 12000 epochs, FID50k=2.93

(d) 16000 epochs, FID50k=2.95

Figure 10: Images generated by ESC trained with CIFAR-10 of different epochs

