# OpenReview forum: "On the Design of One-step Diffusion via Shortcutting Flow Paths"
_ICLR.cc/2026/Conference — ICLR 2026 Poster_

### Official Review · Reviewer_k1TZ · 2025-10-30

**Soundness:** 4
**Presentation:** 3
**Contribution:** 3
**Rating:** 8
**Confidence:** 2

**Summary:**

This paper addresses the problem of designing one-step diffusion models, often called shortcut models, which can be trained efficiently from scratch. The authors argue that existing works are complex, with theoretical derivations and practical implementations being "closely coupled." This makes it difficult to understand the core design principles, compare methods, or innovate on individual components. To solve this, this paper proposes a unifying framework that reframes existing shortcut models under a single principle: approximating a two-step flow map target. They empirically and theoretically demonstrate that continuous-time shortcut models (CTSC) using a linear flow path consistently outperform discrete-time (DTSC) or cosine-path-based models. Building on this analysis, the authors propose an improved CTSC model called ESC with three key designs: plug-in velocity, class-consistent batching, and a gradual time sampler. Using these improvements, the authors train a scaled-up (SiT-XL/2) ESC model from scratch, achieving a new state-of-the-art FID of 2.85 on ImageNet-256x256 with a single function evaluation (1-NFE), without requiring any pre-training, distillation, or curriculum learning.

**Strengths:**

1. The paper's primary strength is its common design framework. The "one-step prediction vs. two-step target" (Eq. 5) is a powerful and intuitive abstraction that successfully unifies a complex and dense field of research. Table 1, which decomposes prominent models into their constituent parts, is an excellent contribution to the community.
2. The motivations are clearly stated and verified. The analysis in Section 3, which ablates flow paths (linear vs. cosine) and time sampling (discrete vs. continuous), is thorough and convincing, backed by both empirical results (Fig. 2) and theoretical analysis.
3. The final result, an FID of 2.85 (1-NFE) on ImageNet-256x256, is a significant achievement, especially given that the model is trained entirely from scratch. This result challenges the long-held belief that one-step models require costly distillation from a pre-trained, multi-step teacher.

**Weaknesses:**

This is a strong paper without major technical weaknesses. Please see my questions below.

**Questions:**

1. Does the trained model actually satisfy the flow consistency property $X_{0.5,0}^{\theta}(X_{1,0.5}^{\theta}(x_1)) \approx X_{1,0}^{\theta}(x_1)$? The 2-NFE failure suggests it may not (Figure 6). Could this imply the model has simply memorized a single shortcut rather than learning the underlying flow field?
2. Following the previous question, I'd like to know if ESC can be used in a multi-step fashion. If yes, are there any results or visualizations?
2. Is ESC applicable to video DiT models? Are there any results?

---

> ### Author Response · Authors · 2025-11-17
> **Response to Questions**
>
> We really thank you for your recognition of the value of our work, including  theoretical justification, empirical analysis and technical improvements.
> ## **Response to Questions**
> ### 1&2.  Response to *"The 2-NFE failure suggests it may not (Figure 6). Could this imply the model has simply memorized a single shortcut rather than learning the underlying flow field?"* and "I'd like to know if ESC can be used in a multi-step fashion.":
> As discussed in our future work, in our subsequent research, we further investigated this issue. We found that models trained with our proposed technique quickly learn effective one-step generation; however, achieving high-fidelity few-step generation requires additional training iterations to allow the model to better capture the full probabilistic trajectory. To verify this, we extended the training from 1.2M to 2.4M and 3.6M iterations. As shown in the table below (also included in *Table 8 in Appendix D.8 in the revised manuscript*), the results demonstrate that longer training substantially improves few-step generation performance:
>
>  | Iterations | 1-NFE | 2-NFE |
>  |-|--|-|
>  | 1.2M | 2.85 | 3.18 |
>  | 2.4M | 2.65 | 2.47 |
>  | 3.6M | 2.58 | 2.34 |
>
> These results confirm that increasing the number of training iterations enables the model to progressively learn the underlying probabilistic path, leading to significantly better generation quality for few-step sampling.
>
> ### 3. Response to *"Is ESC applicable to video DiT models?"*:
> This is indeed a very interesting question. Currently, most video generation research remains focused on distillation-based models, and we attribute it mainly to two practical reasons:
> - (1) High computational cost. Discrete-time shortcut models tend to perform poorly when trained from scratch, while continuous-time models require computing Jacobian–Vector Products (JVPs) during training (as discussed in our Limitations and Future Work section). This operation is extremely memory-intensive, making it difficult to train large networks while performing JVPs simultaneously under typical academic GPU constraints.
> - (2) Training iterations. It is well known that larger models require more iterations to converge. Shortcut models usually require more training iterations. For instance, compared with DiT, which typically trains for about 400K iterations, MeanFlow requires around 1.2M iterations when trained from scratch under DiT-XL/2 architecture. Such computational and time costs are often prohibitive in video generation settings.
> Nevertheless, we have observed a growing interest in few-step generation for video models. For example, rCM-WAN [1] leverages a Triton-based attention kernel, which substantially accelerates CTSC training under the distillation setting.
> We believe that with further engineering optimizations and advances in shortcut model design, large-scale applications in shortcut models like ESC, such as text-to-image and video generation will become increasingly feasible in the near future.
>
> [1]Zheng, Kaiwen, et. al, Large Scale Diffusion Distillation via Score-Regularized Continuous-Time Consistency, arXiv:2510.08431.

---

> ### Comment · Reviewer_k1TZ · 2025-11-26
>
> My concerns have been addressed. Thanks for the detailed response. The score remains the same.

---

> > ### Author Response · Authors · 2025-11-27
> > **Thanks**
> >
> > Thanks for your recognition of our works! Your insightful and helpful suggestions help us to improve our works a lot. Your support serves as a driving force for us to continue our efforts in this field and contribute to its advancement for the community.

---

### Official Review · Reviewer_rWuT · 2025-11-01

**Soundness:** 4
**Presentation:** 3
**Contribution:** 4
**Rating:** 8
**Confidence:** 2

**Summary:**

This submission summarizes and organizing algorithmic design space of one-step diffusion models including consistency training, shortcut diffusion models, and mean flows.
Although differences in derivation and wordings of the methods, they share computational framework of training by aligning one-step predictions toward targets that are acquired by two-step computation, which can be viewed as learning "shortcut".
The manuscript includes theoretical analyses claim that 1) both two-step DTSC and CTSC have bounded errors w.r.t. Lipschitz constants of the velocity field, and 2) inference error of  shortcut models measured Wasserstein-2 distance are bounded using bias and variance defined using targets and losses.
Following the insights,  improved training method Explicit&easier Shortcut Model (ESC) is proposed. It uses techniques from the summarized literatures and new plug-in velocity, which aggregates in-minibatch velocity.
Experiments using CIFAR and ImageNet shows that ESC can make improvements over MeanFlow slightly (-0.1 -- -0.3 in FID).

**Strengths:**

- One-step diffusion models are a hot topic in 2025 after Shortcut diffusion and Mean flow. This submission is a timely and nice follow-up to help understanding.
- Theoretical analyses are deep and well motivated for understanding shortcut behavior of continuous and discrete models. (although I could not check all of the proofs in the near-30-page appendix.)
- In the method part, plug-in velocity (Algorithm 1) is novel and shown to be useful in the experiments.
- The overall paper are well organized.

**Weaknesses:**

- Practical impact: surely the SOTA results in FID are achieved but the FID gains provided by ESC may be marginal. I think that the differences of FID ranging within 0.1 -- -0.3 hardly impacts on human opinions on the quality of images.

- Experimental/theoretical supports of design choices are relatively weak: Table 2 shows the design choice picked from the design space, but I think these selections looks empirical and I could not grasp how the theoretical results are exploited. Please point if I missed some parts.

**Questions:**

- In algorithm 1, some lines are too much pythonic to interrupt non-programmer readers. for example, x[:,None,:] and logp_fn = Normal(0, 1).log_prob (function treated as an object) may be replaced if possible.

---

> ### Author Response · Authors · 2025-11-17
> **Response to Weaknesses and Questions**
>
> We really thank you for your recognition of the value of our work, including theoretical justification, empirical analysis and technical improvements.
>
> ## **Response to Weaknesses**
> ### 1. Response to *“I think that the differences of FID ranging within 0.1 -- -0.3 hardly impacts on human opinions on the quality of images”*:
>   In fact, we observe that the FID improvement from 3.43 (MeanFlow) to 2.85 under XL/2 architecture represents a gap of more than 0.5 absolutely and over 15% relatively, which is a substantial difference. However, providing qualitative visualizations is challenging, since it is difficult to ensure that both models generate the images of the same semantic for a fair comparison, thus offering limited visual insight.
>   We therefore commit to releasing the training checkpoints for both MeanFlow and ESC in our future release, enabling the community to perform direct and fair comparisons.
> ### 2. Response to *"Experimental/theoretical supports of design choices are relatively weak."*:
> Here, for clarifying the connection of experimental and theoretical supports of the design choice, we gives some representative examples of  how theoretical results are exploited by the establishment of experimental improvements.
>   - In Table 2, we use MeanFlow-based framework as the starting point, since Section 3 theoretically demonstrates the superiority of the linear path (*Theorem C.7*) and continuous-time shortcut model (*Theorem 3.1*), which in combination, results in MeanFlow.
>   - The optimal velocity proposed in *Theorem 4.1* is introduced to reduce the training variance as provided in Theorem 3.1. Instead, plug-in velocity is theoretically justified through its bias–variance trade-off  in *Theorem 4.1*, explaining its stabilizing effect during training.
>
>   If there are still questions regarding the connection between our theoretical analysis and the proposed methods, we would be happy to discuss them further.
>
> ## **Response to Questions**
> ### 1. Response to *"In algorithm 1, some lines are too much pythonic to interrupt non-programmer readers."*：
> We have added a standard algorithm format (instead of the previous Python-style version) in *Appendix D.5 in the revised manuscript* in the newly updated pdf. We sincerely appreciate your valuable suggestion, and we plan to include this content in the main text or appendix in the revised version.

---

### Official Review · Reviewer_YqvP · 2025-11-04

**Soundness:** 2
**Presentation:** 3
**Contribution:** 2
**Rating:** 4
**Confidence:** 4

**Summary:**

This paper proposes a unified framework for ODE based one-step diffusion models trained from scratch. The authors systematically analyze existing methods (CT, SCD, IMM, sCT, MeanFlow) by decomposing them into modular components and studying their design choices.
Building on this analysis, they introduce ESC (Explicit & Easier Shortcut model), which incorporates several technical improvements including plug-in velocity, gradual time sampling, and adaptive loss weighting. ESC achieves improved FID50k of 2.85 on ImageNet-256×256 with one-step generation, without requiring pre-training, distillation, or curriculum learning.

**Strengths:**

- The paper is clearly written and easy to follow.
- While the work does not introduce a fundamentally new method and offers only moderate novelty, it contributes a valuable unifying perspective on ODE-based one-step diffusion models.
- The theoretical contributions are strong: Theorem 2.2 establishes error bounds for both DTSC and CTSC; Proposition 3.1 presents an insightful bias–variance analysis clarifying when CTSC outperforms DTSC; and Theorem C.7 theoretically justifies why linear paths are optimal for shortcut models under Fisher information metrics.
- It provides comprehensive ablations and analyses that thoroughly examine the effectiveness of the proposed techniques, demonstrating overall high quality.

**Weaknesses:**

- Limited Novelty of Core Framework: While the unified view is valuable for understanding, this is already well established in previous work Flow Map Matching[1,2], which further hurt the contribution of "propose a common design framework for representative shortcut models".
    -from discrete to continuous, one can either set $s= t−dt$ to get backward formula such as MeanFlow and sCM; or set $s=r+dt$ to obtain the forward formula such as AlighYourFlow [3].
- Empirical Gaps:
    - Slow convergence in 2-NFE generation (Fig. 6, Section E) suggests the improvements may be overfitted to 1-NFE
    - Limited comparison with recent distillation-based methods that achieve better FID scores [4,5,6]
    - The improvement compared to the baseline is nevertheless incremental.
- Technical Concerns: The time sampler design (gradual transition from sCT to MeanFlow) and the choice of $p_{plugin}$ appears heuristic without principled justification

[1] Nicholas Matthew Boffi, Michael Samuel Albergo, and Eric Vanden-Eijnden. Flow map matching with stochastic interpolants: A mathematical framework for consistency models. Transactions on Machine Learning Research, 2025.

[2] Qiang Liu. Icml tutorial on the blessing of flow. International conference on machine learning,
2025.

[3] Amirmojtaba Sabour, Sanja Fidler, and Karsten Kreis. Align your flow: Scaling continuous-time flow map distillation. arXiv preprint arXiv:2506.14603, 2025.

[4] Tianwei Yin, Micha¨ el Gharbi, Richard Zhang, Eli Shechtman, Fredo Durand, William T Freeman, and Taesung Park. One-step diffusion with distribution matching distillation. In Proceedings of the IEEE/CVF Conference on Computer Vision and Pattern Recognition, pages 6613–6623, 2024.

[5] Mingyuan Zhou, Huangjie Zheng, Zhendong Wang, Mingzhang Yin, and Hai Huang. Score identity distillation: Exponentially fast distillation of pretrained diffusion models for one-step generation. In Forty-first International Conference on Machine Learning, 2024.

[6] Weijian Luo, Tianyang Hu, Shifeng Zhang, Jiacheng Sun, Zhenguo Li, and Zhihua Zhang. Diff-instruct: A universal approach for transferring knowledge from pre-trained diffusion models. Advances in Neural Information Processing Systems, 36:76525–76546, 2023.

**Questions:**

- how to interpret IMM as a special case in the proposed framework? e.g. build connection between eq(5) and eq(50)
- In fig.1, it shows that practical prediction matches practical target, which is unrealistic. In addition, in fig 1 (c)(e), we have vector addition $v_{t|0}+u_{t|r}^\theta=(r-t) d u_{t|r}^\theta / dt$, which is inconsistent with eq(8).
- In sec 2.4 Q3, the authors claim that "(s)CM benefits from distillation by learning from a pretrained velocity field", however, in sCM, "We always initialize the CM from the EMA parameters of the teacher diffusion model. For sCD, we always use the $F_{pretrain}$ of the teacher diffusion model with its EMA parameters during distillation." That's to say, for sCT the teacher model is used as initialization for the one-step generator, which is critical.
    - Given that the authors admit that at least a pre-trained model is important for better few-step performance as answer to this Q3, and that it's impractical to train large-scale text-to-image model from scratch, the authors failed to justify why in this work they insist working on training from scratch.

-------- below could be biased (does not affect the rating) -------
- focus of the paper. while aims to "disentangles concrete component-level choices, thereby enabling systematic identification of improvements", the empirical results are mainly focus on improved training techniques for MeanFlow baseline.
- the authors show in Fig 1 visually and with prop 3.1 that the challenge of constructing flow map targets and the consequence during inference, however, this challenge is not directly addressed in the paper.
The training technique/design are proposed to enhance training stability, while no empirical results can validate the training instability of the baseline.

---

> ### Author Response · Authors · 2025-11-17
> **Response to Weaknesses (I)**
>
> We sincerely thank the reviewer for their valuable feedback. We have addressed the identified issues and updated the manuscript accordingly. In addition, we have included further clarifications and analyses in red to the Appendix in the newly updated revised manuscript.
> ## **Response to Weaknesses**
> ### 1. Response to *"Limited Novelty of Core Framework"*:
>   We do not believe that the merely adoption the flow map notation diminishes the novelty of our work. We are the first to unify previous approaches under a common conceptual framework, namely, one-step flow map prediction approximating two-step flow map targets, which has not been explicitly discussed in either [1] or [2]. We propose that by setting $s = t - dt$, it naturally bridges the discrete-time shortcut model to continuous-time ones like MeanFlow and sCT with a mathematical formulation, and further decomposes and analyzes the role of each component. These aspects have not been explored in prior works.
>   Moreover, the concurrent work [3], which is published in June 2025, adopts a setting of $s = r + dt$, can also be seamlessly incorporated into our proposed framework. This in fact demonstrates the generality of our formulation, showing that even concurrent methods naturally fit into the structure we propose.
>
> ### 2. Response to *"Empirical Gaps"*:
>   - (1) **Explanation for Slow convergence in 2-NFE generation**: In our subsequent research, we further investigated this issue. We found that models trained with our proposed technique quickly learn effective one-step generation; however, achieving high-fidelity few-step generation requires additional training iterations to allow the model to better capture the full probabilistic trajectory. To verify this, we extended the training from 1.2M to 2.4M and 3.6M iterations. As shown in the table below  (also included in *Table 8 in Appendix D.8 in the revised manuscript*), the results demonstrate that longer training substantially improves few-step generation performance:
>  | Iterations | 1-NFE | 2-NFE |
>  |-|--|-|
>  | 1.2M | 2.85 | 3.18 |
>  | 2.4M | 2.65 | 2.47 |
>  | 3.6M | 2.58 | 2.34 |
>
>   - These results confirm that increasing the number of training iterations enables the model to progressively learn the underlying probabilistic path, leading to significantly better generation quality for few-step sampling.
>
> - (2) **Response to "Limited comparison with recent distillation-based methods that achieve better FID scores [4,5,6]"**. Given that our method is trained from scratch, direct comparisons with distillation-based models would be somewhat unfair. Due to space limitations in the main paper, we therefore did not include those results in the main text. However, we have added a comprehensive comparison table (*Table 7 in Appendix D.6 in the revised manuscript*), which includes results from [4, 5, 6] in CIFAR, and more models in ImageNet 256 X 256 *Table 6 in Appendix D.6 in the revised manuscript*. These tables are also provided below.  If you have any additional methods that you would recommend for comparison, we would greatly appreciate your suggestions.
>
> (ImageNet 256x256)
> | Family     | Method                                | Param. | NFE        | FID50k |
> |------------|-----------|--------|------------|--------|
> | **GAN**    | BigGAN            | 112M   | 1          | 6.95   |
> |            | GigaGAN         | 569M   | 1          | 3.45   |
> |            | StyleGAN-XL     | 166M   | 1          | 2.30   |
> | **AR/Mask**| AR w/ VQGAN       | 227M   | 1024       | 26.52  |
> |            | MaskGIT        | 227M   | 8          | 6.18   |
> |            | VAR-d30         | 2B     | 10×2       | 1.92   |
> |            | MAR-H              | 943M   | 256×2      | 1.55   |
> | **Diff/Flow**| ADM   | 554M   | 250×2      | 10.94  |
> |            | LDM-4-G        | 400M   | 250×2      | 3.60   |
> |            | SimDiff     | 2B     | 512×2      | 2.77   |
> |            | DiT-XL/2       | 675M   | 250×2      | 2.27   |
> |            | SiT-XL/2             | 675M   | 250×2      | 2.06   |
> |            | SiT-XL/2+REPA        | 675M   | 250×2      | **1.42** |
> | **Shortcut**| iCT     | 675M   | 1          | 34.24  |
> |            | SCD        | 675M   | 1          | 10.60  |
> |            | IMM         | 675M   | 1×2        | 7.77   |
> |            | MeanFlow       | 676M   | 2          | 2.93   |
> |            | ESC (w/o-class-consist.)               | 676M   | 1          | 2.92   |
> |            | ESC (w/-class-consist.)                | 676M   | 1          | *2.85*   |

---

> ### Author Response · Authors · 2025-11-17
> **Response to Weaknesses (II)**
>
> (CIFAR-10)
> | Family     | Method                                 | NFE | FID  |
> |------------|-----------------------------------------|-----|------|
> | **Distill**| Diff-Instruct (Luo et al., 2024)        | 1   | 4.53 |
> |            | DMD (Yin et al., 2024b)                 | 1   | 2.66 |
> |            | SID (Zhou et al., 2024)                 | 1   | **1.92** |
> | **Shortcut**| iCT         | 1   | 2.83 |
> |            | ECT           | 1   | 3.60 |
> |            | sCT                | 1   | 2.97 |
> |            | IMM          | 1   | 3.20 |
> |            | MeanFlow         | 1   | 2.92 |
> |            | ESC                                      | 1   | *2.83* |
>
> - (3) Response to **The improvement compared to the baseline is nevertheless incremental.**: As discussed in the Introduction, our contributions go well beyond isolated technical enhancements. We propose a general design framework, systematically analyze the design space and scaling behaviors, and introduce practical improvement techniques that achieve over 15% performance gains, which is not incremental but significant. Collectively, these contributions establish both conceptual and empirical advances for the field.
> ### 3. Response to *Heuristic Techniques without Principled Justification*.
>   First, we respectively disagree with the reviewer's comment about heuristic, as **the comment of "heuristic" has no objective or universally agreed-upon boundary in the context of generative modeling**. In practice, what is labeled as a "heuristic" often reflects a subjective judgment rather than a principled distinction. Under any reasonable interpretation, many widely adopted and highly influential techniques in the field, such as the training strategies in sCM and iCM, are equally heuristic, being derived from empirical observations that successfully stabilize CM training. The same applies to the augmentation mechanisms, log-normal time sampler in EDM and the Masked Pretraining paradigm in MAE, which are purely empirical design choices yet demonstrably lead to substantial improvements. Therefore, dismissing our method solely on the basis of being "heuristic" does not constitute a consistent criterion, especially when comparable or even more heuristic components are broadly accepted in current state-of-the-art models.
>
>   Second, the reviewer's claim of **being without principled justification is not accurate**. For example, regarding the plug-in velocity, we provide a clear theoretical foundation: we derive it from the error bound analysis, which motivates using velocity as supervision and emphasizes the critical role of variance in training stability. From this, we theoretically establish the variance–bias trade-off, leading naturally to the proposed plug-in velocity as solution.  Hence, our method is not only empirically validated but also theoretically grounded. We would sincerely appreciate it if the reviewer could kindly clarify which parts may benefit from further justification, enabling us to respond more accurately in our revision.

---

> ### Author Response · Authors · 2025-11-17
> **Response to Questions**
>
> ## **Response to Questions**
> ### 1. Response to "How to interpret IMM as a special case in the proposed framework?":
>   We provide a  proof in *Page 21, Line 1116 - 1149 in the Appendix of the revised manuscript*, showing that the Eq. (5) is bounded by IMM loss in Eq. (50) in the original submission (corresponding to Eq. (52) in the revised version) in terms of optimization objectives.
>   In brief, the group kernel loss used in IMM is equivalent to the Maximum Mean Discrepancy (MMD) between distributions as described in Appendix B.3 of the original paper of IMM. By applying Jensen’s inequality, we show that the loss in Eq. (5) serves as an upper bound of the IMM loss. Therefore, minimizing the Flow Map Loss implicitly minimizes the IMM loss. Note that for clarity, we replaced the original notation $d(\cdot, \cdot)$ with $\mathrm{ker}(\cdot, \cdot)$ to explicitly denote the RBF kernel function, and then redefined $d(\cdot, \cdot)$ accordingly.
>
> ### 2. Response to confusion from Figure 1:
>   We would like to clarify that the intention of this figure is to illustrate that the practical prediction, even when it matches the practical target, still deviates from the ideal target. We apologize for the confusion caused by the previous version. In the newly updated version, we have added additional arrows to make this relationship clearer and avoid potential misunderstanding. Furthermore, we also identified directional errors in the arrows of continuous-time shortcut models of Figures 2(c) and 2(e). These have been corrected in the updated figures in the revised manuscript. We sincerely appreciate the reviewer's careful observation and feedback.
> ### 3. Response to sCT's training and motivation for training from scratch:
> - (1) **sCT uses a teacher model.**
>   - In Section 2 of sCM, it states: “Consistency training (CT), by contrast, trains CMs from scratch without the need for pretrained diffusion models, which establishes CMs as a standalone family of generative models in their own right.” Besides, according to the official CM codebase, during CT training, there is no pretrained model involved. Therefore, by analogy, it is natural to infer that sCT does not initialize from a teacher model can also achieve a competitive performance. Furthermore, although its initialization makes use of a pretrained diffusion model, our goal here is to  *explore the design space of shortcut diffusion models trained from scratch*. Therefore, we decompose its modules and evaluate them under a unified train-from-scratch protocol, which ensures a fair comparison and analysis. To further make it clearer, we added some notes in *Line 312 in Section 3 in the revised manuscript*.
> - (2) **Why insisting working on training from scratch.**
>   - As stated in *Line 34*, distillation-based models rely on a pretrained diffusion model and therefore require a costly two-stage training process, where the quality of the teacher directly limits the performance of the distilled one-step model. In contrast, as noted in *Line 43*, train-from-scratch models have recently demonstrated strong efficiency and effectiveness, drawing increasing attention from the community. Thus, introducing a one-step diffusion framework trained entirely from scratch offers both conceptual value and practical benefits. **These considerations as motivation are discussed in detail in the Introduction section.**
>
>   - Then, let's discuss the current performance and applicability gap between distillation-based and train-from-scratch approaches. We argue that **research progress in train-from-scratch and distillation-based one-step diffusion models both have significant value**. If, as suggested, the value of such models were to be judged solely by their current performance compared to distillation-based models, one might also question the value of earlier one-step models such as MeanFlow and sCT. More broadly, in 2020, diffusion models initially underperformed GANs and required far longer inference time, yet continued research ultimately led to the breakthroughs that drive today's generative AI progress.
> ### 4. Response to "The empirical results are mainly focus on improved training techniques for MeanFlow baseline.":
>   We have identified the relevant component combinations in Section 3, namely: (1) the linear path and (2) the continuous-time model. Therefore, it is reasonable to build further improvements upon the optimal configuration, i.e., the MeanFlow framework in Section 4. Our detailed explanation is provided in **Point.3 in General Response.**
> ### 5. Response to "No empirical results can validate the training instability of the baseline.":
>   To further illustrate, we provide *Figure 8 in Appendix E.1 in the revised manuscript*  illustrates that incorporating the plug-in velocity stabilizes the training of MeanFlow when trained with a batch size of 512. Moreover, *Figure 9 in Appendix E.2* shows that  ESC consistently outperforms MeanFlow during training according to sample FID.

---

> > ### Comment · Reviewer_YqvP · 2025-11-25
> >
> > I appreciate the authors' effort for the rebuttal, and thank you for the explanation on the IMM part.
> >
> > However, I still want to address several points:
> >
> > 1. It's quite clear, in eq(3.8) in [1] that the idea of "one-step flow map prediction approximating two-step flow map targets" is proposed already (eq (3.6) in the first version on arXiv from 11/June/2024). Similarly the same idea can be find in [2]. I really hope the authors could respect other peoples work, this will not diminish your contribution.
> > > namely, one-step flow map prediction approximating two-step flow map targets, which has not been explicitly discussed in either [1] or [2].
> >
> > 2. in response to Q3 (1), sCM and CM are different things, you can not use CM to argue for sCM. No to mention that I have already quoted you the part stated that sCM always use an initialization.
> >
> > The rebuttal process is not about attacking each other, but helping to shape a better, clearer version of the paper?
> >
> > You made a good paper, but you are also responsible for the readers to make sure the contribution is clear (where 'novelty' is one un-necessary contribution'), and potentially helpful for other researchers in the community.
> >
> > [1] Nicholas Matthew Boffi, Michael Samuel Albergo, and Eric Vanden-Eijnden. Flow map matching with stochastic interpolants: A mathematical framework for consistency models. Transactions on Machine Learning Research, 2025.
> > [2] Qiang Liu. Icml tutorial on the blessing of flow. International conference on machine learning, 2025.
> >
> > I am VERY open to raise my score, if the author could address my key concerns above.

---

> ### Author Response · Authors · 2025-11-25
> **Response to the two concerns (i)**
>
> First and foremost, we would like to clarify that at no point have we regarded the reviewer’s comments on our work as an "attack." Rather, we also have no intention of attacking back, and have always considered them to be extremely valuable feedback. In our responses we have repeatedly emphasized this. For example, your comments help to locate the arrow error in one of our figures, for which we are sincerely grateful as
> > *"We sincerely appreciate the reviewer's careful observation and feedback."*.
>
> Accordingly, we hope that our replies do not contain any aggressive or confrontational tone.
> If any part of our response comes across as too direct or even slightly confrontational, we respectfully ask the reviewer to let us know. We will take extra care of our wording in our future responses.
>
> Here is our response to your concerns:
> ### 1. Response to *"In eq(3.8) in [1] that the idea of "one-step flow map prediction approximating two-step flow map targets" is proposed already."*
>
>  -  Firstly, there may be some  different perspectives regarding our response, and we are sorry that our response brings such confusion. The original sentence in our response is
> > "unifying previous approaches under a common conceptual framework, namely, one-step flow map prediction approximating two-step flow map targets, which has not been explicitly discussed in either [1] or [2],"
>
>     with the emphasis on **unifying previous approaches under a common conceptual framework**, rather than **proposing an entirely new framework**.
>
> -  Secondly, in our understanding, Eq. (3.8) in [1] corresponds to Eq. (4) in our paper, which describes the consistency property of the flow map rather than what we define as unification as we proposed in Eq.(5). We have acknowledged that the initial idea originates from consistency property, and accordingly we cite and emphasize this both in main body and in the Appendix of the original submission. **In this way, we want to clarify that the original submission has shown our respect for prior work.**
>
> - Thirdly, we would like to clarify our contribution on top of and grounded in the foundation laid by [1]. **For our Eq.(5), we consider this to be what we define the unifying framework, which captures the idea that "the one‐step flow map prediction approximates the two‐step flow map targets"**. Although [1] mentions some flow map loss (for example in Algorithm 4: "one‐step flow map prediction approximates multi‐step flow map"), its $d(·,·)$ is specific and it primarily deals with discrete‐time shortcut models. In addition, it discusses continuous‐time distillation models, such as Lagrangian and Eulerian distillation losses, but not from the perspective of "one‐step flow map prediction approximates two‐step flow map" as we develop. Therefore, the claimed novelty of a unifying framework is based on starting with the consistency property (Eq.4), then introducing a general loss(Eq.5), and subsuming discrete-time models into that framework (Table 1: CT,SCD, IMM), extending to continuous-time models under $s \to t$ (Table 1: sCT, MeanFlow). In summary, we do not deny the contributions of earlier work, and that our work is based on them (as we cite [1] multi-times), and want to clarify that the novelty is also what we have contribute over these foundations.
>
> - Moreover, we agree with what you've mentioned,
>   > "where 'novelty' is one un-necessary contribution".
>
>     Actually, in the main body of the paper, we did not even claim that our  proposed frame and summarization is novel in Line. 51-55, since the novelty is a minor contribution, we focus more on practical perspective, as it writes
>   > "We first contribute to proposing a common design framework for these shortcut models from a practical standpoint."
>
> - Finally, to clearly demonstrate that our work is enlightened by and builds upon the prior literature, we have **added multiple additional citations to [1] and [2]** throughout the revised manuscript, so that our paper could be "potentially helpful for other researchers in the community".
>
> We hope this response could eliminate your concerns on the first point.

---

> ### Author Response · Authors · 2025-11-26
> **Response to the two concerns (ii)**
>
> ### 2. Response to *"sCM and CM are different things, you can not use CM to argue for sCM."*
>
> - Likewise, we do not deny your raised question that sCT was initialized from a teacher model. We want to clarify that, in our paper, we include sCT as "train-from-scratch" and evaluate it under "train-from-scratch" protocol for fair comparison, because it can serve as an effective “train-from-scratch” shortcut model is based on three observations:
>     - (1) CT approach is effective;
>     - (2) MeanFlow approach is effective;
>     - (3) sCT is the continuous-time version of CT, and sCT and MeanFlow differ only in the choice of Flow Path and end-time sampler.
>
>      This underpins our claim in the rebuttal that
>      > "it is natural to infer that sCT, when not initialized from a teacher model, can also achieve competitive performance."
>
>     In this way, we included a consistent protocol for evaluation of the train-from-scratch version for fair comparison.
>
> - Moreover, we noted the concern you raised, so, to avoid misleading readers, in the previously uploaded version we added (Lines 312-313):
>  >"While sCT is originally initialized from the teacher diffusion model, we train all the discussed models from scratch, for a fair comparison."
>
>    As we mentioned in our rebuttal:
>   > “To further make it clearer, we added some notes in Line 312 in Section 3 in the revised manuscript.”
>
>     We hope that this refinement both maintains the integrity of the paper and appropriately honors the practical implementation of the original literature, and all these revisions including the revised figures and extra notations were made after carefully considering and fully acknowledging your suggestions.
>
> Finally, we sincerely thank you again for your responses. We do not reply with any adversarial intent as "attack". We also very much appreciate your openness to raising the score and your recognition of our work, as well as the time and effort you have devoted to helping improve this manuscript.
>
> Aside from the two issues mentioned above, we may still have different perspectives on certain points. We wonder whether there remain any other concerns that have not been fully addressed. If you have any questions or uncertainties about any of our responses, please do not hesitate to raise them immediately. We are ready to resolve them.

---

> > ### Comment · Reviewer_YqvP · 2025-11-27
> >
> > Thank you for the careful and detailed clarifications.
> >
> > With these clarifications and revisions, I consider my major concerns addressed. I am updating my score to 6.

---

### Official Review · Reviewer_enqR · 2025-11-05

**Soundness:** 3
**Presentation:** 3
**Contribution:** 3
**Rating:** 4
**Confidence:** 4

**Summary:**

This paper proposes a unified framework for various shortcut model methods and provides both theoretical and empirical analyses of the advantages and disadvantages of different designs. Based on these analyses, several improvements are introduced, such as plug-in velocity, gradual time samplers, and class-consistent mini-batching, collectively referred to as the explicit&easier shortcut modeling method, which achieves state-of-the-art performance in one-step ImageNet-256×256 generation.

**Strengths:**

- This paper provides a comprehensive framework for a series of shortcut model methods, offering effective tools for analyzing this family of approaches.
- The analysis of each component is detailed and well-structured. By disentangling the individual components, the paper makes the design space considerably more transparent. Theoretical analyses are extensive and appear to be well-structured.
- The proposed method achieves SOTA results on one-step ImageNet-256×256 generation, demonstrating the effectiveness of the improvements.
- This paper releases detailed code, ensuring reproducibility.

**Weaknesses:**

- According to Figure 3, it is difficult to claim that “the convergence of FID50k during training is substantially faster with the class-consistent mini-batching technique.” More evidence is needed to support this statement.
- As shown in Figure 6, at the XL scale, the proposed method achieves worse FID under 2-NFE compared to 1-NFE. This might indicate a potential scalability issue of the proposed approach.
- There are a few minor typo errors. For example, in line 198, $X^\theta_{t,r}(xt)$ should be $X^\theta_{t,r}(x_t)$; in line 263, $l_{scm}$ seems to refer to $l_{sct}$ mentioned earlier.

**Questions:**

- The results in Tables 2 and 3 suggest that the proposed method brings more improvements with the large-scale network architecture than with the basic one. What causes this difference?
- In Figure 3(b), several comparison curves included in Figure 3(a) are missing. What is the reason for this omission? The same question applies to Figure 3(c).

---

> ### Author Response · Authors · 2025-11-17
> **Response to Weaknesses**
>
> We sincerely thank the reviewer for recognizing the contributions of our work, including the proposed framework, theoretical analysis, and the improvements that lead to state-of-the-art performance. We have added several additional figures and analysis in red to the Appendix in the newly updated revised manuscript for your reference. Below, we address each of your concerns point by point.
>
> ## **Response to Weaknesses**
> ### 1. Response to "More evidence is needed to support this statement that the convergence of FID50k during training is substantially faster with the class-consistent mini-batching technique.":
>   Due to space limitations, we only presented the convergence curves near the end of training in the main text. To further address your concern, we have now included the full convergence trajectories as additional evidence. Specifically, *Figure 7 in Appendix D.7 in the revised manuscript* compares the FID convergence throughout the entire training process, with and without class-consistent mini-batching, demonstrating the correctness of our claim.
> ### 2. Response to "The proposed method achieves worse FID under 2-NFE compared to 1-NFE":
>   As discussed in our future work, in our subsequent research, we will further investigate this issue. We found that models trained with our proposed technique quickly learn effective one-step generation; however, achieving high-fidelity few-step generation requires additional training iterations to allow the model to better capture the full probabilistic trajectory. To verify this, we extended the training from 1.2M to 2.4M and 3.6M iterations. As shown in the table below  (also included in *Table 8 in Appendix D.8 in the revised manuscript*), the results demonstrate that longer training substantially improves few-step generation performance:
>
>  | Iterations | 1-NFE | 2-NFE |
>  |------|-------|-------|
>  | 1.2M | 2.85 | 3.18 |
>  | 2.4M | 2.65 | 2.47 |
>  | 3.6M | 2.58 | 2.34 |
>
>  These results confirm that increasing the number of training iterations enables the model to progressively learn the underlying probabilistic path, leading to significantly better generation quality for few-step sampling.
> ### 3. Response to typo:
> Thank you for pointing out typo, and we have fixed them in the revised manuscript.

---

> ### Author Response · Authors · 2025-11-17
> **Response to Questions**
>
> ## **Response to Questions**
> ### 1. Response to *"The proposed method brings more improvements with the large-scale network architecture than with the basic one. What causes this difference?"*:
>   We appreciate the reviewer's insightful question and agree that this is a valuable direction for further exploration. We have include a more detailed discussion of this topic in the full version of the paper in *Appendix E.2 in the revised manuscript*. The performance gap can be attributed to two key factors:
> - (1) **Optimization Dynamics.** In larger networks (e.g., XL/2), the representational capacity increases substantially, amplifying the impact of optimization stability. As shown in *Figure 8  in Appendix E.1 in the revised manuscript*, MeanFlow exhibits higher variance and less stable loss behavior during training, whereas ESC maintains stable optimization and is therefore more likely to converge to a better solution. In smaller models (e.g., B/2), the representational capacity is nearly saturated, leaving limited room for further improvement. In contrast, for larger models, ESC’s improved stability enables it to better exploit the additional capacity, resulting in more noticeable performance gains.
> -  (2) **Statistical Generalization.** As the parameter space dimensionality increases, gradient noise also grows, making variance-reduction mechanisms such as EMA, momentum, or the proposed plug-in velocity more beneficial. This observation aligns with the theoretical intuition in [1.Kaplan et al. (2020)], where the generalization gap (or overfitting) is linked to the variance term scaling. Within the scaling law framework, bias dominates in smaller models, while variance becomes the main factor as the model scales up. To illustrate this, we compare the FID convergence curves of ESC-B/2 vs. MeanFlow-B/2 (trained for 600k iterations) and ESC-XL/2 vs. MeanFlow-XL/2 (trained for 1.2M iterations), as shown in *Figure 9 in Appendix E.2 in the revised manuscript*. Empirically, in the smaller B/2 setting, both methods converge rapidly to similar FID values. However, in the larger XL/2 model, MeanFlow’s FID curve plateaus in the later training stages, while ESC continues to improve and reaches 2.85. This suggests that in large-scale models, variance dominates generalization behavior, and the variance reduction introduced by plug-in velocity significantly enhances final performance.
> ### 2.  Response to *"In Figure 3(b) and (c), several comparison curves included in Figure 3(a) are missing. What is the reason for this omission?"*:
>   Regarding the missing models in Figure 2, the reasons are explained in *Appendix D.1*.  First, compared to Figure 2(a), in Figure 2(b), the CT and CT-linear models are omitted because, as stated in *Lines 2036-2037 in the original submission*, which is the same as *Lines 2129–2130 in the revised submission*: *"Since CTs require LPIPS as their loss metric, replacing it with a squared L2 loss on latents causes training to diverge."*  Second, in Figure 2(c), the IMM and sCT models are not included. For sCT, as noted in *Line 2063  in the original submission*, which is the same with *Line 2157 in the revised manuscript* : *"As it (CFG-training) is proposed to bridge the instantaneous velocity and average velocity under classifier-free guidance, applying this technique directly to sCT, which models the instantaneous velocity, is not entirely straightforward."* For IMM, *Line 2056  in the original submission*, which corresponds the *Line 2159 in the revised manuscript* explains: *"Applying CFG requires two NFEs during inference to compute $v^{cfg}$. As our focus is on one-step generation (i.e., 1-NFE), we therefore do not include IMM in the comparisons."*
>
> [1] Kaplan et al. (2020): Jared Kaplan et. al, Scaling Laws for Neural Language Models, 2020, https://arxiv.org/abs/2001.08361

---

> ### Comment · Reviewer_enqR · 2025-11-28
> **response to rebuttal**
>
> Dear authors, I appreciate your direct rebuttal. Some of my concerns are addressed, e.g., FID results and convergence. I'm considering increasing my score. Your rebuttals are appreciated.

---

### Official Review · Reviewer_jD45 · 2025-11-05

**Soundness:** 4
**Presentation:** 4
**Contribution:** 3
**Rating:** 8
**Confidence:** 4

**Summary:**

This paper first summarizes and presents the design space of shortcut flow paths in diffusion models for one-step generation, such as consistency model, shortcut diffusion, MeanFlow, etc. The authors further propose plug-in velocity along with multiple pratical techniques to improve training of continuous-time shortcut models (SC), and enhance generation performance of shortchut models in their experiments.

**Strengths:**

+ The paper presents a formalization of shortcut models within a unified framework. This can provide a valuable foundation for subsequent work.
+ The paper elucidates the design space of SC models. Both the mathematical formulation and the overall writing quality are presented with clarity.
+ The paper makes several theoretical contributions that will likely benefit future research. These include: i) A Wasserstein distance bound for the objectives of discrete-time (DT-SC) and continuous-time (CT-SC) models. ii) An inference error bound for both CT-SC and DT-SC in terms of the variance of the average velocity. iii) The optimality of linear paths for SC models under Fisher information metrics.

**Weaknesses:**

- The empirical results indicate that the proposed plug-in velocity yields marginal performance gains, suggesting that its practical benefits over existing methods may be limited.

- The experimental comparison would be strengthened by the inclusion of other state-of-the-art baselines, such as rectified flow and reflow, for a more comprehensive evaluation.

- The improvement techniques presented appear to be specific to the MeanFlow architecture. The paper's impact could be broadened by exploring the generalizability of these techniques to other SC models. For example, the authors might include a discussion or experimental analysis on the selection of loss metrics for different SC model variants.

**Questions:**

1. Regarding the multi-GPU training implementation, are the batches gathered across all devices to compute a plug-in velocity, or is the plug-in velocity computed locally on each device's batch?

2. Do the authors investigate the impacts of the number of samples used for calculating the plug-in velocity? For instance, setting computational efficiency aside, does increasing the sample size lead to further performance gains?

3. Table 3 indicates a significant performance gap between the MeanFlow-based approaches and the other methods benchmarked. Could the authors offer an explanation or formalize the reasons for this observation?

---

> ### Author Response · Authors · 2025-11-17
> **Response to Weaknesses**
>
> We sincerely thank you for your recognition of the value of our work, including theoretical justification, empirical analysis and technical improvements. Additional figures are appended at the end of the newly updated pdf.
>
> ## **Response to Weaknesses**
> ### 1. Response to *"The proposed plug-in velocity yields marginal performance gains."*:
>   The performance gain under SiT-B/2 architecture is marginal, while it is significant under SiT-XL/2, i.e. over 0.5 in absolute improvements  and over 15% in relative improvements in FID. We argue that the proposed training improvements can benefit the larger model more, so that its application is extensive. Further, we give a detailed explanation of the phenomenon in *Appendix E.2 in the newly updated revised manuscript*, attributing the low training variance to the key to improvements for learning a large model.
>
> ### 2. Response to *"Experimental comparison would be strengthened by the inclusion of other state-of-the-art baselines, such as rectified flow and reflow, for a more comprehensive evaluation."*:
>   Due to space limitations, we compare several representative one-step models in the main text. In *Table.6&7 in Appendix D.6 in the updated revised manuscript*, we further include comparisons with the current state-of-the-art diffusion, autoregressive, and distillation-based models for completeness. We plan to include more comprehensive method comparisons, if could let us know the references.
>
> ### 3. Response to *"The paper's impact could be broadened by exploring the generalizability of these techniques to other SC models."*:
>   Thank you for the constructive suggestions that the impact of our paper could be further broadened by "exploring the generalizability of these techniques to other shortcut models". Since our paper includes extensive modular decomposition and performance comparisons across a wide range of methods, it is impractical to perform with/without plug-in velocity evaluations for all models under limited computational resources. Instead, we identify and optimize the most representative baseline model through theoretical and empirical analysis. We will consider extending the plug-in velocity evaluation to a broader set of models as part of our future work. In particular, we have expanded the discussion in the *“Future Work” section in the updated revised manuscript* under "Generalization to downstream tasks and more models" to explicitly address this direction.

---

> ### Author Response · Authors · 2025-11-17
> **Response to Questions**
>
> ## **Response to Questions**
> ### 1. Response to *"Is the plug-in velocity computed locally on each device's batch?"*:
>   It is computed locally on each device's batch, as stated in *Line 432 of the original submission: “In multi-GPU training, the class labels of mini-batches across different processes are independent of each other.”* This is also reflected in *Line 136 of loss/esc_loss.py in the updated code base (https://anonymous.4open.science/r/ExplicitShortCut-00EE/loss/esc_loss.py)*. Since the plug-in velocity is computed within the model, and in a multi-GPU setting each model processes different mini-batches, the computation is therefore performed independently on each device in practice.
>
> ### 2. Response to *"Do the authors investigate the impacts of the number of samples used for calculating the plug-in velocity?"*:
>   We have conducted additional experiments on CIFAR-10 with batch sizes of 512 and 1024 to compare the effects of different sample counts used for computing the plug-in velocity. All experiments were performed on 8×A100 GPUs. When the batch size was 512, each GPU computed the plug-in velocity using 64 samples; when the batch size was 1024, the number of samples per GPU increased to 128. Under the same experimental settings as in the paper, the results are summarized in the table below:
> | Batch size | MeanFlow | ESC | Improvement (%) |
> |----|----|------|-----|
> | 512 | 3.15 | 3.08 | 2.2 |
> | 1024 | 2.92 | 2.84 | 3.0 |
>
>   Note that ESC is implemented as MeanFlow with plug-in velocity and other tricks removed.
>   Further, we observe that increasing the batch size improves performance, which is consistent with our theoretical observation that the variance decreases as $O(1 - 1/B)$ while the bias increases as $O(1/B)$. Although the improvements over MeanFlow appear modest, they are consistent and meaningful, considering that MeanFlow already performs very effectively on CIFAR-10. Moreover, *Figure 8 in the Appendix E.1* in the revised manuscript, further illustrates that incorporating the plug-in velocity stabilizes the training of MeanFlow when trained with a batch size of 512.  A similar phenomenon is also observed when the batch size is 1024; to avoid redundancy, we do not repeat the figure here. We believe these results clearly demonstrate the robustness and effectiveness of the proposed plug-in velocity techniques.
>
>
> ### 3. Response to *"Could the authors offer an explanation or formalize the reasons for this observation of a significant performance gap between the MeanFlow-based approaches and the other methods benchmarked?"*:
>   As discussed in Section 3, the observed performance advantage can be attributed to the following factors:
> - (1). Linear path vs. cosine path. In the shortcut model setting, the linear path outperforms the cosine path. We also theoretically justify in *Appendix C.3* that linear paths are optimal under Fisher information metrics in shortcut models, whereas cosine paths are optimal in diffusion and flow-matching settings. In the table, MeanFlow, SCD, and IMM adopt linear paths, while iCT employs a cosine path. This explains the first source of the performance gap.
> - (2). Continuous-time shortcut models usually exhibit lower inference error. As shown in *Theorem 2.2 and Figure 2(a)*, we provide a proof of the corresponding error bound and empirical demonstration. Since MeanFlow is a continuous-time model, it inherently benefits from a smaller inference error, which further enhances its performance.
> - (3). The importance of classifier-free guidance (CFG). As demonstrated in *Figures 2(b) and 2(c)*, CFG plays a crucial role. MeanFlow integrates the CFG signal directly into the training process, enabling it to achieve CFG-level inference performance in a single step, which significantly improves image FID.
>
> Taken together, these factors, supported by extensive theoretical analysis and empirical evidence in the paper, demonstrate that MeanFlow combines multiple advantages, resulting in its superior performance. We appreciate the reviewer's insightful comments and would be glad to further discuss any additional questions or perspectives regarding this point.

---

### Author Response · Authors · 2025-11-17
**General Response**

## General Response
We thank the reviewers for their thorough and constructive comments. We are glad that the reviewers agree that our  formulation of the designing space of shortcut models is valuable (reviewers **jD45, enqR, YqvP, rWuT, k1TZ**), empirical and theoretical analysis is thorough  (reviewers **jD45, enqR, YqvP, rWuT, k1TZ**), proposed improvements are solid and novel (reviewers **jD45, rWuT**), our experimental results are convincing and promising (reviewers **enqR, k1TZ**) , and our presentation is clear (reviewers **jD45, enqR, YqvP, rWuT**). Reviewers also pointed out doubts about the convergence of few-step generation, training stability comparison, utilization of MeanFlow as a baseline, and more SOTA methods' comparison. Based on the reviewers' valuable feedback, we have conducted additional experiments, which hopefully resolve the reviewers' concerns. We have updated the manuscript, where we highlight modifications with dark red color.

In detail, we carefully organize the major concerns raised across reviews and summarize the common questions below, along with our point-by-point clarifications. We hope these responses address the reviewers' doubts and provide a clearer understanding of our work.

### 1. Convergence of 2-step generation vs. 1-step generation
Reviewers **enqR**, **YqvP**, and **k1TZ** raised questions related to the limitation stated in our original submission: namely, that 2-step generation initially converges more slowly than 1-step generation. After the submission, we extended the training iterations and found that **achieving high-fidelity few-step generation simply requires additional training iterations**, allowing the model to better capture the full probabilistic trajectory.
As shown in *Table 8 in Appendix D.8* of the revised manuscript, where the results confirm that increasing training iterations enables the model to progressively learn the underlying probabilistic path, leading to **substantial improvements in few-step generation quality**, ultimately reversing the initial gap.

### 2. Impact of improvement techniques on training stability
Reviewers **jD45** and **YqvP** asked how the introduced improvement techniques affect training stability.
To clarify this, we include additional results in the revised manuscript:
- *Figure 8 in Appendix E.1* shows that plug-in velocity **stabilizes the training of continuous-time shortcut models** when trained with a batch size of 512.
- *Figure 9 in Appendix E.2* further shows that **ESC consistently outperforms MeanFlow throughout training**, as reflected by the sample FID curves.
These results demonstrate that the proposed techniques improve, not compromise, training stability.

### 3. Why the Section 4 improvements are evaluated on the MeanFlow baseline
Reviewers **jD45** and **YqvP** also asked why technical improvements in  Section 4 are evaluated under the MeanFlow baseline.
This choice is fully intentional and clearly stated in the paper:
- Section 3 theoretically demonstrates the superiority of **linear paths** and the **continuous-time shortcut model**, whose combination **leads directly to MeanFlow**.
- Accordingly, at the beginning of Section 4 (*Lines 369–372 in the original submission, corresponding to Line 399-401 in the revised manuscript*), we write:
“*Building on the above analysis, all subsequent techniques and developments will be carried out under the continuous-time shortcut model with linear paths, so we choose MeanFlow with the SiT-B/2 architecture as our baseline implementation.*”
Thus, evaluating the improvements under the MeanFlow baseline directly follows from the theoretical findings established earlier.

Additionally, because the proposed techniques are general, considering extending their evaluation to a broader set of models will be parts of our future work. Since our paper includes extensive modular decomposition and performance comparisons across a wide range of methods, it is impractical to perform with/without plug-in velocity evaluations for all models under limited computational resources. Therefore,  identifying and optimizing the most representative baseline model through theoretical and empirical analysis is reasonable.

### 4. Adding more baseline models for comparison
Reviewer **jD45** and **YqvP** asked to add more comparison with models including auto-regressive and flow-based generative models, as well as distillation-based few-step models.
Due to space limitations, we compare several representative one-step models in the main text. In *Table.6&7 in Appendix D.6 in the updated revised manuscript*, we further include comparisons with the current state-of-the-art diffusion, autoregressive, and distillation-based models for completeness. We plan to include more comprehensive method comparisons, if could let us know any appropriate reference.

### 5. Other comments
Additional reviewer-specific concerns are addressed individually in our detailed point-by-point responses.

---

### Meta-Review · Area_Chair_Jfmq · 2025-12-03

**Summary:**

This paper presents a framework for one-step diffusion models (shortcut models), providing an empirical and theoretical analysis of the design space, proposing improvements to training, and demonstrating state-of-the-art performance through scaling-up evaluation. The initial reviews were generally positive, with three reviewers (jD45, rWuT, k1TZ) recommending acceptance (score 8) and two reviewers (enqR, YqvP) giving scores of 4. The primary concerns raised by reviewers included questions about the limited practical gain and novelty of the proposed plug-in velocity technique, the need for more comparisons with state-of-the-art baselines (especially distillation-based methods), the rationale for focusing improvements primarily on the MeanFlow baseline, the slower convergence of 2-step generation compared to 1-step, and the theoretical novelty of the unifying framework itself. In the rebuttal and subsequent discussion, the authors provided comprehensive point-by-point responses. They conducted additional experiments showing that extended training resolves the 2-step convergence issue, included further comparisons with other models in the appendix, clarified the theoretical and empirical justification for focusing on MeanFlow, and added analysis on training stability. They also engaged in a detailed discussion regarding the framework's foundational prior work. The rebuttal successfully addressed most key concerns. The consensus after discussion is positive. As a result, this paper is recommended for acceptance.

**Reviewer Concerns:**

Most significant reviewer concerns, including those regarding convergence, empirical comparisons, training stability, and framework justification, were convincingly addressed by the authors' rebuttal and revised manuscript. Minor points about presentation and algorithm pseudocode were also corrected.

**Reviewer Scores:**

Reviewer jD45 provided a positive review with minor requests. Given the thorough responses, their score would likely remain. Reviewer enqR (initial score 4) indicated consideration to increase their score after the rebuttal, so a final score of 6 is plausible. Reviewer YqvP (initial score 4) explicitly raised their score to 6 after the discussion. Reviewer rWuT (initial score 8) raised a point about the magnitude of improvement, which was clarified; their score would likely remain an 8. Reviewer k1TZ (initial score 8) confirmed their score remained the same after the rebuttal.

---

### Decision · Program_Chairs · 2026-01-26

Accept (Poster)